# Stability and Generalization of Asynchronous SGD: Sharper Bounds Beyond Lipschitz and Smoothness

**Xiaoge Deng**    **Tao Sun**$^*$    **Shengwei Li**    **Dongsheng Li**$^*$    **Xicheng Lu**

College of Computer Science and Technology
National University of Defense Technology, China
dengxg@nudt.edu.cn, suntao.saltfish@outlook.com, lucasleesw9@gmail.com
dsli@nudt.edu.cn, xclu@nudt.edu.cn

## Abstract

Asynchronous stochastic gradient descent (ASGD) has evolved into an indispensable optimization algorithm for training modern large-scale distributed machine learning tasks. Therefore, it is imperative to explore the generalization performance of the ASGD algorithm. However, the existing results are either pessimistic and vacuous or restricted by strict assumptions that fail to reveal the intrinsic impact of asynchronous training on generalization. In this study, we establish sharper stability and generalization bounds for ASGD under much weaker assumptions. Firstly, this paper studies the on-average model stability of ASGD and provides a non-vacuous upper bound on the generalization error, without relying on the Lipschitz assumption. Furthermore, we investigate the excess generalization error of the ASGD algorithm, revealing the effects of asynchronous delay, model initialization, number of training samples and iterations on generalization performance. Secondly, for the first time, this study explores the generalization performance of ASGD in the non-smooth case. We replace smoothness with the much weaker Hölder continuous assumption and achieve similar generalization results as in the smooth case. Finally, we validate our theoretical findings by training numerous machine learning models, including convex problems and non-convex tasks in computer vision and natural language processing.

## 1   Introduction

The last decade has witnessed explosive growth in the scale of models and datasets in the machine learning (ML) community [9, 12]. In light of this tendency, asynchronous distributed optimization has become crucial to ensure efficient training of large-scale ML models [4]. Specifically, the *asynchronous stochastic gradient descent* (ASGD) algorithm eliminates the synchronization barrier between the distributed training workers, enabling each worker to independently perform idle-free asynchronous gradient updates, thereby accelerating model training. Despite this asynchronous updating introduces delays that result in model inconsistency, the convergence of ASGD is still guaranteed under some mild assumptions [1, 26, 42, 40, 19, 44].

An intriguing observation is that ML models learned by stochastic gradient descent (SGD) [35] not only achieve zero training error but also demonstrate good generalization performance on unknown test datasets [52]. Generalizability is a classical topic in the statistical ML fields, and associated analytical techniques include VC dimension [46], Rademacher complexity [20], PAC-Bayesian [27], uniform convergence [29, 31], information-based, and compression-based bounds [3, 50]. In this paper, we are going to study generalizability in the sense of algorithmic stability [8]. This stability-based analytical framework allows bypassing the model dimensionality so that we can focus on

---

$^*$Corresponding authors

38th Conference on Neural Information Processing Systems (NeurIPS 2024).

exploring the generalization properties of optimization algorithms. Hardt et al. [17] investigated the generalization error of SGD on the basis of algorithmic uniform stability. Assuming that the loss function is convex, $L$-Lipschitz and $\beta$-smooth, and running SGD for $K$ iterations with a learning rate $\eta_k < 2/\beta$, they obtained an upper bound on the generalization error of $\mathcal{O}(L^2 \sum_{k=1}^{K} \eta_k/n)$, where $n$ represents the total number of training samples. In a recent work [23], they proposed the on-average model stability and established a tighter generalization bound of $\mathcal{O}(1/n)$ for low-noise settings, without requiring the $L$-Lipschitz assumption.

Research on the generalization of asynchronous stochastic gradient descent algorithms mainly concentrates on parsing the effect of asynchronous delay $\tau$ on algorithm stability and generalization. Leveraging the algorithmic uniform stability tool, Regatti et al. [33] presented an upper generalization error bound of $\mathcal{O}(L^2 K^{\beta\tau}/n\beta\tau)$ in the non-convex case, assuming $L$-Lipschitz, $\beta$-smooth functions, and a decreasing learning rate. However, empirical experiments [13] show that this bound is too loose to reflect the effect of asynchronous delay on algorithmic stability accurately. Deng et al. [13] directed their attention towards convex quadratic functions and established an upper bound on the generalization error as $\widetilde{\mathcal{O}}((K-\tau)/n\tau)$ by utilizing the algorithmic average stability [36]. This bound suggests that the introduced asynchronous delays can enhance algorithm stability, consequently improving its generalization performance under appropriate learning rates. Unfortunately, the analytical technique proposed in [13] is confined to quadratic optimizations.

In this study, we delve deeper into the generalization performance of the ASGD algorithm. In particular, we utilize the on-average model stability tool to conduct a fine-grained analysis of the stability and generalization for ASGD under much weaker assumptions. Our contributions are summarized as follows.

- Without relying on the Lipschitz assumption, this study establishes the on-average model stability of ASGD and provides an upper bound on the generalization error of $\mathcal{O}(1/\overline{\tau} + 1/\sqrt{K})$. In contrast to existing work [13, 33], our results are non-vacuous and applicable to the general convex case.
- For the first time, we study the excess generalization error and provide an upper bound of $\mathcal{O}(1/\overline{\tau} + \|\mathbf{w}_1 - \mathbf{w}^*\|^2/n)$ for ASGD. Our findings demonstrate that appropriately increasing the asynchronous delay, selecting a good initial model, and increasing the number of training samples can improve the generalization performance.
- Under the much weaker $(\alpha, \beta)$-Hölder continuous gradient assumption, we establish an excess generalization error bound of $\mathcal{O}(1/\sqrt{\overline{\tau}} + \|\mathbf{w}_1 - \mathbf{w}^*\|^{\frac{4\alpha}{1+\alpha}}/\sqrt{n}^{1+\alpha})$, which reveals similar properties to the smooth case. To the best of our knowledge, this is the first study of the stability and generalization of ASGD in the non-smooth case.
- We conduct comprehensive experiments using the ASGD algorithm, covering convex optimization problems and non-convex computer vision and natural language processing tasks. Empirical evidence confirms that appropriately increasing the asynchronous delay improves the algorithm stability and reduces the generalization error, which is consistent with our theoretical findings.

## 2 Related Work

**Asynchronous training**, with origins dating back at least to [6, 45], has emerged as an essential distributed method for training modern large-scale ML tasks. It effectively addresses the synchronization bottleneck among multiple workers and mitigates the straggler problem inherent in distributed systems [4]. This study focuses on the stochastic gradient descent algorithm with asynchronous updates [1, 30]. Lian et al. [26] proved that ASGD has an asymptotic sublinear convergence rate in non-convex smooth optimization, which is consistent with SGD. Arjevani et al. [2] provided tight upper and lower complexity bounds for ASGD in convex quadratic optimization. These theoretical results were subsequently extended to general quasi-convex and non-convex settings [40]. It is noteworthy that the aforementioned theoretical analyses are based on bounded or fixed delay assumptions, whereas recent studies [11, 28] explored the performance of ASGD under arbitrary delays.

A crucial aspect of asynchronous research revolves around the interaction between learning rates and delays. For one thing, most existing theoretical analyses require the learning rate to be inversely proportional to the asynchronous delay to guarantee the convergence of the ASGD algorithm [2, 26, 40]. For another, numerous studies opt for adaptive adjustments of the learning rate based on varying asynchronous delays to improve the convergence rate of ASGD [34, 38, 49, 53]. The dependence of

learning rate on asynchronous delay also influences the stability and generalization studies of ASGD presented in this paper.

**Algorithm stability** originated from perturbation analysis [7], which measures the difference in the algorithm's output from changing a single input training sample. Generalization error refers to the performance disparity of the output model between training and testing datasets. Hence, algorithm stability is naturally connected to generalizability [8, 16, 36]. For the mainstream SGD algorithm, extensive stability-based studies have been conducted for convex, non-convex, smooth and non-smooth cases [5, 17, 22, 23, 32, 54]. Recently, algorithm stability analysis has been extended to distributed training scenarios [48]. Considerable research has explored the generalization performance of distributed decentralized SGD from the stability perspective [14, 43, 55].

However, the current generalization studies for ASGD remain inadequate. Building upon the algorithmic uniform stability, Regatti et al. [33] presented a pessimistic generalization error bound $\mathcal{O}(K^{\hat{\tau}}/n\hat{\tau})$ of ASGD in the smooth non-convex case, where $\hat{\tau}$ represents the maximum delay. In a recent development, Deng et al. [13] established a tighter upper generalization error bound of $\mathcal{O}((K - \hat{\tau})/n\hat{\tau})$ using average stability, and Sun et al. [41] investigated a high-probability PAC-Bayesian generalization error bound $\mathcal{O}(1/\sqrt{n})$ for ASGD. However, the theoretical analyses presented in [13, 41] only hold in quadratic optimization problems, limiting their applications. To the best of the authors' knowledge, existing generalization analyses of ASGD are either pessimistic and vacuous or constrained by strict assumptions. Therefore, the objective of this study is to establish sharper stability and generalization bounds for ASGD under much milder assumptions.

# 3 Preliminaries

**Notations**. Lowercase and bold letters represent scalars and $d$-dimensional column vectors, respectively. The $\ell_2$-norm of a vector $\mathbf{x}$ is denoted by $\|\mathbf{x}\|$. Calligraphic capital letters represent mathematical sets. We write $a = \mathcal{O}(b)$ if there exists a constant $0 < c < +\infty$ such that $a \leq c \cdot b$, and $\widetilde{\mathcal{O}}(\cdot)$ hides logarithmic factors. Moreover, we denote $a \asymp b$ if $a = \mathcal{O}(b)$ and $b = \mathcal{O}(a)$.

Let $\mathcal{X} \subseteq \mathbb{R}^d$ and $\mathcal{Y} \subseteq \mathbb{R}$ denote the input and output spaces, respectively. In this study, we focus on the general supervised learning problem in ML. This task involves training a model on a data set $\mathcal{S} = \{\mathbf{z}_1, \ldots, \mathbf{z}_n\}$, where each data point $\mathbf{z}_i = (\mathbf{x}_i, y_i) \in \mathcal{Z} = \mathcal{X} \times \mathcal{Y}$ is independently and identically distributed (i.i.d.) sampled from an unknown distribution $\mathcal{D}$. We evaluate the performance of model $\mathbf{w}$ on training sample $\mathbf{z}$ with a loss function $f(\mathbf{w}; \mathbf{z})$. The training process can be formalized as learning a model parameter $\mathbf{w} \in \Omega \subseteq \mathbb{R}^d$ to minimize the empirical risk, denoted as

$$\min_{\mathbf{w} \in \Omega} F_{\mathcal{S}}(\mathbf{w}) = \frac{1}{n} \sum_{i=1}^{n} f(\mathbf{w}; \mathbf{z}_i). \tag{1}$$

SGD is the workhorse for solving the empirical risk minimization (ERM) problem (1), which iteratively updates the model parameter by $\mathbf{w}_{k+1} = \mathbf{w}_k - \eta_k \nabla f(\mathbf{w}_k; \mathbf{z}_{i_k})$.

ASGD is a powerful variant of SGD for distributed learning, which fully exploits the computational power of distributed clusters to accelerate the training process. In the distributed parameter server architecture [25], the distributed workers are responsible for computing gradients, while the model updates occur on the parameter server side. Upon receiving the gradient from a worker, the server immediately utilizes it to update the model without waiting for gradient information from other workers. The ASGD procedure is described in Algorithm 1 (located in Appendix A.1). It is noteworthy that although ASGD avoids synchronization overhead, it introduces delays in model updating. To be specific, while worker $m$ is computing and uploading the gradient, the model parameter on the server side may has already been updated by another worker $m'$. In essence, the model used for gradient computation on the worker is inconsistent with the model updated by the server. This characteristic renders ASGD a delayed gradient update, expressed as

$$\mathbf{w}_{k+1} = \mathbf{w}_k - \eta_k \nabla f(\mathbf{w}_{k-\tau_k}; \mathbf{z}_{i_k}), \tag{2}$$

where $\mathbf{w}_k, \eta_k, \tau_k$, and $\mathbf{z}_{i_k}$ denote the model parameter, learning rate, asynchronous delay, and training sample at the $k$-th iteration, respectively. It is worth noting that the index $i_k$ is chosen uniformly at random from the set $\{1, \ldots, n\}$.

For the model $\mathbf{w}$ learned through ASGD by minimizing the empirical risk (1) on the training data set $\mathcal{S}$, people are more concerned with its performance on the unknown distribution $\mathcal{D}$, i.e., the following popular risk

$$F(\mathbf{w}) = \mathbb{E}_{\mathbf{z} \sim \mathcal{D}}[f(\mathbf{w}; \mathbf{z})]. \tag{3}$$

The empirical risk (1) and the popular risk (3) of a model are not the same, and the difference between them is referred to generalization error. More formally, denote the model learned by algorithm $A$ on data set $\mathcal{S}$ as $A(\mathcal{S})$, and its *generalization error* is defined as

$$\epsilon_{\mathrm{gen}} := \mathbb{E}_{\mathcal{S}, A}\left[F(A(\mathcal{S})) - F_{\mathcal{S}}(A(\mathcal{S}))\right]. \tag{4}$$

The expectation here is taken over the randomness of the algorithm and the training data. This study is dedicated to bounding $\epsilon_{\mathrm{gen}}$ by algorithmic stability. Let

$$\mathcal{S}' = \{\mathbf{z}'_1, .., \mathbf{z}'_n\}, \quad \mathcal{S}^{(i)} = \{\mathbf{z}_1, .., \mathbf{z}_{i-1}, \mathbf{z}'_i, \mathbf{z}_{i+1}, .., \mathbf{z}_n\}. \tag{5}$$

$\mathcal{S}'$ is also a data set i.i.d. sampled from the unknown distribution $\mathcal{D}$, but is independent of the data set $\mathcal{S} = \{\mathbf{z}_1, \ldots, \mathbf{z}_n\}$. $\mathcal{S}^{(i)}$ is a perturbed data set formed by replacing the $i$-th sample in $\mathcal{S}$ with $\mathbf{z}'_i$. Based on these notations, Lei and Ying [23] defined the following *on-average model stability*.

**Definition 1** (On-average model stability). A randomized algorithm $A$ is on-average model $\epsilon_{\mathrm{stab}}$-stable if

$$\mathbb{E}_{\mathcal{S}, \mathcal{S}', A}\left[\frac{1}{n}\sum_{i=1}^{n}\left\|A(\mathcal{S}) - A(\mathcal{S}^{(i)})\right\|^2\right] \leq \epsilon_{\mathrm{stab}}.$$

Leveraging the smoothness (Definition 2) assumption, the connection between this algorithmic stability and the generalization error $\epsilon_{\mathrm{gen}}$ is established in the following lemma [Theorem 2, [23]].

**Lemma 1.** *Let $\gamma > 0$. Assume that the function $\mathbf{w} \mapsto f(\mathbf{w}; \mathbf{z})$ is non-negative and $\beta$-smooth for any $\mathbf{z} \in \mathcal{Z}$. Then, if algorithm $A$ is on-average model $\epsilon_{\mathrm{stab}}$-stable, the generalization error satisfies*

$$\mathbb{E}_{\mathcal{S}, A}\left[F(A(\mathcal{S})) - F_{\mathcal{S}}(A(\mathcal{S}))\right] \leq \frac{\beta}{\gamma}\mathbb{E}_{\mathcal{S}, A}[F_{\mathcal{S}}(A(\mathcal{S}))] + \frac{\beta + \gamma}{2}\epsilon_{\mathrm{stab}}.$$

While the smooth function assumption is common in optimization and generalization analyses [13, 17, 33, 40], it does impose constraints on the applicability [5]. For instance, the hinge loss, which is widely used in the ML fields, does not satisfy the smooth property. In this paper, therefore, we also investigate the stability of ASGD under the much weaker Hölder continuous gradient assumption (Definition 3), so as to establish broader and fine-grained generalization results. With the Hölder continuous condition, stability and generalization can be connected similarly to Lemma 1.

**Lemma 2** (Theorem 2, [23]). *Let $\gamma > 0$. For any $\mathbf{z} \in \mathcal{Z}$, the function $\mathbf{w} \mapsto f(\mathbf{w}; \mathbf{z})$ is non-negative, convex, and the gradient $\nabla f(\mathbf{w}; \mathbf{z})$ is $(\alpha, \beta)$-Hölder continuous. If algorithm $A$ is on-average model $\epsilon_{\mathrm{stab}}$-stable, then the generalization error satisfies*

$$\mathbb{E}_{\mathcal{S}, A}\left[F(A(\mathcal{S})) - F_{\mathcal{S}}(A(\mathcal{S}))\right] \leq \frac{c_{\alpha,\beta}^2}{2\gamma}\mathbb{E}_{\mathcal{S}, A}[F^{\frac{2\alpha}{1+\alpha}}(A(\mathcal{S}))] + \frac{\gamma}{2}\epsilon_{\mathrm{stab}}.$$

*Here, $\alpha \in [0, 1]$, and $c_{\alpha,\beta}$ is a constant dependent on $\alpha, \beta$.*

Furthermore, since the generalization performance of a model is primarily reflected in the popular risk (3), this study also examines the *excess generalization error*, denoted as $\epsilon_{\mathrm{ex\text{-}gen}}$, where $\mathbf{w}^* \in \mathrm{argmin}_{\mathbf{w} \in \Omega} F(\mathbf{w})$ and

$$\epsilon_{\mathrm{ex\text{-}gen}} := \mathbb{E}_{\mathcal{S}, A}\left[F(A(\mathcal{S})) - F(\mathbf{w}^*)\right]. \tag{6}$$

**Definition 2** (Smoothness). The function $\mathbf{w} \mapsto f(\mathbf{w}; \mathbf{z})$ is $\beta$-smooth ($\beta > 0$) if for any $\mathbf{z} \in \mathcal{Z}$ and $\mathbf{w}, \mathbf{v} \in \mathbb{R}^d$,

$$\|\nabla f(\mathbf{w}; \mathbf{z}) - \nabla f(\mathbf{v}; \mathbf{z})\| \leq \beta\|\mathbf{w} - \mathbf{v}\|.$$

**Definition 3** (Hölder continuous). Let $\alpha \in [0, 1]$, $\beta > 0$. The function $\mathbf{w} \mapsto \nabla f(\mathbf{w}; \mathbf{z})$ is $(\alpha, \beta)$-Hölder continuous if for any $\mathbf{z} \in \mathcal{Z}$ and $\mathbf{w}, \mathbf{v} \in \mathbb{R}^d$,

$$\|\nabla f(\mathbf{w}; \mathbf{z}) - \nabla f(\mathbf{v}; \mathbf{z})\| \leq \beta\|\mathbf{w} - \mathbf{v}\|^{\alpha}.$$

It is noteworthy that the $(\alpha, \beta)$-Hölder continuous gradient is equivalent to a $\beta$-smooth function when $\alpha = 1$. Whereas $\alpha = 0$ implies that the function gradient is bounded, i.e., there exists a constant $L > 0$ such that $\|\nabla f(\mathbf{w}; \mathbf{z})\| \leq L$. Although many analyses of ASGD are grounded on the bounded gradient condition [26, 28, 33], this assumption is somewhat unrealistic [24]. Notably, our analysis of the algorithm stability and the generalization error does not rely on the bounded gradient assumption.

# 4    Stability and Generalization Bounds

This section explores the stability and generalization of the ASGD algorithm in the context of smooth loss functions, and the proof is given in Appendix B. Firstly, we present the assumption required for this study.

**Assumption 1.** The parameter space $\Omega \subseteq \mathbb{R}^d$ is a bounded convex set. Then, for any $\mathbf{w}, \mathbf{v} \in \Omega$, there exists a constant $r > 0$ such that $\|\mathbf{w} - \mathbf{v}\| \leq r$.

Assumption 1 is standard in analyzing SGD and its variants, as it is easy to hold with the projection operator [5, 17, 23, 30, 43]. More specifically, we consider the following projected ASGD updates

$$\mathbf{w}_{k+1} = \Pi_\Omega\big(\mathbf{w}_k - \eta_k \nabla f(\mathbf{w}_{k-\tau_k}; \mathbf{z}_{i_k})\big). \tag{7}$$

Since the projection operator $\Pi_\Omega$ is non-expansive, it has no impact on the stability and generalization analysis of the ASGD algorithm.

**Remark 1.** Let $\mathbf{w}_k$ and $\mathbf{w}_k^{(i)}$ denote the models produced by ASGD (7) after $k$ iterations on the datasets $\mathcal{S}$ and $\mathcal{S}^{(i)}$ (defined in (5)), respectively. According to Assumption 1, it follows that $\|\mathbf{w}_k - \mathbf{w}_k^{(i)}\| \leq r$. Notably, this result is intuitively understandable as the datasets $\mathcal{S}, \mathcal{S}^{(i)}$ differ only by a single sample, and the initialization is the same ($\mathbf{w}_1 = \mathbf{w}_1^{(i)}$). In contrast to a recent work [55], where the authors assumed a normal distribution with bounded mean and variance for the difference between models $\mathbf{w}_k$ and $\mathbf{w}_k^{(i)}$, our study does not necessitate such a strong assumption.

## 4.1    Algorithmic Stability of ASGD

The stability-based analysis of SGD hinges significantly on the non-expansiveness of the gradient update operator [17, 23]. Namely, if function $f$ is convex and smooth, then $\forall \mathbf{w}, \mathbf{v} \in \Omega, \mathbf{z} \in \mathcal{Z}$

$$\|\mathbf{w} - \eta \nabla f(\mathbf{w}; \mathbf{z}) - (\mathbf{v} - \eta \nabla f(\mathbf{v}; \mathbf{z}))\| \leq \|\mathbf{w} - \mathbf{v}\|.$$

However, this well-posed property is no longer applicable in the context of asynchronous gradient updates. To address this issue, we present the following critical lemma to bound the delayed gradient update operator.

**Lemma 3.** *Let the loss function be convex, $\beta$-smooth, and Assumption 1 holds. Denote $\mathbf{w}_k$ and $\mathbf{w}_k^{(i)}$ as the models produced by ASGD (7) with learning rates $\eta_k \leq 2/\beta$ for $k$ iterations on the datasets $\mathcal{S}$ and $\mathcal{S}^{(i)}$, respectively. Then*

$$\left\|\mathbf{w}_k - \eta_k \nabla f(\mathbf{w}_{k-\tau_k}; \mathbf{z}_{i_k}) - \big(\mathbf{w}_k^{(i)} - \eta_k \nabla f(\mathbf{w}_{k-\tau_k}^{(i)}; \mathbf{z}_{i_k})\big)\right\|^2 \leq \left\|\mathbf{w}_k - \mathbf{w}_k^{(i)}\right\|^2 + 2\eta_k \beta^2 r^2 \sum_{j=1}^{\tau_k} \eta_{k-j}.$$

By leveraging the properties established in Lemma 3, we can demonstrate an approximately non-expansive recursive property for $\|\mathbf{w}_{k+1} - \mathbf{w}_{k+1}^{(i)}\|^2$ and subsequently establish the on-average model stability (Definition 1) of the ASGD algorithm as follows.

**Theorem 1** (Stability). *Suppose the loss function is non-negative, convex, and $\beta$-smooth. Let Assumption 1 holds. If we run ASGD (7) with a non-increasing learning rate $\eta_k \leq 1/2\beta$ for $k$ iterations, then the on-average model stability satisfies (e is the natural constant)*

$$\epsilon_{\text{stab}} = \frac{16\beta e(1 + k/n)}{n}\left[\eta_1 \|\mathbf{w}_1 - \mathbf{w}^*\|^2 + \big(4\beta r^2 + 2F(\mathbf{w}^*)\big)\sum_{l=1}^{k}\eta_l^2\right] + 2\beta^2 r^2 e \sum_{l=1}^{k}\eta_l \sum_{j=1}^{\tau_l}\eta_{l-j}.$$

In line with the findings of study [17], increasing the number of training iterations impairs the stability of ASGD. Compared to SGD [23], we introduce an additional term $\mathcal{O}(\sum_{l=1}^{k}\eta_l \sum_{j=1}^{\tau_l}\eta_{l-j})$ to characterize the effect of asynchronous delay on the stability of ASGD. Also similar to the data-dependent stability study [22], Theorem 1 indicates that model initialization affects the algorithmic stability, i.e., selecting a better model initiation point $\mathbf{w}_1$ can effectively improve the stability.

## 4.2    Generalization Error Bounds

Together with Lemma 1 and Theorem 1, we can now present the generalization error (4) of the ASGD algorithm under smooth conditions.

**Theorem 2** (Generalization error). *Let Assumption 1 holds, and assume that the loss function is non-negative, convex, and $\beta$-smooth. Running ASGD (7) with a non-increasing learning rate $\eta_k \leq 1/2\beta$ for $K$ iterations, then the generalization error is given by*

$$\epsilon_{\text{gen}} = \mathcal{O}\left( \mathbb{E}_{\mathcal{S},A}[F_{\mathcal{S}}(\mathbf{w}_K)] + \sum_{k=1}^{K} \eta_k \sum_{j=1}^{\tau_k} \eta_{k-j} + \frac{1+K/n}{n}\left[\eta_1\|\mathbf{w}_1 - \mathbf{w}^*\|^2 + \left(1 + F(\mathbf{w}^*)\right)\sum_{k=1}^{K}\eta_k^2\right]\right).$$

This finding suggests that both the model initialization and optimization processes have an impact on the generalization performance. In practical applications, one can reduce the generalization error by selecting a good initial model $\mathbf{w}_1$ to start the training task. Additionally, it is crucial to finish the optimization process promptly since too many training iterations can detrimentally affect the generalization performance.

Furthermore, Theorem 2 reveals a close relationship between the generalizability of ASGD and the learning rate. As discussed in Section 2, asynchronous training typically utilizes delay-inverse correlated learning rates to ensure algorithmic performance. In the low-noise case, namely, $F(\mathbf{w}^*) = 0$, Stich and Karimireddy [40] demonstrated that $F_{\mathcal{S}}(\mathbf{w}_K) = \mathcal{O}(1/\sqrt{K})$ for ASGD under the conditions of smooth and general quasi-convex loss functions, with a learning rate of $\eta_k = c(\tau\sqrt{K})^{-1}$. Employing this learning rate strategy, the following corollary can be derived.

**Corollary 1.** *Let $F(\mathbf{w}^*) = 0$, $K \asymp n$, and the conditions specified in Theorem 2 hold. If we set the learning rate $\eta_k = c(\overline{\tau}\sqrt{K})^{-1}$ with a constant $c > 0$ and $\overline{\tau} = \sum_{k=1}^{K}\tau_k/K$, then the generalization error satisfies*

$$\mathbb{E}_{\mathcal{S},A}\left[F(\mathbf{w}_K) - F_{\mathcal{S}}(\mathbf{w}_K)\right] = \mathcal{O}\left(\frac{1}{\overline{\tau}} + \frac{1}{\sqrt{K}}\right).$$

At this point, although the asynchronous training also introduces an additional generalization error term of $\mathcal{O}(1/\overline{\tau})$, increasing the delay can instead mitigate this detriment. Unlike previous ASGD generalization research [14, 33], this study does not rely on the Lipschitz assumption. In contrast to the vacuous upper bound of $\mathcal{O}(K^{\hat{\tau}}/n\hat{\tau})$ in [33], we provide a sharper result and demonstrate that increasing the asynchronous delay reduces the generalization error. While Deng et al. [13] present a similar result $\mathcal{O}((K-\hat{\tau})/n\hat{\tau})$ with respect to the maximum delay $\hat{\tau}$ in the convex quadratic optimization, our bound holds in general convex settings. Furthermore, our results are associated with the average delay $\overline{\tau}$ rather than the pessimistic maximum delay $\hat{\tau}$ in [13, 14, 33].

### 4.3 Excess Generalization Error

According to definitions (4) and (6), the excess generalization error $\epsilon_{\text{ex-gen}}$ can be decomposed as

$$\epsilon_{\text{ex-gen}} = \epsilon_{\text{gen}} + \mathbb{E}_{\mathcal{S},A}\left[F_{\mathcal{S}}(A(\mathcal{S})) - F_{\mathcal{S}}(\mathbf{w}^*)\right], \tag{8}$$

where the second term is known as the optimization error. The analysis of optimization error for ASGD usually requires the following bounded gradient assumption [26, 28, 33].

**Assumption 2.** The gradient $\mathbf{w} \mapsto \nabla f(\mathbf{w}; \mathbf{z})$ is bounded. That is, for any $\mathbf{w} \in \Omega, \mathbf{z} \in \mathcal{Z}$, there exists a constant $L > 0$ such that $\|\nabla f(\mathbf{w}; \mathbf{z})\| \leq L$.

**Remark 2.** Assumption 2, also known as the Lipschitz condition, is used in the optimization analysis of ASGD to bound the model deviations induced by asynchronous delays, i.e., $\|\mathbf{w}_k - \mathbf{w}_{k-\tau_k}\| \leq L\sum_{j=1}^{\tau_k}\eta_{k-j}$.

For the excess generalization error of ASGD, we shift our focus to the average model $\overline{\mathbf{w}}_K := \sum_{k=1}^{K}\eta_k\mathbf{w}_k / \sum_{k=1}^{K}\eta_k$. It is noteworthy that since the parameter space $\Omega$ is a convex set, $\overline{\mathbf{w}}_K \in \Omega$ and is frequently considered as the output of the ASGD algorithm. We first present the optimization error with respect to this average model in the following lemma, followed by the excess generalization error theorem of ASGD.

**Lemma 4.** *Assuming that the loss function is non-negative, convex, and $\beta$-smooth. Let Assumptions 1 and 2 hold, if we run ASGD (7) with a non-increasing learning rate $\eta_k \leq 1/2\beta$, then the optimization error satisfies*

$$\mathbb{E}_{\mathcal{S},A}[F_{\mathcal{S}}(\overline{\mathbf{w}}_K) - F_{\mathcal{S}}(\mathbf{w}^*)] = \mathcal{O}\left(\frac{\|\mathbf{w}_1 - \mathbf{w}^*\|^2}{\sum_{k=1}^{K}\eta_k} + \left(1 + F(\mathbf{w}^*)\right)\frac{\sum_{k=1}^{K}\eta_k^2}{\sum_{k=1}^{K}\eta_k} + \frac{\sum_{k=1}^{K}\eta_k\sum_{j=1}^{\tau_k}\eta_{k-j}}{\sum_{k=1}^{K}\eta_k}\right).$$

**Theorem 3** (Excess generalization error). *Let Assumptions 1, 2 hold, and assume that the loss function is non-negative, convex, and $\beta$-smooth. Running ASGD (7) with the non-increasing learning rate $\eta_k \leq 1/2\beta$ for $K$ iterations, then the excess generalization error is*

$$
\epsilon_{\text{ex-gen}} = \mathcal{O}\Bigg( \Big[1 + \frac{\sum_{k=1}^{K} \eta_k^2}{\sum_{k=1}^{K} \eta_k}\Big] F(\mathbf{w}^*) + \frac{1 + K/n}{n}\Big[\eta_1 \|\mathbf{w}_1 - \mathbf{w}^*\|^2 + \big(1 + F(\mathbf{w}^*)\big)\sum_{k=1}^{K} \eta_k^2\Big]
$$
$$
+ \frac{\|\mathbf{w}_1 - \mathbf{w}^*\|^2}{\sum_{k=1}^{K} \eta_k} + \Big[\sum_{k=1}^{K} \eta_k\big(\eta_k + \sum_{j=1}^{\tau_k} \eta_{k-j} + \sum_{l=1}^{k} \eta_l \sum_{j=1}^{\tau_l} \eta_{l-j}\big)\Big] / \sum_{k=1}^{K} \eta_k \Bigg).
$$

Compared to the generalization error in Theorem 2, the excess generalization error is no longer explicitly dependent on the optimization error $F_{\mathcal{S}}(\mathbf{w}_K)$ and is more closely coupled to the learning rate. Considering the low-noise case $F(\mathbf{w}^*) = 0$, which is common in modern deep learning, the following corollary can be further derived.

**Corollary 2.** *Let $F(\mathbf{w}^*) = 0$, $K \asymp n$ and the conditions in Theorem 3 hold. Set the learning rate as $\eta_k = c(\overline{\tau}\sqrt{K})^{-1}$ with a constant $c > 0$ and $\overline{\tau} = \sum_{k=1}^{K} \tau_k / K$. Then if $\overline{\tau} \leq K^{\frac{1}{4}}$, the excess generalization error satisfies*

$$
\mathbb{E}_{\mathcal{S},A}\left[F(\overline{\mathbf{w}}_K) - F(\mathbf{w}^*)\right] = \mathcal{O}\Big(\frac{1}{\overline{\tau}} + \frac{\|\mathbf{w}_1 - \mathbf{w}^*\|^2}{n}\Big).
$$

To the best of our knowledge, this is the first excess generalization error result for the ASGD algorithm. Compared to Corollary 1, this generalization bound with appropriate delays is sharper and no longer relies on the optimization error result in [40].

## 5 Generalization in Non-smooth Case

This section investigates the stability and generalization of the ASGD algorithm in the context of non-smooth cases. The analysis follows a similar technical roadmap as in Section 4. Firstly, we derive the stability of ASGD by leveraging the approximately non-expansive property of the delayed gradient update operators. Then, the generalization error is given in conjunction with Lemma 2. Subsequently, we analyze the optimization process of ASGD and present the excess generalization error for the non-smooth settings.

However, without the smooth condition, the non-expansive property of asynchronous gradient updates is further compromised, and the optimization process also introduces additional errors. Under the much weaker Hölder continuous gradient assumption, We establish similar stability and generalizability results for ASGD as in the smooth case, which has not been explored in existing research. Please refer to Appendix C for the proof details of this section.

**Lemma 5.** *Let Assumption 1 holds, and assume that the loss function is non-negative, convex, and has a $(\alpha, \beta)$-Hölder continuous gradient. Then, the delayed gradient update operator satisfies*

$$
\big\|\mathbf{w}_k - \eta_k \nabla f(\mathbf{w}_{k-\tau_k}; \mathbf{z}_{i_k}) - (\mathbf{w}_k^{(i)} - \eta_k \nabla f(\mathbf{w}_{k-\tau_k}^{(i)}; \mathbf{z}_{i_k}))\big\|^2 = \|\mathbf{w}_k - \mathbf{w}_k^{(i)}\|^2 + \mathcal{O}(\eta_k \sum_{j=1}^{\tau_k} \eta_{k-j} + \eta_k^{\frac{2}{1-\alpha}}).
$$

Compared to Lemma 3, an additional term $\mathcal{O}(\eta_k^{\frac{2}{1-\alpha}})$ is introduced here to compensate for the absence of smoothness. Fortunately, since the coefficient of $\|\mathbf{w}_k - \mathbf{w}_k^{(i)}\|^2$ is not larger than 1, the delayed gradient update of ASGD remains approximately non-expansive at an appropriate learning rate. Leveraging this property, we are able to give the on-average model stability of ASGD in the non-smooth case.

**Theorem 4** (Stability). *Suppose the loss function is non-negative, convex, and has a $(\alpha, \beta)$-Hölder continuous gradient. Let Assumption 1 holds. Then, the on-average model stability of ASGD satisfies*

$$
\epsilon_{\text{stab}} = \mathcal{O}\Bigg( \frac{1 + k/n}{n}\sum_{l=1}^{k} \eta_l^2 \mathbb{E}_{\mathcal{S},A}\Big[F_{\mathcal{S}}^{\frac{2\alpha}{1+\alpha}}(\mathbf{w}_{l-\tau_l})\Big] + \sum_{l=1}^{k} \eta_l \sum_{j=1}^{\tau_l} \eta_{l-j} + \sum_{l=1}^{k} \eta_l^{\frac{2}{1-\alpha}} \Bigg).
$$

Theorem 4 shows that the algorithmic stability of ASGD not only depends on the learning rate, but is also closely related to the optimization process. Similar to [23], we replace the gradient bound (Lipschitz constant) in the uniform stability [17] with the loss function value, which leads to sharper stability and generalizability results when combined with the subsequent optimization analysis. Substituting this algorithm stability into Lemma 2 yields the generalization error of the ASGD algorithm under the Hölder continuous condition (omitted in Appendix C.3).

**Remark 3.** Although Assumption 1 and the smooth (or Hölder continuous) condition implies Lipschitz continuity, our point is to replace the upper gradient bound with function value, thereby establishing sharper stability and generalization bounds that do not depend on the Lipschitz constant.

Subsequently, we present the optimization error of ASGD in the non-smooth case, and the excess generalization error is followed by the decomposition (8).

**Lemma 6.** *Assuming that the loss function is non-negative, convex, and has a $(\alpha, \beta)$-Hölder continuous gradient. Let Assumptions 1 and 2 hold, then the optimization error of ASGD (7) with a non-increasing learning rate satisfies*

$$\mathbb{E}_{\mathcal{S},A}[F_{\mathcal{S}}(\overline{\mathbf{w}}_K) - F_{\mathcal{S}}(\mathbf{w}^*)] = \mathcal{O}\Bigg( \frac{\|\mathbf{w}_1 - \mathbf{w}^*\|^2 + \sum_{k=1}^{K} \eta_k \sum_{j=1}^{\tau_k} \eta_{k-j}^{\alpha}}{\sum_{k=1}^{K} \eta_k}$$
$$+ \frac{\left( \sum_{k=1}^{K} \eta_k^2 \right)^{\frac{1-\alpha}{1+\alpha}}}{\sum_{k=1}^{K} \eta_k} \Big[ \eta_1 \|\mathbf{w}_1 - \mathbf{w}^*\|^2 + \left(1 + F(\mathbf{w}^*)\right) \sum_{k=1}^{K} \eta_k^2 + \sum_{k=1}^{K} \eta_k^{\frac{3-\alpha}{1-\alpha}} \Big]^{\frac{2\alpha}{1+\alpha}} \Bigg).$$

**Theorem 5** (Excess generalization error). *Let Assumptions 1, 2 hold, and assume that the loss function is non-negative, convex, and has a $(\alpha, \beta)$-Hölder continuous gradient. Running ASGD (7) with the learning rate $\eta_k = c(\overline{\tau}\sqrt{K})^{-1}$ for $K \asymp n$ iterations, then if $F(\mathbf{w}^*) = 0$ and the average delay satisfies $\overline{\tau} \leq K^{\alpha'}$ with $\alpha' = \min\{\frac{1}{3}, \frac{\alpha}{3-2\alpha}\}$, the excess generalization error is*

$$\mathbb{E}_{\mathcal{S},A}[F(\overline{\mathbf{w}}_K) - F(\mathbf{w}^*)] = \mathcal{O}\Big( \frac{1}{\sqrt{\overline{\tau}}} + \frac{\|\mathbf{w}_1 - \mathbf{w}^*\|^{\frac{4\alpha}{1+\alpha}}}{\sqrt{n}^{1+\alpha}} \Big).$$

Notably, the generalization performance decreases in the non-smooth case, but the underlying properties remain consistent with the smooth setting (Corollary 2). That is, the generalization performance can be improved by choosing a good initial model, increasing the number of training samples, and appropriately adjusting the asynchronous delays. Additionally, when there is no asynchronous delay in the training system, the first term in Theorem 5 vanishes, yielding an excess generalization error bound of $\mathcal{O}(1/\sqrt{n}^{1+\alpha})$. This outcome is consistent with the findings from the study of the SGD algorithm in [23], but without requiring more computation $K \asymp n^{\frac{2}{1+\alpha}}$.

## 6   Experimental Validation

In this section, we extensively evaluated various machine learning tasks under the distributed parameter server architecture to investigate the practical stability and generalization performance of ASGD. Our experiments included convex optimization problems as well as non-convex computer vision (CV) and natural language processing (NLP) tasks. We simulated a distributed system with $M = 16$ workers and performed asynchronous training in a more general stochastic gradient descent format as follows

$$\mathbf{w}_{k+1} = \mathbf{w}_k - \eta_k \sum_{m \in \mathcal{M}_k} \mathbf{g}_{k-\tau_k}^m. \tag{9}$$

Here, $\mathcal{M}_k$ is a non-empty subset of $\{1, \ldots, M\}$ containing the workers that participated in asynchronous training at the $k$-th iteration, and $\mathbf{g}_{k-\tau_k}^m$ represents the delayed gradient computed by worker $m$ on model $\mathbf{w}_{k-\tau_k}$. Our experiments also focus on parsing the impact of asynchronous delays on algorithmic stability and generalization. Following our theoretical findings, we set the learning rate to $0.1/\overline{\tau}$ for different delays, where $\overline{\tau}$ denotes the average delay.

For the convex optimization problem, we employed a single-layer linear network with the mean squared error for a classification task on the RCV1 data set from the LIBSVM database [10]. This data set contains $20,242$ training data with $47,236$ features per sample. In the field of computer vision, we chose the popular ResNet18 model for image classification on the CIFAR10 and CIFAR100

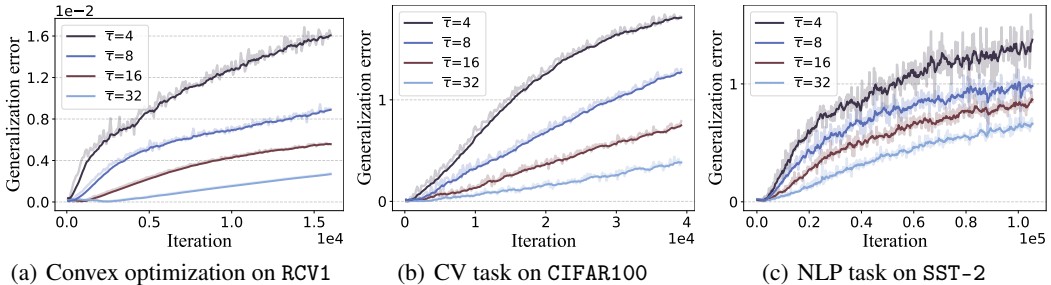

(a) Convex optimization on `RCV1`  (b) CV task on `CIFAR100`  (c) NLP task on `SST-2`

Figure 1: The generalization errors of three categories of machine learning models trained using ASGD with learning rate $\eta_k = 0.1/\overline{\tau}$. The horizontal axis denotes the number of asynchronous training iterations, and the legend represents the average delay. A degradation in generalization performance is observed as the number of training iterations increases, and the generalization performance can be improved by appropriately increasing the asynchronous delay.

datasets. ResNet [18], a convolutional neural network with residual modules and shortcut connections, has demonstrated remarkable performance across various CV tasks. `CIFAR10` and `CIFAR100` [21] are widely used image datasets, both containing $60,000$ color images of $32 \times 32$ pixels. For natural language processing tasks, we conducted experiments using BERT on the `SST-2` task within the GLUE platform [47]. BERT [15] is a pre-trained language model based on the Transformer architecture, known for its impressive performance in handling various NLP tasks. The `SST-2` [37] task in the GLUE evaluation benchmark comprises a total of $67,350$ training samples for single-sentence categorization.

Due to computational resource limitations, this experiment cannot sequentially replace a single sample to train $n$ models and calculate the on-average model stability (Definition 1). Instead, we construct a perturbed data set $\mathcal{S}^{(i)}$ by randomly removing a sample from the data set $\mathcal{S}$, and then train on the two datasets separately to record the model difference $\|A(\mathcal{S}) - A(\mathcal{S}^{(i)})\|^2$. Repeating the process multiple times, we take the average value to approximate the algorithmic stability. As for the generalization error (4), it is directly approximated by the absolute difference between the training error and the testing error of the model.

Figure 1 and Figure 2 (located in Appendix D) illustrate the generalizability and stability of the ASGD algorithm in training the three types of machine learning tasks. The experimental results show that continuous training impairs the stability and generalization of ASGD, which is consistent with the theorems presented in Sections 4 and 5. Conversely, when training with a learning rate that is inversely correlated with the asynchronous delay, an appropriate increase in the delay improves the algorithm stability and thus reduces the generalization error. This observation is in consistent with the theoretical bound in Corollary 1, which utilizes the specific learning rate $\eta_k = c/\overline{\tau}$.

## 7    Concluding Remarks

This study establishes sharper and broader stability and generalization bounds for ASGD under much weaker assumptions. We provide upper bounds for the on-average model stability and generalization error of ASGD without relying on the Lipschitz continuous condition. Moreover, for the first time, we study the stability and generalizability of ASGD in the non-smooth setting. Our generalization results are non-vacuous and applicable to the general convex case. Furthermore, we validate our theoretical findings with experiments on various machine learning tasks.

We also conducted experiments using delay-independent learning rates (Figures 3 and 4 in Appendix D). Interestingly, these results also suggest that asynchronous training is beneficial for generalization. This empirical finding challenges the pessimism of our generalization error result under constant learning rates (omitted in Appendix B.4), and motivates further exploration of the generalizability of ASGD. There are several directions for future research. The study of non-convex problems can focus on showing that asynchronous updates are approximately non-expansive even without convexity, then leading to non-vacuous stability and generalization results. Another avenue for research involves investigating tighter high probability bounds that attenuate the dominant role of the learning rate on generalization, thereby elucidating the experimental phenomena in Appendix D.

## Acknowledgments and Disclosure of Funding

This work is sponsored in part by the National Natural Science Foundation of China under Grant No. 62025208, 62421002, and 62376278, Hunan Provincial Natural Science Foundation of China (No. 2022JJ10065), Young Elite Scientists Sponsorship Program by CAST (No. 2022QNRC001), Continuous Support of PDL (No. WDZC20235250101).

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

# Appendix for

## Stability and Generalization of Asynchronous SGD: Sharper Bounds Beyond Lipschitz and Smoothness

## A  Background Knowledge

### A.1  ASGD Process

In the distributed parameter server architecture, the distributed workers are responsible for computing gradients, while the model updates occur on the parameter server side. Upon receiving the gradient from a worker, the server immediately utilizes it to update the model without waiting for gradient information from other workers. The ASGD procedure is described in Algorithm 1.

---

**Algorithm 1** Asynchronous SGD

---

**Initialization:** model parameter $\mathbf{w}$
**Input:** learning rate $\eta$
`// Worker` $m$
 1: **repeat**
 2:     pull the current model $\mathbf{w}$ from the server
 3:     compute gradient $\mathbf{g}^m = \nabla f(\mathbf{w}; \mathbf{z})$ with local data $\mathbf{z}$
 4:     push $\mathbf{g}^m$ to the server
 5: **until** terminated
`// Server`
 6: **if** server received gradient from any worker $m$ **then**
 7:     update the model as $\mathbf{w} \leftarrow \mathbf{w} - \eta \mathbf{g}^m$
 8:     send $\mathbf{w}$ back to worker $m$
 9: **end if**
**Output:** model $\mathbf{w}$

---

It is noteworthy that although ASGD avoids synchronization overhead, it introduces delays in model updating. To be specific, while worker $m$ is computing and uploading the gradient, the model parameter on the server side may has already been updated by another worker $m'$. In essence, the model used for gradient computation on the worker ($\mathbf{w}$ in line 3 of Algorithm 1) is inconsistent with the model updated by the server ($\mathbf{w}$ in line 7 of Algorithm 1). This characteristic renders ASGD a delayed gradient update, expressed as

$$\mathbf{w}_{k+1} = \mathbf{w}_k - \eta_k \nabla f(\mathbf{w}_{k-\tau_k}; \mathbf{z}_{i_k}),$$

where $\mathbf{w}_k, \eta_k, \tau_k,$ and $\mathbf{z}_{i_k}$ denote the model parameter, learning rate, asynchronous delay, and training sample at the $k$-th iteration, respectively. It is worth noting that the index $i_k$ is chosen uniformly at random from the set $\{1, \ldots, n\}$.

### A.2  Useful Inequalities

Our analysis frequently uses the following inequalities.

**Lemma A.1** (Young's inequality). *If $p > 1$ and $q > 1$ are real numbers such that $\frac{1}{p} + \frac{1}{q} = 1$, then for any $a, b \in \mathbb{R}^+$,*

$$ab \leq \frac{1}{p}a^p + \frac{1}{q}b^q. \tag{A.1}$$

**Lemma A.2** (Cauchy–Schwarz inequality). *For any* $\mathbf{w}, \mathbf{v} \in \mathbb{R}^d$, *the following inequality holds.*

$$\langle \mathbf{w}, \mathbf{v} \rangle \leq \|\mathbf{w}\| \cdot \|\mathbf{v}\|. \tag{A.2}$$

**Lemma A.3.** *Let* $p > 0$. *For any* $a, b \in \mathbb{R}^+$, *the following inequalities hold.*

$$2ab \leq pa^2 + \frac{1}{p}b^2, \tag{A.3}$$

$$(a + b)^2 \leq (1 + p)a^2 + (1 + \frac{1}{p})b^2. \tag{A.4}$$

In addition, we rely on the self-bounding properties of the smooth and Hölder continuous gradient functions. The proof can be found in [24, 39, 51].

**Lemma A.4.** *If the function* $\mathbf{w} \mapsto f(\mathbf{w}; \mathbf{z})$ *is non-negative and the gradient* $\nabla f$ *is* $(\alpha, \beta)$-*Hölder continuous (Definition 3) with* $\alpha \in [0, 1], \beta > 0$. *Then for any* $\mathbf{w}, \mathbf{z}$, *we have*

$$\|\nabla f(\mathbf{w}; \mathbf{z})\| \leq c_{\alpha,\beta} f^{\frac{\alpha}{1+\alpha}}(\mathbf{w}; \mathbf{z}), \tag{A.5}$$

*where the constant* $c_{\alpha,\beta}$ *is defied as*

$$c_{\alpha,\beta} := \begin{cases} (1 + 1/\alpha)^{\frac{\alpha}{1+\alpha}} \beta^{\frac{1}{1+\alpha}}, & \text{if} \quad \alpha > 0 \\ \sup_{\mathbf{z}} \|\nabla f(\mathbf{0}; \mathbf{z})\| + \beta, & \text{if} \quad \alpha = 0 \end{cases} \tag{A.6}$$

**Remark A.1.** The case $\alpha = 1$ implies that the function $\mathbf{w} \mapsto f(\mathbf{w}; \mathbf{z})$ is $\beta$-smooth (Definition 2). At this point, the constant $c_{\alpha,\beta} = \sqrt{2\beta}$, and the gradient satisfies

$$\|\nabla f(\mathbf{w}; \mathbf{z})\|^2 \leq 2\beta f(\mathbf{w}; \mathbf{z}). \tag{A.7}$$

**Lemma A.5.** *The projection operator is defined as* $\Pi_\Omega(\mathbf{v}) = \arg\min_{\mathbf{w} \in \Omega} \|\mathbf{w} - \mathbf{v}\|$, *and this operator is non-expansive, i.e.,*

$$\|\Pi_\Omega(\mathbf{w}) - \Pi_\Omega(\mathbf{v})\| \leq \|\mathbf{w} - \mathbf{v}\|, \forall \mathbf{w}, \mathbf{v} \in \mathbb{R}^d \quad and \quad \|\Pi_\Omega(\mathbf{v}) - \mathbf{w}\| \leq \|\mathbf{v} - \mathbf{w}\|, \forall \mathbf{v} \in \mathbb{R}^d, \mathbf{w} \in \Omega. \tag{A.8}$$

The proof of Lemma A.5 can be found in [56]. This non-expansive property not only ensures the plausibility of Assumption 1, but also facilitates the stability and generalization analysis of the projected ASGD algorithm (7), making it no inherently different from the standard ASGD (2).

## A.3 Proof of Lemma 1 and Lemma 2 (Section 3 in the main text)

Lemma 1 and Lemma 2 were established by Lei et al. [Theorem 2, [23]]. The following proof is derived from [Appendix B, [23]]. Recall the following definitions

$$\mathcal{S} = \{\mathbf{z}_1, \ldots, \mathbf{z}_{i-1}, \mathbf{z}_i, \mathbf{z}_{i+1}, \ldots, \mathbf{z}_n\}, \quad \mathcal{S}' = \{\mathbf{z}'_1, \ldots, \mathbf{z}'_{i-1}, \mathbf{z}'_i, \mathbf{z}'_{i+1}, \ldots, \mathbf{z}'_n\},$$

$$\mathcal{S}^{(i)} = \{\mathbf{z}_1, \ldots, \mathbf{z}_{i-1}, \mathbf{z}'_i, \mathbf{z}_{i+1}, \ldots, \mathbf{z}_n\},$$

and

$$F(\mathbf{w}) = \mathbb{E}_{\mathbf{z} \sim \mathcal{D}}[f(\mathbf{w}; \mathbf{z})], \quad F_{\mathcal{S}}(\mathbf{w}) = \frac{1}{n}\sum_{i=1}^n f(\mathbf{w}; \mathbf{z}_i), \quad \epsilon_{\text{gen}} := \mathbb{E}_{\mathcal{S}, A}[F(A(\mathcal{S})) - F_{\mathcal{S}}(A(\mathcal{S}))]. \tag{A.9}$$

Since for any $i$, the data samples $\mathbf{z}_i$ and $\mathbf{z}'_i$ are both drawn i.i.d. from $\mathcal{D}$, then $A(\mathcal{S}^{(i)})$ is independent of $\mathbf{z}_i$ and we have the following fact

$$\mathbb{E}_{\mathcal{S}}[F(A(\mathcal{S}))] = \mathbb{E}_{\mathcal{S}, \mathcal{S}'}[f(A(\mathcal{S}^{(i)}); \mathbf{z}_i)] = \frac{1}{n}\sum_{i=1}^n \mathbb{E}_{\mathcal{S}, \mathcal{S}'}[f(A(\mathcal{S}^{(i)}); \mathbf{z}_i)].$$

Hence, the generalization error satisfies

$$\epsilon_{\text{gen}} = \mathbb{E}_{\mathcal{S}, A}[F(A(\mathcal{S})) - F_{\mathcal{S}}(A(\mathcal{S}))] = \mathbb{E}_{\mathcal{S}, A}\left[\mathbb{E}_{\mathcal{S}'}[f(A(\mathcal{S}^{(i)}); \mathbf{z}_i)] - \frac{1}{n}\sum_{i=1}^n f(A(\mathcal{S}); \mathbf{z}_i)\right]$$

$$= \frac{1}{n}\sum_{i=1}^n \mathbb{E}_{\mathcal{S}, \mathcal{S}', A}\left[f(A(\mathcal{S}^{(i)}); \mathbf{z}_i) - f(A(\mathcal{S}); \mathbf{z}_i)\right]. \tag{A.10}$$

By incorporating the $\beta$-smoothness property of the function $\mathbf{w} \mapsto f(\mathbf{w}; \mathbf{z})$, we have

$$\mathbb{E}_{\mathcal{S},A}\left[F(A(\mathcal{S})) - F_{\mathcal{S}}(A(\mathcal{S}))\right] \leq \frac{1}{n}\sum_{i=1}^{n}\mathbb{E}_{\mathcal{S},\mathcal{S}',A}\left[\left\langle A(\mathcal{S}^{(i)}) - A(\mathcal{S}), \nabla f(A(\mathcal{S}); \mathbf{z}_i)\right\rangle + \frac{\beta}{2}\|A(\mathcal{S}^{(i)}) - A(\mathcal{S})\|^2\right].$$

(A.11)

Using the inequalities (A.2), (A.3) and self-bounding property (A.7), let $\gamma > 0$ then we know that

$$\left\langle A(\mathcal{S}^{(i)}) - A(\mathcal{S}), \nabla f(A(\mathcal{S}); \mathbf{z}_i)\right\rangle \leq \|A(\mathcal{S}^{(i)}) - A(\mathcal{S})\|\|\nabla f(A(\mathcal{S}); \mathbf{z}_i)\|$$

$$\leq \frac{\gamma}{2}\|A(\mathcal{S}^{(i)}) - A(\mathcal{S})\|^2 + \frac{1}{2\gamma}\|\nabla f(A(\mathcal{S}); \mathbf{z}_i)\|^2$$

$$\leq \frac{\gamma}{2}\|A(\mathcal{S}^{(i)}) - A(\mathcal{S})\|^2 + \frac{\beta}{\gamma}f(A(\mathcal{S}); \mathbf{z}_i).$$

Substituting back into inequality (A.11) yields

$$\mathbb{E}_{\mathcal{S},A}\left[F(A(\mathcal{S})) - F_{\mathcal{S}}(A(\mathcal{S}))\right] \leq \frac{\beta}{\gamma}\mathbb{E}_{\mathcal{S},A}\left[\frac{1}{n}\sum_{i=1}^{n}f(A(\mathcal{S}); \mathbf{z}_i)\right] + \frac{\beta+\gamma}{2}\mathbb{E}_{\mathcal{S},\mathcal{S}',A}\left[\frac{1}{n}\sum_{i=1}^{n}\|A(\mathcal{S}) - A(\mathcal{S}^{(i)})\|^2\right].$$

By further combining with (A.9) and Definition 1, Lemma 1 is thus derived, i.e.,

$$\mathbb{E}_{\mathcal{S},A}\left[F(A(\mathcal{S})) - F_{\mathcal{S}}(A(\mathcal{S}))\right] \leq \frac{\beta}{\gamma}\mathbb{E}_{\mathcal{S},A}[F_{\mathcal{S}}(A(\mathcal{S}))] + \frac{\beta+\gamma}{2}\epsilon_{\text{stab}}. \tag{A.12}$$

Without the smoothness assumption, we then need to utilize the convexity property of the function $\mathbf{w} \mapsto f(\mathbf{w}; \mathbf{z})$, i.e.,

$$f(A(\mathcal{S}^{(i)}); \mathbf{z}_i) - f(A(\mathcal{S}); \mathbf{z}_i) \leq \left\langle A(\mathcal{S}^{(i)}) - A(\mathcal{S}), \nabla f(A(\mathcal{S}^{(i)}); \mathbf{z}_i)\right\rangle$$

$$\leq \frac{\gamma}{2}\|A(\mathcal{S}^{(i)}) - A(\mathcal{S})\|^2 + \frac{1}{2\gamma}\|\nabla f(A(\mathcal{S}^{(i)}); \mathbf{z}_i)\|^2$$

$$\leq \frac{\gamma}{2}\|A(\mathcal{S}^{(i)}) - A(\mathcal{S})\|^2 + \frac{c_{\alpha,\beta}^2}{2\gamma}f^{\frac{2\alpha}{1+\alpha}}(A(\mathcal{S}^{(i)}); \mathbf{z}_i),$$

where the last two inequalities use (A.2), (A.3) and the self-bounding property (A.5). Substituting it back into inequality (A.10) leads to Lemma 2, i.e.,

$$\mathbb{E}_{\mathcal{S},A}\left[F(A(\mathcal{S})) - F_{\mathcal{S}}(A(\mathcal{S}))\right]$$

$$\leq \frac{c_{\alpha,\beta}^2}{2\gamma}\frac{1}{n}\sum_{i=1}^{n}\mathbb{E}_{\mathcal{S},\mathcal{S}',A}\left[f^{\frac{2\alpha}{1+\alpha}}(A(\mathcal{S}^{(i)}); \mathbf{z}_i)\right] + \frac{\gamma}{2}\mathbb{E}_{\mathcal{S},\mathcal{S}',A}\left[\frac{1}{n}\sum_{i=1}^{n}\|A(\mathcal{S}) - A(\mathcal{S}^{(i)})\|^2\right] \tag{A.13}$$

$$\leq \frac{c_{\alpha,\beta}^2}{2\gamma}\mathbb{E}_{\mathcal{S},A}[F^{\frac{2\alpha}{1+\alpha}}(A(\mathcal{S}))] + \frac{\gamma}{2}\epsilon_{\text{stab}}.$$

Here we use the concavity of the map $x \mapsto x^{\frac{2\alpha}{1+\alpha}}$ and the following fact

$$\mathbb{E}_{\mathcal{S},\mathcal{S}',A}\left[f^{\frac{2\alpha}{1+\alpha}}(A(\mathcal{S}^{(i)}); \mathbf{z}_i)\right] \leq \mathbb{E}_{\mathcal{S},\mathcal{S}',A}\left[\left(\mathbb{E}_{\mathbf{z}_i}\left[f(A(\mathcal{S}^{(i)}); \mathbf{z}_i)\right]\right)^{\frac{2\alpha}{1+\alpha}}\right]$$

$$= \mathbb{E}_{\mathcal{S},\mathcal{S}',A}\left[F^{\frac{2\alpha}{1+\alpha}}(A(\mathcal{S}^{(i)}))\right] = \mathbb{E}_{\mathcal{S},A}\left[F^{\frac{2\alpha}{1+\alpha}}(A(\mathcal{S}))\right].$$

# B  Proof of Stability and Generalization Bounds (Section 4 in the main text)

## B.1  Proof of Lemma 3 (approximately non-expansive property of delayed gradient updates)

Due to that the function $\mathbf{w} \mapsto f(\mathbf{w}; \mathbf{z})$ is convex and $\beta$-smooth, the gradient $\nabla f$ is co-coercive, namely

$$\left\langle \mathbf{w}_{k-\tau_k} - \mathbf{w}_{k-\tau_k}^{(i)}, \nabla f(\mathbf{w}_{k-\tau_k}; \mathbf{z}_{i_k}) - \nabla f(\mathbf{w}_{k-\tau_k}^{(i)}; \mathbf{z}_{i_k})\right\rangle \geq \frac{1}{\beta}\|\nabla f(\mathbf{w}_{k-\tau_k}; \mathbf{z}_{i_k}) - \nabla f(\mathbf{w}_{k-\tau_k}^{(i)}; \mathbf{z}_{i_k})\|^2.$$

Using this co-coercivity with learning rate $\eta_k \le 2/\beta$, we have

$$\left\| \mathbf{w}_k - \eta_k \nabla f(\mathbf{w}_{k-\tau_k}; \mathbf{z}_{i_k}) - \left( \mathbf{w}_k^{(i)} - \eta_k \nabla f(\mathbf{w}_{k-\tau_k}^{(i)}; \mathbf{z}_{i_k}) \right) \right\|^2$$

$$= \|\mathbf{w}_k - \mathbf{w}_k^{(i)}\|^2 + \eta_k^2 \|\nabla f(\mathbf{w}_{k-\tau_k}; \mathbf{z}_{i_k}) - \nabla f(\mathbf{w}_{k-\tau_k}^{(i)}; \mathbf{z}_{i_k})\|^2$$

$$\quad - 2\eta_k \langle \mathbf{w}_k - \mathbf{w}_k^{(i)}, \nabla f(\mathbf{w}_{k-\tau_k}; \mathbf{z}_{i_k}) - \nabla f(\mathbf{w}_{k-\tau_k}^{(i)}; \mathbf{z}_{i_k}) \rangle$$

$$= \|\mathbf{w}_k - \mathbf{w}_k^{(i)}\|^2 + \eta_k^2 \|\nabla f(\mathbf{w}_{k-\tau_k}; \mathbf{z}_{i_k}) - \nabla f(\mathbf{w}_{k-\tau_k}^{(i)}; \mathbf{z}_{i_k})\|^2$$

$$\quad - 2\eta_k \langle \mathbf{w}_{k-\tau_k} - \mathbf{w}_{k-\tau_k}^{(i)}, \nabla f(\mathbf{w}_{k-\tau_k}; \mathbf{z}_{i_k}) - \nabla f(\mathbf{w}_{k-\tau_k}^{(i)}; \mathbf{z}_{i_k}) \rangle$$

$$\quad - 2\eta_k \langle \mathbf{w}_k - \mathbf{w}_{k-\tau_k} - (\mathbf{w}_k^{(i)} - \mathbf{w}_{k-\tau_k}^{(i)}), \nabla f(\mathbf{w}_{k-\tau_k}; \mathbf{z}_{i_k}) - \nabla f(\mathbf{w}_{k-\tau_k}^{(i)}; \mathbf{z}_{i_k}) \rangle$$

$$\le \|\mathbf{w}_k - \mathbf{w}_k^{(i)}\|^2 - 2\eta_k \langle \mathbf{w}_k - \mathbf{w}_{k-\tau_k} - (\mathbf{w}_k^{(i)} - \mathbf{w}_{k-\tau_k}^{(i)}), \nabla f(\mathbf{w}_{k-\tau_k}; \mathbf{z}_{i_k}) - \nabla f(\mathbf{w}_{k-\tau_k}^{(i)}; \mathbf{z}_{i_k}) \rangle.$$

From the iterative scheme of ASGD (7), we know that

$$\langle \mathbf{w}_k - \mathbf{w}_{k-\tau_k} - (\mathbf{w}_k^{(i)} - \mathbf{w}_{k-\tau_k}^{(i)}), \nabla f(\mathbf{w}_{k-\tau_k}; \mathbf{z}_{i_k}) - \nabla f(\mathbf{w}_{k-\tau_k}^{(i)}; \mathbf{z}_{i_k}) \rangle$$

$$= \sum_{j=1}^{\tau_k} \langle \mathbf{w}_{k-j+1} - \mathbf{w}_{k-j} - (\mathbf{w}_{k-j+1}^{(i)} - \mathbf{w}_{k-j}^{(i)}), \nabla f(\mathbf{w}_{k-\tau_k}; \mathbf{z}_{i_k}) - \nabla f(\mathbf{w}_{k-\tau_k}^{(i)}; \mathbf{z}_{i_k}) \rangle$$

$$\le \sum_{j=1}^{\tau_k} \eta_{k-j} \|\nabla f(\mathbf{w}_{k-j-\tau_{k-j}}; \mathbf{z}_{i_{k-j}}) - \nabla f(\mathbf{w}_{k-j-\tau_{k-j}}^{(i)}; \mathbf{z}_{i_{k-j}})\| \|\nabla f(\mathbf{w}_{k-\tau_k}; \mathbf{z}_{i_k}) - \nabla f(\mathbf{w}_{k-\tau_k}^{(i)}; \mathbf{z}_{i_k})\|,$$

$$\tag{B.1}$$

where the last inequality is due to (A.2) and (A.8). Following the $\beta$-smooth property and Assumption 1, we can derive

$$\|\nabla f(\mathbf{w}_{s-\tau_s}; \mathbf{z}_{i_s}) - \nabla f(\mathbf{w}_{s-\tau_s}^{(i)}; \mathbf{z}_{i_s})\| \le \beta \|\mathbf{w}_{s-\tau_s} - \mathbf{w}_{s-\tau_s}^{(i)}\| \le \beta r, \text{ for } s = k, k-j;\ j = 1, \ldots, \tau_k. \tag{B.2}$$

With inequalities (B.1) and (B.2), we are arrive at

$$\left\| \mathbf{w}_k - \eta_k \nabla f(\mathbf{w}_{k-\tau_k}; \mathbf{z}_{i_k}) - \left( \mathbf{w}_k^{(i)} - \eta_k \nabla f(\mathbf{w}_{k-\tau_k}^{(i)}; \mathbf{z}_{i_k}) \right) \right\|^2 \le \|\mathbf{w}_k - \mathbf{w}_k^{(i)}\|^2 + 2\eta_k \beta^2 r^2 \sum_{j=1}^{\tau_k} \eta_{k-j}. \tag{B.3}$$

## B.2 Proof of Theorem 1 (algorithm stability under the smooth assumption)

Let $\mathbf{w}_k$ and $\mathbf{w}_k^{(i)}$ denote the models produced by ASGD (7) after $k$ iterations on the datasets $\mathcal{S}$ and $\mathcal{S}^{(i)}$, respectively. Given that the index $i_k$ at the $k$-th iteration is chosen randomly from the set $\{1, 2, \ldots, n\}$, there is a probability of $1 - 1/n$ that $i_k \ne i$. Then, by the approximately non-expansive property (B.3) of the ASGD iteration and (A.8), we have

$$\|\mathbf{w}_{k+1} - \mathbf{w}_{k+1}^{(i)}\|^2 \le \left\| \mathbf{w}_k - \eta_k \nabla f(\mathbf{w}_{k-\tau_k}; \mathbf{z}_{i_k}) - \left( \mathbf{w}_k^{(i)} - \eta_k \nabla f(\mathbf{w}_{k-\tau_k}^{(i)}; \mathbf{z}_{i_k}) \right) \right\|^2$$

$$\le \|\mathbf{w}_k - \mathbf{w}_k^{(i)}\|^2 + 2\eta_k \beta^2 r^2 \sum_{j=1}^{\tau_k} \eta_{k-j}.$$

On the other hand, there is a probability of $1/n$ such that the algorithm accurately selects the $i$-th sample point ($i_k = i$) that is different in the two datasets $\mathcal{S}$ and $\mathcal{S}^{(i)}$. In this case, we can perform the following analysis based on the inequality (A.4) with $p > 0$, self-bounding property (A.7) and non-expansive projection (A.8)

$$\|\mathbf{w}_{k+1} - \mathbf{w}_{k+1}^{(i)}\|^2 \le \|\mathbf{w}_k - \eta_k \nabla f(\mathbf{w}_{k-\tau_k}; \mathbf{z}_i) - \mathbf{w}_k^{(i)} + \eta_k \nabla f(\mathbf{w}_{k-\tau_k}^{(i)}; \mathbf{z}_i')\|^2$$

$$\le (1+p)\|\mathbf{w}_k - \mathbf{w}_k^{(i)}\|^2 + (1+1/p)\eta_k^2 \|\nabla f(\mathbf{w}_{k-\tau_k}; \mathbf{z}_i) - \nabla f(\mathbf{w}_{k-\tau_k}^{(i)}; \mathbf{z}_i')\|^2$$

$$\le (1+p)\|\mathbf{w}_k - \mathbf{w}_k^{(i)}\|^2 + 2(1+1/p)\eta_k^2 \left[ \|\nabla f(\mathbf{w}_{k-\tau_k}; \mathbf{z}_i)\|^2 + \|\nabla f(\mathbf{w}_{k-\tau_k}^{(i)}; \mathbf{z}_i')\|^2 \right]$$

$$\le (1+p)\|\mathbf{w}_k - \mathbf{w}_k^{(i)}\|^2 + 4\beta(1+1/p)\eta_k^2 \left[ f(\mathbf{w}_{k-\tau_k}; \mathbf{z}_i) + f(\mathbf{w}_{k-\tau_k}^{(i)}; \mathbf{z}_i') \right].$$

Combining the two cases above and taking the expectation with respect to the randomness of the algorithm yields

$$\mathbb{E}_A \|\mathbf{w}_{k+1} - \mathbf{w}_{k+1}^{(i)}\|^2 \leq (1 + \frac{p}{n})\mathbb{E}_A\|\mathbf{w}_k - \mathbf{w}_k^{(i)}\|^2 + \frac{2\eta_k \beta^2 r^2(n-1)}{n}\sum_{j=1}^{\tau_k}\eta_{k-j}$$
$$+ \frac{4\beta(1+1/p)\eta_k^2}{n}\mathbb{E}_A\left[f(\mathbf{w}_{k-\tau_k};\mathbf{z}_i) + f(\mathbf{w}_{k-\tau_k}^{(i)};\mathbf{z}_i')\right].$$

Since $\mathbf{z}_i$ and $\mathbf{z}_i'$ are i.i.d. sampled from the same distribution $\mathcal{D}$, we have the following fact

$$\mathbb{E}_{\mathcal{S},\mathcal{S}',A}[f(\mathbf{w}_{k-\tau_k}^{(i)};\mathbf{z}_i')] = \mathbb{E}_{\mathcal{S},A}[f(\mathbf{w}_{k-\tau_k};\mathbf{z}_i)].$$

A subsequent expectation over the randomness of data produces

$$\mathbb{E}_{\mathcal{S},\mathcal{S}',A}\|\mathbf{w}_{k+1} - \mathbf{w}_{k+1}^{(i)}\|^2$$
$$\leq (1 + \frac{p}{n})\mathbb{E}_{\mathcal{S},\mathcal{S}',A}\|\mathbf{w}_k - \mathbf{w}_k^{(i)}\|^2 + \frac{8\beta(1+1/p)\eta_k^2}{n}\mathbb{E}_{\mathcal{S},A}\left[f(\mathbf{w}_{k-\tau_k};\mathbf{z}_i)\right] + 2\eta_k\beta^2 r^2\sum_{j=1}^{\tau_k}\eta_{k-j}$$
$$\leq \sum_{l=1}^{k}(1 + \frac{p}{n})^{(k-l)}\left[\frac{8\beta(1+1/p)\eta_l^2}{n}\mathbb{E}_{\mathcal{S},A}\left[f(\mathbf{w}_{l-\tau_l};\mathbf{z}_i)\right] + 2\eta_l\beta^2 r^2\sum_{j=1}^{\tau_l}\eta_{l-j}\right],$$

where the second inequality is due to the same initialization $\mathbf{w}_1 = \mathbf{w}_1^{(i)}$. Let $p = n/k$, then $(1 + p/n)^{(k-1)} \leq \mathrm{e}$ (where $\mathrm{e}$ is the natural constant), and the on-average model stability of ASGD satisfies

$$\mathbb{E}_{\mathcal{S},\mathcal{S}',A}\left[\frac{1}{n}\sum_{i=1}^{n}\|\mathbf{w}_{k+1} - \mathbf{w}_{k+1}^{(i)}\|^2\right]$$
$$\leq \sum_{l=1}^{k}(1 + \frac{p}{n})^{(k-l)}\left[\frac{8\beta(1+1/p)\eta_l^2}{n}\mathbb{E}_{\mathcal{S},A}\left[F_{\mathcal{S}}(\mathbf{w}_{l-\tau_l})\right] + 2\eta_l\beta^2 r^2\sum_{j=1}^{\tau_l}\eta_{l-j}\right]$$
$$\leq (1 + \frac{p}{n})^{(k-1)}\left[\frac{8\beta(1+1/p)}{n}\sum_{l=1}^{k}\eta_l^2\mathbb{E}_{\mathcal{S},A}F_{\mathcal{S}}(\mathbf{w}_{l-\tau_l}) + 2\beta^2 r^2\sum_{l=1}^{k}\eta_l\sum_{j=1}^{\tau_l}\eta_{l-j}\right] \tag{B.4}$$
$$\leq \frac{8\beta\mathrm{e}(1+k/n)}{n}\sum_{l=1}^{k}\eta_l^2\mathbb{E}_{\mathcal{S},A}\left[F_{\mathcal{S}}(\mathbf{w}_{l-\tau_l})\right] + 2\beta^2 r^2\mathrm{e}\sum_{l=1}^{k}\eta_l\sum_{j=1}^{\tau_l}\eta_{l-j}.$$

For the further investigation of the algorithm stability of ASGD, it is imperative to bound error $\sum_{l=1}^{k}\eta_l^2\mathbb{E}_{\mathcal{S},A}[F_{\mathcal{S}}(\mathbf{w}_{l-\tau_l})]$ of the delayed model. With the ASGD update (7), inequality (A.8), convexity and smooth property (A.7), we have the following derivation

$$\|\mathbf{w}_{k+1} - \mathbf{w}^*\|^2 \leq \|\mathbf{w}_k - \eta_k\nabla f(\mathbf{w}_{k-\tau_k};\mathbf{z}_{i_k}) - \mathbf{w}^*\|^2$$
$$= \|\mathbf{w}_k - \mathbf{w}^*\|^2 + \eta_k^2\|\nabla f(\mathbf{w}_{k-\tau_k};\mathbf{z}_{i_k})\|^2 + 2\eta_k\langle \mathbf{w}^* - \mathbf{w}_k, \nabla f(\mathbf{w}_{k-\tau_k};\mathbf{z}_{i_k})\rangle$$
$$\leq \|\mathbf{w}_k - \mathbf{w}^*\|^2 + 2\beta\eta_k^2 f(\mathbf{w}_{k-\tau_k};\mathbf{z}_{i_k}) + 2\eta_k\langle \mathbf{w}^* - \mathbf{w}_{k-\tau_k}, \nabla f(\mathbf{w}_{k-\tau_k};\mathbf{z}_{i_k})\rangle$$
$$+ 2\eta_k\langle \mathbf{w}_{k-\tau_k} - \mathbf{w}_k, \nabla f(\mathbf{w}_{k-\tau_k};\mathbf{z}_{i_k})\rangle$$
$$\leq \|\mathbf{w}_k - \mathbf{w}^*\|^2 + 2\beta\eta_k^2 f(\mathbf{w}_{k-\tau_k};\mathbf{z}_{i_k}) + 2\eta_k(f(\mathbf{w}^*;\mathbf{z}_{i_k}) - f(\mathbf{w}_{k-\tau_k};\mathbf{z}_{i_k}))$$
$$+ 2\eta_k\langle \mathbf{w}_{k-\tau_k} - \mathbf{w}_k, \nabla f(\mathbf{w}_{k-\tau_k};\mathbf{z}_{i_k})\rangle$$
$$\leq \|\mathbf{w}_k - \mathbf{w}^*\|^2 - \eta_k f(\mathbf{w}_{k-\tau_k};\mathbf{z}_{i_k}) + 2\eta_k f(\mathbf{w}^*;\mathbf{z}_{i_k}) + 2\eta_k\langle \mathbf{w}_{k-\tau_k} - \mathbf{w}_k, \nabla f(\mathbf{w}_{k-\tau_k};\mathbf{z}_{i_k})\rangle,$$

where $\mathbf{w}^* \in \mathrm{argmin}_{\mathbf{w}\in\Omega} F(\mathbf{w})$, and the last inequality is due to $\eta_k \leq 1/2\beta$. Then

$$\eta_k f(\mathbf{w}_{k-\tau_k};\mathbf{z}_{i_k}) \leq \|\mathbf{w}_k - \mathbf{w}^*\|^2 - \|\mathbf{w}_{k+1} - \mathbf{w}^*\|^2 + 2\eta_k f(\mathbf{w}^*;\mathbf{z}_{i_k})$$
$$+ 2\eta_k\langle \mathbf{w}_{k-\tau_k} - \mathbf{w}_k, \nabla f(\mathbf{w}_{k-\tau_k};\mathbf{z}_{i_k})\rangle. \tag{B.5}$$

Following the inequality (A.2), self-bounding property (A.7), and Assumption 1, we know that

$$2\eta_k\langle \mathbf{w}_{k-\tau_k}-\mathbf{w}_k, \nabla f(\mathbf{w}_{k-\tau_k};\mathbf{z}_{i_k})\rangle \le 2r\eta_k\|\nabla f(\mathbf{w}_{k-\tau_k};\mathbf{z}_{i_k})\| \le 2r\eta_k\sqrt{2\beta f(\mathbf{w}_{k-\tau_k};\mathbf{z}_{i_k})}$$
$$\le 2r\sqrt{2\beta\eta_k}\cdot\sqrt{\eta_k f(\mathbf{w}_{k-\tau_k};\mathbf{z}_{i_k})} \le 4\beta r^2\eta_k + \frac{\eta_k}{2}f(\mathbf{w}_{k-\tau_k};\mathbf{z}_{i_k}),$$

where last inequality uses (A.3) with $p=1$. Turning to (B.5), we have

$$\eta_k f(\mathbf{w}_{k-\tau_k};\mathbf{z}_{i_k}) \le 2\|\mathbf{w}_k-\mathbf{w}^*\|^2 - 2\|\mathbf{w}_{k+1}-\mathbf{w}^*\|^2 + 4\eta_k f(\mathbf{w}^*;\mathbf{z}_{i_k}) + 8\beta r^2\eta_k.$$

Multiplying both sides with the non-increasing learning rate, we get

$$\eta_k^2 f(\mathbf{w}_{k-\tau_k};\mathbf{z}_{i_k}) \le 2\eta_k\|\mathbf{w}_k-\mathbf{w}^*\|^2 - 2\eta_{k+1}\|\mathbf{w}_{k+1}-\mathbf{w}^*\|^2 + 4\eta_k^2 f(\mathbf{w}^*;\mathbf{z}_{i_k}) + 8\beta r^2\eta_k^2.$$

Taking an expectation on both sides followed by a summation leads to

$$\sum_{l=1}^{k}\eta_l^2\mathbb{E}_{\mathcal{S},A}[F_{\mathcal{S}}(\mathbf{w}_{l-\tau_l})] \le 2\eta_1\|\mathbf{w}_1-\mathbf{w}^*\|^2 + 4\sum_{l=1}^{k}\eta_l^2 F(\mathbf{w}^*) + 8\beta r^2\sum_{l=1}^{k}\eta_l^2, \qquad \text{(B.6)}$$

where we use $\mathbb{E}_{\mathcal{S}}[f(\mathbf{w}^*;\mathbf{z}_{i_k})] = \mathbb{E}_{\mathcal{S}}[F_{\mathcal{S}}(\mathbf{w}^*)] = F(\mathbf{w}^*)$. Substituting (B.6) into (B.4), we are arrive at

$$\mathbb{E}_{\mathcal{S},\mathcal{S}',A}\left[\frac{1}{n}\sum_{i=1}^{n}\|\mathbf{w}_{k+1}-\mathbf{w}_{k+1}^{(i)}\|^2\right]$$
$$\le \frac{16\beta e(1+k/n)}{n}\left[\eta_1\|\mathbf{w}_1-\mathbf{w}^*\|^2 + 2\sum_{l=1}^{k}\eta_l^2 F(\mathbf{w}^*) + 4\beta r^2\sum_{l=1}^{k}\eta_l^2\right] + 2\beta^2 r^2 e\sum_{l=1}^{k}\eta_l\sum_{j=1}^{\tau_l}\eta_{l-j}.$$
$$\text{(B.7)}$$

## B.3 Proof of Theorem 2 (generalization error under the smooth assumption)

Together with Lemma 1 (A.12) and Theorem 1 (B.7), we have

$$\mathbb{E}_{\mathcal{S},A}[F(\mathbf{w}_{k+1})-F_{\mathcal{S}}(\mathbf{w}_{k+1})] \le \frac{\beta}{\gamma}\mathbb{E}_{\mathcal{S},A}[F_{\mathcal{S}}(\mathbf{w}_{k+1})] + \frac{\beta+\gamma}{2}\mathbb{E}_{\mathcal{S},\mathcal{S}',A}\left[\frac{1}{n}\sum_{i=1}^{n}\|\mathbf{w}_{k+1}-\mathbf{w}_{k+1}^{(i)}\|^2\right]$$
$$\le \frac{\beta}{\gamma}\mathbb{E}_{\mathcal{S},A}[F_{\mathcal{S}}(\mathbf{w}_{k+1})] + \frac{8\beta e(\beta+\gamma)(1+k/n)}{n}\left[\eta_1\|\mathbf{w}_1-\mathbf{w}^*\|^2 + 2\sum_{l=1}^{k}\eta_l^2 F(\mathbf{w}^*) + 4\beta r^2\sum_{l=1}^{k}\eta_l^2\right]$$
$$+ \beta^2 r^2(\beta+\gamma)e\sum_{l=1}^{k}\eta_l\sum_{j=1}^{\tau_l}\eta_{l-j}.$$
$$\text{(B.8)}$$

Let $\gamma=1$, then the generalization error $\epsilon_{\text{gen}}$ of ASGD satisfies

$$\mathbb{E}_{\mathcal{S},A}[F(\mathbf{w}_K)-F_{\mathcal{S}}(\mathbf{w}_K)]$$
$$= \mathcal{O}\left(\mathbb{E}_{\mathcal{S},A}[F_{\mathcal{S}}(\mathbf{w}_K)] + \sum_{k=1}^{K}\eta_k\sum_{j=1}^{\tau_k}\eta_{k-j} + \frac{1+K/n}{n}\left[\eta_1\|\mathbf{w}_1-\mathbf{w}^*\|^2 + (1+F(\mathbf{w}^*))\sum_{k=1}^{K}\eta_k^2\right]\right).$$
$$\text{(B.9)}$$

## B.4 Proof of Corollary 1 (generalization error with specific learning rates)

Let $K \asymp n$. Following Theorem 2 (B.9), if we use the delay-independent constant learning rate $\eta_k \equiv \eta$, the generalization error $\epsilon_{\text{gen}}$ satisfies

$$\mathbb{E}_{\mathcal{S},A}[F(\mathbf{w}_K)-F_{\mathcal{S}}(\mathbf{w}_K)] = \mathcal{O}\left(\sum_{k=1}^{K}\tau_k + \frac{\|\mathbf{w}_1-\mathbf{w}^*\|^2}{n} + \mathbb{E}_{\mathcal{S},A}[F_{\mathcal{S}}(\mathbf{w}_K)+F(\mathbf{w}^*)]\right). \quad \text{(B.10)}$$

In comparison to SGD [23], asynchronous training with a fixed learning rate $\eta$ introduces an additional error $\mathcal{O}(\sum_{k=1}^{K} \tau_k)$ due to delay. This error accumulates as iterations increase, which subsequently deteriorates the generalization performance of ASGD. However, the experimental results in Appendix D demonstrate that the generalization error bound in (B.10) is pessimistic.

In the low-noise case, i.e., $F(\mathbf{w}^*) = 0$, Stich and Karimireddy [40] proved that $F_{\mathcal{S}}(\mathbf{w}_K) = \mathcal{O}(1/\sqrt{K})$ under the smooth and general quasi-convex loss function conditions with the learning rate $\eta_k = c(\overline{\tau}\sqrt{K})^{-1}$ ($c > 0$ is a constant, $\overline{\tau} = \sum_{k=1}^{K} \tau_k/K$). At this point, the generalization error of ASGD is

$$\mathbb{E}_{\mathcal{S},A}\left[F(\mathbf{w}_K) - F_{\mathcal{S}}(\mathbf{w}_K)\right] = \mathcal{O}\left(\frac{1}{\sqrt{K}}\right) + \mathcal{O}\left(\frac{1 + K/n}{n}\left[\frac{c\|\mathbf{w}_1 - \mathbf{w}^*\|^2}{\overline{\tau}\sqrt{K}} + \frac{c^2}{\overline{\tau}^2}\right] + \frac{c^2}{K\overline{\tau}^2}\sum_{k=1}^{K}\tau_k\right)$$

$$= \mathcal{O}\left(\frac{1}{\overline{\tau}} + \frac{1}{\sqrt{K}}\right).$$

**Remark B.1.** Under the assumptions of $\beta$-smoothness and $(M, \sigma^2)$-bounded noise, Stich and Karimireddy [40] proved that $F_{\mathcal{S}}(\mathbf{w}_K) = \mathcal{O}(1/\sqrt{K})$ when the learning rate is chosen as $\eta_k \leq \frac{1}{10\beta(\overline{\tau}+M)}$. In the above proof, we can flexibly adjust the constant $c$ to make the learning rate satisfy the requirements of the study [40], enabling the safe utilization of its optimization results.

**Remark B.2.** The notation $\mathcal{O}$ hides the numerical values and specific fixed constants, such as $c$, e, $\beta$, and $r$. This notation facilitates the reader's comprehension by allowing for an intuitive understanding of the effects of important variables such as asynchronous delay $\overline{\tau}$, the number of iterations $K$, and the amount of training data $n$ on stability and generalization.

## B.5 Proof of Lemma 4 (optimization error under the smooth assumption)

By the ASGD update (7), convexity, smooth property (A.7) and the non-expansive projection (A.8), we can derive

$$\begin{aligned}
\|\mathbf{w}_{k+1} - \mathbf{w}^*\|^2 &\leq \|\mathbf{w}_k - \eta_k\nabla f(\mathbf{w}_{k-\tau_k}; \mathbf{z}_{i_k}) - \mathbf{w}^*\|^2 \\
&= \|\mathbf{w}_k - \mathbf{w}^*\|^2 + \eta_k^2\|\nabla f(\mathbf{w}_{k-\tau_k}; \mathbf{z}_{i_k})\|^2 + 2\eta_k\langle\mathbf{w}^* - \mathbf{w}_k, \nabla f(\mathbf{w}_{k-\tau_k}; \mathbf{z}_{i_k})\rangle \\
&\leq \|\mathbf{w}_k - \mathbf{w}^*\|^2 + 2\beta\eta_k^2 f(\mathbf{w}_{k-\tau_k}; \mathbf{z}_{i_k}) + 2\eta_k\langle\mathbf{w}^* - \mathbf{w}_k, \nabla f(\mathbf{w}_k; \mathbf{z}_{i_k})\rangle \\
&\quad + 2\eta_k\langle\mathbf{w}^* - \mathbf{w}_k, \nabla f(\mathbf{w}_{k-\tau_k}; \mathbf{z}_{i_k}) - \nabla f(\mathbf{w}_k; \mathbf{z}_{i_k})\rangle \\
&\leq \|\mathbf{w}_k - \mathbf{w}^*\|^2 + 2\beta\eta_k^2 f(\mathbf{w}_{k-\tau_k}; \mathbf{z}_{i_k}) + 2\eta_k(f(\mathbf{w}^*; \mathbf{z}_{i_k}) - f(\mathbf{w}_k; \mathbf{z}_{i_k})) \\
&\quad + 2\beta\eta_k\|\mathbf{w}_k - \mathbf{w}^*\| \cdot \|\mathbf{w}_k - \mathbf{w}_{k-\tau_k}\|.
\end{aligned} \tag{B.11}$$

From the iterative scheme of ASGD (7) and (A.8), we know that

$$\|\mathbf{w}_k - \mathbf{w}_{k-\tau_k}\| \leq \sum_{j=1}^{\tau_k}\|\mathbf{w}_{k-j+1} - \mathbf{w}_{k-j}\| \leq \sum_{j=1}^{\tau_k}\eta_{k-j}\|\nabla f(\mathbf{w}_{k-j-\tau_{k-j}}; \mathbf{z}_{i_{k-j}})\|. \tag{B.12}$$

Then taking an expectation on both sides of (B.11) and combing with Assumptions 1 and 2, we have

$$2\eta_k\mathbb{E}_{\mathcal{S},A}[F_{\mathcal{S}}(\mathbf{w}_k) - F_{\mathcal{S}}(\mathbf{w}^*)] \leq \mathbb{E}_{\mathcal{S},A}\|\mathbf{w}_k - \mathbf{w}^*\|^2 - \mathbb{E}_{\mathcal{S},A}\|\mathbf{w}_{k+1} - \mathbf{w}^*\|^2 + 2\beta\eta_k^2\mathbb{E}_{\mathcal{S},A}[F_{\mathcal{S}}(\mathbf{w}_{k-\tau_k})]$$

$$+ 2\beta Lr\eta_k\sum_{j=1}^{\tau_k}\eta_{k-j}.$$

Subsequently a summation of the inequality produces

$$2\sum_{k=1}^{K}\eta_k\mathbb{E}_{\mathcal{S},A}[F_{\mathcal{S}}(\mathbf{w}_k) - F_{\mathcal{S}}(\mathbf{w}^*)] \leq \|\mathbf{w}_1 - \mathbf{w}^*\|^2 + 2\beta\sum_{k=1}^{K}\eta_k^2\mathbb{E}_{\mathcal{S},A}[F_{\mathcal{S}}(\mathbf{w}_{k-\tau_k})] + 2\beta Lr\sum_{k=1}^{K}\eta_k\sum_{j=1}^{\tau_k}\eta_{k-j}.$$

Leveraging the optimization bound (B.6) with $\eta_k \leq 1/2\beta$, we can derive

$$\sum_{k=1}^{K}\eta_k\mathbb{E}_{\mathcal{S},A}[F_{\mathcal{S}}(\mathbf{w}_k) - F_{\mathcal{S}}(\mathbf{w}^*)] \leq \frac{1}{2}\|\mathbf{w}_1 - \mathbf{w}^*\|^2 + \beta\sum_{k=1}^{K}\eta_k^2\mathbb{E}_{\mathcal{S},A}[F_{\mathcal{S}}(\mathbf{w}_{k-\tau_k})] + \beta Lr\sum_{k=1}^{K}\eta_k\sum_{j=1}^{\tau_k}\eta_{k-j}$$

$$\leq (\frac{1}{2} + 2\beta\eta_1)\|\mathbf{w}_1 - \mathbf{w}^*\|^2 + 4\beta\sum_{k=1}^{K}\eta_k^2 F(\mathbf{w}^*) + 8\beta^2 r^2\sum_{k=1}^{K}\eta_k^2 + \beta Lr\sum_{k=1}^{K}\eta_k\sum_{j=1}^{\tau_k}\eta_{k-j}.$$

$$\tag{B.13}$$

Let the average model

$$\overline{\mathbf{w}}_K := \frac{\sum_{k=1}^{K} \eta_k \mathbf{w}_k}{\sum_{k=1}^{K} \eta_k} \in \Omega.$$

Following the convexity of the function $F_{\mathcal{S}}$, we know that

$$\mathbb{E}_{\mathcal{S},A}[F_{\mathcal{S}}(\overline{\mathbf{w}}_K) - F_{\mathcal{S}}(\mathbf{w}^*)] \leq \frac{\sum_{k=1}^{K} \eta_k \mathbb{E}_{\mathcal{S},A}[F_{\mathcal{S}}(\mathbf{w}_k) - F_{\mathcal{S}}(\mathbf{w}^*)]}{\sum_{k=1}^{K} \eta_k}$$

$$\leq (\frac{1}{2} + 2\beta\eta_1)\frac{\|\mathbf{w}_1 - \mathbf{w}^*\|^2}{\sum_{k=1}^{K} \eta_k} + 4\beta(F(\mathbf{w}^*) + 2\beta r^2)\frac{\sum_{k=1}^{K} \eta_k^2}{\sum_{k=1}^{K} \eta_k} + \beta L r \frac{\sum_{k=1}^{K} \eta_k \sum_{j=1}^{\tau_k} \eta_{k-j}}{\sum_{k=1}^{K} \eta_k}.$$

## B.6 Proof of Theorem 3 (excess generalization error under the smooth assumption)

Multiplying both sides of the generalization error (B.8) by $\eta_{k+1}$ followed with a summation gives

$$\sum_{k=1}^{K} \eta_k \mathbb{E}_{\mathcal{S},A}[F(\mathbf{w}_k)] \leq (1 + \frac{\beta}{\gamma}) \sum_{k=1}^{K} \eta_k \mathbb{E}_{\mathcal{S},A}[F_{\mathcal{S}}(\mathbf{w}_k)] + \beta^2 r^2 (\beta + \gamma) e \sum_{k=1}^{K} \eta_k \sum_{l=1}^{k} \eta_l \sum_{j=1}^{\tau_l} \eta_{l-j}$$

$$+ \frac{8\beta e(\beta + \gamma)}{n} \sum_{k=1}^{K} \eta_k (1 + \frac{k}{n}) \left[ \eta_1 \|\mathbf{w}_1 - \mathbf{w}^*\|^2 + 2 \sum_{l=1}^{k} \eta_l^2 F(\mathbf{w}^*) + 4\beta r^2 \sum_{l=1}^{k} \eta_l^2 \right].$$
(B.14)

Substituting the optimization error (B.13) into the above inequality (B.14), we have

$$\sum_{k=1}^{K} \eta_k \mathbb{E}_{\mathcal{S},A}[F(\mathbf{w}_k) - F(\mathbf{w}^*)] \leq \beta^2 r^2 (\beta + \gamma) e \sum_{k=1}^{K} \eta_k \sum_{l=1}^{k} \eta_l \sum_{j=1}^{\tau_l} \eta_{l-j} + \frac{\beta}{\gamma} \sum_{k=1}^{K} \eta_k F(\mathbf{w}^*)$$

$$+ (1 + \frac{\beta}{\gamma}) \left[ (\frac{1}{2} + 2\beta\eta_1)\|\mathbf{w}_1 - \mathbf{w}^*\|^2 + 4\beta \sum_{k=1}^{K} \eta_k^2 F(\mathbf{w}^*) + 8\beta^2 r^2 \sum_{k=1}^{K} \eta_k^2 + \beta L r \sum_{k=1}^{K} \eta_k \sum_{j=1}^{\tau_k} \eta_{k-j} \right]$$

$$+ \frac{8\beta e(\beta + \gamma)}{n} \sum_{k=1}^{K} \eta_k (1 + \frac{k}{n}) \left[ \eta_1 \|\mathbf{w}_1 - \mathbf{w}^*\|^2 + 2 \sum_{l=1}^{k} \eta_l^2 F(\mathbf{w}^*) + 4\beta r^2 \sum_{l=1}^{k} \eta_l^2 \right].$$

Utilizing the convexity property of the function $F$, we can derive

$$\mathbb{E}_{\mathcal{S},A}[F(\overline{\mathbf{w}}_K) - F(\mathbf{w}^*)] \leq \frac{\sum_{k=1}^{K} \eta_k \mathbb{E}_{\mathcal{S},A}[F(\mathbf{w}_k) - F(\mathbf{w}^*)]}{\sum_{k=1}^{K} \eta_k}$$

$$\leq \frac{1 + \beta/\gamma}{\sum_{k=1}^{K} \eta_k} \left[ (\frac{1}{2} + 2\beta\eta_1)\|\mathbf{w}_1 - \mathbf{w}^*\|^2 + 4\beta \sum_{k=1}^{K} \eta_k^2 F(\mathbf{w}^*) + 8\beta^2 r^2 \sum_{k=1}^{K} \eta_k^2 + \beta L r \sum_{k=1}^{K} \eta_k \sum_{j=1}^{\tau_k} \eta_{k-j} \right]$$

$$+ \frac{8\beta e(\beta + \gamma)(1 + K/n)}{n} \left[ \eta_1 \|\mathbf{w}_1 - \mathbf{w}^*\|^2 + 2 \sum_{k=1}^{K} \eta_k^2 F(\mathbf{w}^*) + 4\beta r^2 \sum_{k=1}^{K} \eta_k^2 \right]$$

$$+ \beta^2 r^2 (\beta + \gamma) e \sum_{k=1}^{K} \eta_k \sum_{l=1}^{k} \eta_l \sum_{j=1}^{\tau_l} \eta_{l-j} / \sum_{k=1}^{K} \eta_k + \frac{\beta}{\gamma} F(\mathbf{w}^*).$$
(B.15)

By setting $\gamma = 1$, we conclude that the excess generalization error $\epsilon_{\text{ex-gen}}$ is

$$\epsilon_{\text{ex-gen}} = \mathcal{O}\left( \left[ 1 + \frac{\sum_{k=1}^{K} \eta_k^2}{\sum_{k=1}^{K} \eta_k} \right] F(\mathbf{w}^*) + \frac{1 + K/n}{n} \left[ \eta_1 \|\mathbf{w}_1 - \mathbf{w}^*\|^2 + (1 + F(\mathbf{w}^*)) \sum_{k=1}^{K} \eta_k^2 \right] \right.$$

$$\left. + \frac{\|\mathbf{w}_1 - \mathbf{w}^*\|^2}{\sum_{k=1}^{K} \eta_k} + \left[ \sum_{k=1}^{K} \eta_k^2 + \sum_{k=1}^{K} \eta_k \sum_{j=1}^{\tau_k} \eta_{k-j} + \sum_{k=1}^{K} \eta_k \sum_{l=1}^{k} \eta_l \sum_{j=1}^{\tau_l} \eta_{l-j} \right] / \sum_{k=1}^{K} \eta_k \right).$$

## B.7 Proof of Corollary 2 (excess generalization error with a specific learning rate)

Let the learning rate $\eta_k = c(\bar{\tau}\sqrt{K})^{-1}$ with a constant $c>0$, and $\bar{\tau}=\sum_{k=1}^{K}\tau_k/K$, direct calculation gives

$$\sum_{k=1}^{K}\eta_k = c\sqrt{K}/\bar{\tau}, \qquad \sum_{k=1}^{K}\eta_k^2 = c^2/\bar{\tau}^2, \qquad \sum_{k=1}^{K}\eta_k\sum_{j=1}^{\tau_k}\eta_{k-j} = \frac{c^2}{\bar{\tau}^2 K}\sum_{k=1}^{K}\tau_k = \frac{c^2}{\bar{\tau}},$$

$$\sum_{k=1}^{K}\eta_k\sum_{l=1}^{k}\eta_l\sum_{j=1}^{\tau_l}\eta_{l-j} = \Big(\frac{c}{\bar{\tau}\sqrt{K}}\Big)^3\sum_{k=1}^{K}\sum_{l=1}^{k}\tau_l \le \frac{c^3\sqrt{K}}{\bar{\tau}^2}.$$

From the excess generalization error (B.15), we can derive

$$\mathbb{E}_{\mathcal{S},A}\left[F(\overline{\mathbf{w}}_K) - F(\mathbf{w}^*)\right]$$

$$\le (1+\frac{\beta}{\gamma})\left[(\frac{1}{2}+\frac{2\beta c}{\bar{\tau}\sqrt{K}})\|\mathbf{w}_1-\mathbf{w}^*\|^2 + \frac{4\beta c^2}{\bar{\tau}^2}F(\mathbf{w}^*) + \frac{8\beta^2 r^2 c^2}{\bar{\tau}^2} + \frac{\beta L r c^2}{\bar{\tau}}\right]\cdot\frac{\bar{\tau}}{c\sqrt{K}}$$

$$+ \frac{8\beta e(\beta+\gamma)(1+K/n)}{n}\left[\frac{c}{\bar{\tau}\sqrt{K}}\|\mathbf{w}_1-\mathbf{w}^*\|^2 + \frac{2c^2}{\bar{\tau}^2}F(\mathbf{w}^*) + \frac{4\beta r^2 c^2}{\bar{\tau}^2}\right]$$

$$+ \beta^2 r^2(\beta+\gamma)e\frac{c^3\sqrt{K}}{\bar{\tau}^2}\cdot\frac{\bar{\tau}}{c\sqrt{K}} + \frac{\beta}{\gamma}F(\mathbf{w}^*)$$

$$\le (1+\frac{\beta}{\gamma})\left[\frac{\bar{\tau}}{\sqrt{K}}\frac{\|\mathbf{w}_1-\mathbf{w}^*\|^2}{2c} + \beta rc(\frac{8\beta r}{\sqrt{K}\bar{\tau}} + \frac{L}{\sqrt{K}}) + \frac{2\beta\|\mathbf{w}_1-\mathbf{w}^*\|^2}{K})\right]$$

$$+ \frac{8\beta e(\beta+\gamma)(1+K/n)}{n}\left[\frac{c}{\bar{\tau}\sqrt{K}}\|\mathbf{w}_1-\mathbf{w}^*\|^2 + \frac{2c^2}{\bar{\tau}^2}F(\mathbf{w}^*) + \frac{4\beta r^2 c^2}{\bar{\tau}^2}\right] + \frac{\beta^2 r^2(\beta+\gamma)ec^2}{\bar{\tau}}$$

$$+ \left(\frac{\beta}{\gamma} + (1+\frac{\beta}{\gamma})\frac{4\beta c}{\bar{\tau}\sqrt{K}}\right)F(\mathbf{w}^*).$$

Let $F(\mathbf{w}^*) = 0$, $K\asymp n$ and $\gamma = 1$. If the average delay satisfies $\bar{\tau}\le K^{\frac{1}{4}}$ (quite reasonable in asynchronous training), we have $\max\{\frac{\bar{\tau}}{\sqrt{K}}, \frac{1}{\sqrt{K}\bar{\tau}}, \frac{1}{\sqrt{K}}\}\le\frac{1}{\bar{\tau}}$. Then the excess generalization error is

$$\mathbb{E}_{\mathcal{S},A}\left[F(\overline{\mathbf{w}}_K) - F(\mathbf{w}^*)\right] = \mathcal{O}\Big(\frac{1}{\bar{\tau}} + \frac{\|\mathbf{w}_1-\mathbf{w}^*\|^2}{n}\Big).$$

# C   Proof of Generalization in Non-smooth Case (Section 5 in the main text)

## C.1   Proof of Lemma 5 (approximately non-expansive property in the non-smooth case)

Under $(\alpha,\beta)$-Hölder continuous condition, the gradients exhibit the following co-coercivity

$$\langle\mathbf{w}_{k-\tau_k}-\mathbf{w}^{(i)}_{k-\tau_k}, \nabla f(\mathbf{w}_{k-\tau_k};\mathbf{z}_{i_k})-\nabla f(\mathbf{w}^{(i)}_{k-\tau_k};\mathbf{z}_{i_k})\rangle \ge \frac{2\beta^{-\frac{1}{\alpha}}\alpha}{1+\alpha}\|\nabla f(\mathbf{w}_{k-\tau_k};\mathbf{z}_{i_k})-\nabla f(\mathbf{w}^{(i)}_{k-\tau_k};\mathbf{z}_{i_k})\|^{\frac{1+\alpha}{\alpha}}.$$

A detailed proof of this co-coercivity can be found in [24, 39, 51], and also in Lemma D.2 of [23]. Then we have

$$\|\nabla f(\mathbf{w}_{k-\tau_k};\mathbf{z}_{i_k})-\nabla f(\mathbf{w}^{(i)}_{k-\tau_k};\mathbf{z}_{i_k})\|^2 \le \Big(\frac{1+\alpha}{2\beta^{-\frac{1}{\alpha}}\alpha}\langle\mathbf{w}_{k-\tau_k}-\mathbf{w}^{(i)}_{k-\tau_k}, \nabla f(\mathbf{w}_{k-\tau_k};\mathbf{z}_{i_k})-\nabla f(\mathbf{w}^{(i)}_{k-\tau_k};\mathbf{z}_{i_k})\rangle\Big)^{\frac{2\alpha}{1+\alpha}}$$

$$= \Big(\frac{1+\alpha}{\eta_k\alpha}\langle\mathbf{w}_{k-\tau_k}-\mathbf{w}^{(i)}_{k-\tau_k}, \nabla f(\mathbf{w}_{k-\tau_k};\mathbf{z}_{i_k})-\nabla f(\mathbf{w}^{(i)}_{k-\tau_k};\mathbf{z}_{i_k})\rangle\Big)^{\frac{2\alpha}{1+\alpha}}\cdot\Big(2^{-\frac{2\alpha}{1+\alpha}}\eta_k^{\frac{2\alpha}{1+\alpha}}\beta^{\frac{2}{1+\alpha}}\Big)$$

$$\le \frac{2}{\eta_k}\langle\mathbf{w}_{k-\tau_k}-\mathbf{w}^{(i)}_{k-\tau_k}, \nabla f(\mathbf{w}_{k-\tau_k};\mathbf{z}_{i_k})-\nabla f(\mathbf{w}^{(i)}_{k-\tau_k};\mathbf{z}_{i_k})\rangle + \frac{1-\alpha}{1+\alpha}\eta_k^{\frac{2\alpha}{1-\alpha}}\big(2^{-\alpha}\beta\big)^{\frac{2}{1-\alpha}},$$

where the last inequality we use the Young's inequality (A.1) with $p = \frac{1+\alpha}{2\alpha}, q = \frac{1+\alpha}{1-\alpha}$. That is

$$\eta_k^2\|\nabla f(\mathbf{w}_{k-\tau_k};\mathbf{z}_{i_k})-\nabla f(\mathbf{w}^{(i)}_{k-\tau_k};\mathbf{z}_{i_k})\|^2 \le 2\eta_k\langle\mathbf{w}_{k-\tau_k}-\mathbf{w}^{(i)}_{k-\tau_k}, \nabla f(\mathbf{w}_{k-\tau_k};\mathbf{z}_{i_k})-\nabla f(\mathbf{w}^{(i)}_{k-\tau_k};\mathbf{z}_{i_k})\rangle$$

$$+ d_{\alpha,\beta}^2\eta_k^{\frac{2}{1-\alpha}}, \tag{C.1}$$

where $d_{\alpha,\beta} = \sqrt{\frac{1-\alpha}{1+\alpha}} \left(2^{-\alpha}\beta\right)^{\frac{1}{1-\alpha}} > 0$ is a constant dependent on $\alpha, \beta$. Without smoothness, the non-expansive property of delayed gradient updates would be further compromised. However, in conjunction with inequality (C.1), we can make the following derivation.

$$\left\| \mathbf{w}_k - \eta_k \nabla f(\mathbf{w}_{k-\tau_k}; \mathbf{z}_{i_k}) - \left( \mathbf{w}_k^{(i)} - \eta_k \nabla f(\mathbf{w}_{k-\tau_k}^{(i)}; \mathbf{z}_{i_k}) \right) \right\|^2$$

$$= \|\mathbf{w}_k - \mathbf{w}_k^{(i)}\|^2 + \eta_k^2 \|\nabla f(\mathbf{w}_{k-\tau_k}; \mathbf{z}_{i_k}) - \nabla f(\mathbf{w}_{k-\tau_k}^{(i)}; \mathbf{z}_{i_k})\|^2$$

$$\quad - 2\eta_k \langle \mathbf{w}_k - \mathbf{w}_k^{(i)}, \nabla f(\mathbf{w}_{k-\tau_k}; \mathbf{z}_{i_k}) - \nabla f(\mathbf{w}_{k-\tau_k}^{(i)}; \mathbf{z}_{i_k}) \rangle$$

$$= \|\mathbf{w}_k - \mathbf{w}_k^{(i)}\|^2 + \eta_k^2 \|\nabla f(\mathbf{w}_{k-\tau_k}; \mathbf{z}_{i_k}) - \nabla f(\mathbf{w}_{k-\tau_k}^{(i)}; \mathbf{z}_{i_k})\|^2$$

$$\quad - 2\eta_k \langle \mathbf{w}_{k-\tau_k} - \mathbf{w}_{k-\tau_k}^{(i)}, \nabla f(\mathbf{w}_{k-\tau_k}; \mathbf{z}_{i_k}) - \nabla f(\mathbf{w}_{k-\tau_k}^{(i)}; \mathbf{z}_{i_k}) \rangle$$

$$\quad - 2\eta_k \langle \mathbf{w}_k - \mathbf{w}_{k-\tau_k} - (\mathbf{w}_k^{(i)} - \mathbf{w}_{k-\tau_k}^{(i)}), \nabla f(\mathbf{w}_{k-\tau_k}; \mathbf{z}_{i_k}) - \nabla f(\mathbf{w}_{k-\tau_k}^{(i)}; \mathbf{z}_{i_k}) \rangle$$

$$\leq \|\mathbf{w}_k - \mathbf{w}_k^{(i)}\|^2 - 2\eta_k \langle \mathbf{w}_k - \mathbf{w}_{k-\tau_k} - (\mathbf{w}_k^{(i)} - \mathbf{w}_{k-\tau_k}^{(i)}), \nabla f(\mathbf{w}_{k-\tau_k}; \mathbf{z}_{i_k}) - \nabla f(\mathbf{w}_{k-\tau_k}^{(i)}; \mathbf{z}_{i_k}) \rangle$$

$$\quad + d_{\alpha,\beta}^2 \eta_k^{\frac{2}{1-\alpha}}.$$

Following the $(\alpha, \beta)$-Hölder continuous condition and Assumption 1, we can derive

$$\|\nabla f(\mathbf{w}_{s-\tau_s}; \mathbf{z}_{i_s}) - \nabla f(\mathbf{w}_{s-\tau_s}^{(i)}; \mathbf{z}_{i_s})\| \leq \beta \|\mathbf{w}_{s-\tau_s} - \mathbf{w}_{s-\tau_s}^{(i)}\|^\alpha \leq \beta r^\alpha, \text{ for } s = k, k-j; \ j = 1, \ldots, \tau_k.$$
(C.2)

By combing (B.1) and (C.2), the asynchronous gradient updates satisfy the following approximately non-expansive property under the $(\alpha, \beta)$-Hölder continuous condition.

$$\left\| \mathbf{w}_k - \eta_k \nabla f(\mathbf{w}_{k-\tau_k}; \mathbf{z}_{i_k}) - \left( \mathbf{w}_k^{(i)} - \eta_k \nabla f(\mathbf{w}_{k-\tau_k}^{(i)}; \mathbf{z}_{i_k}) \right) \right\|^2 \leq \left\| \mathbf{w}_k - \mathbf{w}_k^{(i)} \right\|^2 + d_{\alpha,\beta}^2 \eta_k^{\frac{2}{1-\alpha}}$$

$$+ 2\beta^2 r^{2\alpha} \eta_k \sum_{j=1}^{\tau_k} \eta_{k-j}.$$
(C.3)

## C.2 Proof of Theorem 4 (algorithm stability under the $(\alpha, \beta)$-Hölder continuous gradient assumption)

We now examine the on-average model stability of the ASGD algorithm under the $(\alpha, \beta)$-Hölder continuous gradient assumption. Following (C.3), if ASGD selects the same sample in both $\mathcal{S}$ and $\mathcal{S}^{(i)}$ at the $k$-th iteration (with probability $1 - 1/n$), we have

$$\left\| \mathbf{w}_{k+1} - \mathbf{w}_{k+1}^{(i)} \right\|^2 \leq \left\| \mathbf{w}_k - \eta_k \nabla f(\mathbf{w}_{k-\tau_k}; \mathbf{z}_{i_k}) - \left( \mathbf{w}_k^{(i)} - \eta_k \nabla f(\mathbf{w}_{k-\tau_k}^{(i)}; \mathbf{z}_{i_k}) \right) \right\|^2$$

$$\leq \|\mathbf{w}_k - \mathbf{w}_k^{(i)}\|^2 + 2\eta_k \beta^2 r^{2\alpha} \sum_{j=1}^{\tau_k} \eta_{k-j} + d_{\alpha,\beta}^2 \eta_k^{\frac{2}{1-\alpha}},$$

where the first inequality uses the no-expansive projection (A.8). On the other hand, with probability $1/n$ the selected example is different ($i_k = i$), then

$$\|\mathbf{w}_{k+1} - \mathbf{w}_{k+1}^{(i)}\|^2 \leq \|\mathbf{w}_k - \eta_k \nabla f(\mathbf{w}_{k-\tau_k}; \mathbf{z}_i) - \mathbf{w}_k^{(i)} + \eta_k \nabla f(\mathbf{w}_{k-\tau_k}^{(i)}; \mathbf{z}_i')\|^2$$

$$\leq (1+p)\|\mathbf{w}_k - \mathbf{w}_k^{(i)}\|^2 + (1+1/p)\eta_k^2 \|\nabla f(\mathbf{w}_{k-\tau_k}; \mathbf{z}_i) - \nabla f(\mathbf{w}_{k-\tau_k}^{(i)}; \mathbf{z}_i')\|^2$$

$$\leq (1+p)\|\mathbf{w}_k - \mathbf{w}_k^{(i)}\|^2 + 2(1+1/p)\eta_k^2 \left[ \|\nabla f(\mathbf{w}_{k-\tau_k}; \mathbf{z}_i)\|^2 + \|\nabla f(\mathbf{w}_{k-\tau_k}^{(i)}; \mathbf{z}_i')\|^2 \right]$$

$$\leq (1+p)\|\mathbf{w}_k - \mathbf{w}_k^{(i)}\|^2 + 2(1+1/p)c_{\alpha,\beta}^2 \eta_k^2 \left[ f^{\frac{2\alpha}{1+\alpha}}(\mathbf{w}_{k-\tau_k}; \mathbf{z}_i) + f^{\frac{2\alpha}{1+\alpha}}(\mathbf{w}_{k-\tau_k}^{(i)}; \mathbf{z}_i') \right].$$

Here we use the inequality (A.4) and the self-bounding property of the $\alpha, \beta$-Hölder continuous gradient (A.5). Combining the above two cases gives

$$\|\mathbf{w}_{k+1} - \mathbf{w}_{k+1}^{(i)}\|^2 \leq (1 + \frac{p}{n})\|\mathbf{w}_k - \mathbf{w}_k^{(i)}\|^2 + 2\eta_k \beta^2 r^{2\alpha} \sum_{j=1}^{\tau_k} \eta_{k-j} + d_{\alpha,\beta}^2 \eta_k^{\frac{2}{1-\alpha}}$$

$$+ \frac{2(1+1/p)c_{\alpha,\beta}^2 \eta_k^2}{n} \left[ f^{\frac{2\alpha}{1+\alpha}}(\mathbf{w}_{k-\tau_k}; \mathbf{z}_i) + f^{\frac{2\alpha}{1+\alpha}}(\mathbf{w}_{k-\tau_k}^{(i)}; \mathbf{z}_i') \right].$$
(C.4)

Following the fact

$$\mathbb{E}_{\mathcal{S},\mathcal{S}',A}[f^{\frac{2\alpha}{1+\alpha}}(\mathbf{w}^{(i)}_{k-\tau_k};\mathbf{z}'_i)] = \mathbb{E}_{\mathcal{S},A}[f^{\frac{2\alpha}{1+\alpha}}(\mathbf{w}_{k-\tau_k};\mathbf{z}_i)],$$

and the same model initialization $\mathbf{w}_1 = \mathbf{w}^{(i)}_1$, taking the expectation followed by summation of inequality (C.4) yields

$$\mathbb{E}_{\mathcal{S},\mathcal{S}',A}\|\mathbf{w}_{k+1} - \mathbf{w}^{(i)}_{k+1}\|^2 \le (1+\frac{p}{n})\mathbb{E}_{\mathcal{S},\mathcal{S}',A}\|\mathbf{w}_k - \mathbf{w}^{(i)}_k\|^2 + 2\eta_k\beta^2 r^{2\alpha}\sum_{j=1}^{\tau_k}\eta_{k-j} + d^2_{\alpha,\beta}\eta_k^{\frac{2}{1-\alpha}}$$

$$+ \frac{4(1+1/p)c^2_{\alpha,\beta}\eta_k^2}{n}\mathbb{E}_{\mathcal{S},A}\left[f^{\frac{2\alpha}{1+\alpha}}(\mathbf{w}_{k-\tau_k};\mathbf{z}_i)\right]$$

$$\le \sum_{l=1}^{k}(1+\frac{p}{n})^{(k-l)}\left[\frac{4(1+1/p)c^2_{\alpha,\beta}\eta_l^2}{n}\mathbb{E}_{\mathcal{S},A}\left[f^{\frac{2\alpha}{1+\alpha}}(\mathbf{w}_{l-\tau_l};\mathbf{z}_i)\right] + 2\eta_l\beta^2 r^{2\alpha}\sum_{j=1}^{\tau_l}\eta_{l-j} + d^2_{\alpha,\beta}\eta_l^{\frac{2}{1-\alpha}}\right].$$

By the concavity of the function $x \mapsto x^{\frac{2\alpha}{1+\alpha}}$, the on-average model stability of ASGD satisfies

$$\mathbb{E}_{\mathcal{S},\mathcal{S}',A}\left[\frac{1}{n}\sum_{i=1}^{n}\|\mathbf{w}_{k+1} - \mathbf{w}^{(i)}_{k+1}\|^2\right]$$

$$\le (1+\frac{p}{n})^{(k-1)}\sum_{l=1}^{k}\left[\frac{4(1+1/p)c^2_{\alpha,\beta}\eta_l^2}{n}\mathbb{E}_{\mathcal{S},A}\left[F_{\mathcal{S}}^{\frac{2\alpha}{1+\alpha}}(\mathbf{w}_{l-\tau_l})\right] + 2\eta_l\beta^2 r^{2\alpha}\sum_{j=1}^{\tau_l}\eta_{l-j} + d^2_{\alpha,\beta}\eta_l^{\frac{2}{1-\alpha}}\right].$$

Let $p = n/k$, we have $(1+p/n)^{(k-1)} \le (1+1/k)^{(k-1)} \le \mathrm{e}$, then

$$\mathbb{E}_{\mathcal{S},\mathcal{S}',A}\left[\frac{1}{n}\sum_{i=1}^{n}\|\mathbf{w}_{k+1} - \mathbf{w}^{(i)}_{k+1}\|^2\right]$$

$$\le \frac{4\mathrm{e}(1+k/n)c^2_{\alpha,\beta}}{n}\sum_{l=1}^{k}\eta_l^2\mathbb{E}_{\mathcal{S},A}\left[F_{\mathcal{S}}^{\frac{2\alpha}{1+\alpha}}(\mathbf{w}_{l-\tau_l})\right] + 2\beta^2 r^{2\alpha}\mathrm{e}\sum_{l=1}^{k}\eta_l\sum_{j=1}^{\tau_l}\eta_{l-j} + \mathrm{e}d^2_{\alpha,\beta}\sum_{l=1}^{k}\eta_l^{\frac{2}{1-\alpha}}$$

$$= \mathcal{O}\left(\frac{1+k/n}{n}\sum_{l=1}^{k}\eta_l^2\mathbb{E}_{\mathcal{S},A}\left[F_{\mathcal{S}}^{\frac{2\alpha}{1+\alpha}}(\mathbf{w}_{l-\tau_l})\right] + \sum_{l=1}^{k}\eta_l\sum_{j=1}^{\tau_l}\eta_{l-j} + \sum_{l=1}^{k}\eta_l^{\frac{2}{1-\alpha}}\right).$$

(C.5)

### C.3 Generalization Error with $(\alpha,\beta)$-Hölder Continuous Gradient

From the algorithm stability (C.5), it follows that we need to bound $\sum_{l=1}^{k}\eta_l^2\mathbb{E}_{\mathcal{S},A}[F_{\mathcal{S}}^{\frac{2\alpha}{1+\alpha}}(\mathbf{w}_{l-\tau_l})]$ under the $\alpha,\beta$-Hölder continuous gradient condition. Using the non-expansive projection (A.8) and the self-bounding property (A.5), we can derive

$$\begin{aligned}
\|\mathbf{w}_{k+1} - \mathbf{w}^*\|^2 &\le \|\mathbf{w}_k - \eta_k\nabla f(\mathbf{w}_{k-\tau_k};\mathbf{z}_{i_k}) - \mathbf{w}^*\|^2 \\
&= \|\mathbf{w}_k - \mathbf{w}^*\|^2 + \eta_k^2\|\nabla f(\mathbf{w}_{k-\tau_k};\mathbf{z}_{i_k})\|^2 + 2\eta_k\langle\mathbf{w}^* - \mathbf{w}_k, \nabla f(\mathbf{w}_{k-\tau_k};\mathbf{z}_{i_k})\rangle \\
&\le \|\mathbf{w}_k - \mathbf{w}^*\|^2 + c^2_{\alpha,\beta}\eta_k^2 f^{\frac{2\alpha}{1+\alpha}}(\mathbf{w}_{k-\tau_k};\mathbf{z}_{i_k}) + 2\eta_k\langle\mathbf{w}^* - \mathbf{w}_k, \nabla f(\mathbf{w}_{k-\tau_k};\mathbf{z}_{i_k})\rangle.
\end{aligned}$$

By the young's inequality (A.1) with $p = \frac{1+\alpha}{1-\alpha}, q = \frac{1+\alpha}{2\alpha}$, we have

$$\begin{aligned}
c^2_{\alpha,\beta}\eta_k f^{\frac{2\alpha}{1+\alpha}}(\mathbf{w}_{k-\tau_k};\mathbf{z}_{i_k}) &= \left(\left(\frac{2\alpha}{1+\alpha}\right)^{\frac{2\alpha}{1+\alpha}}c^2_{\alpha,\beta}\eta_k\right) \cdot \left(\frac{1+\alpha}{2\alpha}f(\mathbf{w}_{k-\tau_k};\mathbf{z}_{i_k})\right)^{\frac{2\alpha}{1+\alpha}} \\
&\le \frac{1-\alpha}{1+\alpha}\left(\left(\frac{2\alpha}{1+\alpha}\right)^{\frac{2\alpha}{1+\alpha}}c^2_{\alpha,\beta}\eta_k\right)^{\frac{1+\alpha}{1-\alpha}} + \frac{2\alpha}{1+\alpha}\left(\frac{1+\alpha}{2\alpha}f(\mathbf{w}_{k-\tau_k};\mathbf{z}_{i_k})\right)^{\frac{2\alpha}{1+\alpha}\frac{1+\alpha}{2\alpha}} \\
&= f(\mathbf{w}_{k-\tau_k};\mathbf{z}_{i_k}) + c'_{\alpha,\beta}\eta_k^{\frac{1+\alpha}{1-\alpha}},
\end{aligned}$$

where we define the constant $c'_{\alpha,\beta} = \frac{1-\alpha}{1+\alpha}\left(\frac{2\alpha}{1+\alpha}\right)^{\frac{2\alpha}{1-\alpha}} c_{\alpha,\beta}^{\frac{2+2\alpha}{1-\alpha}} > 0$. With the convexity of $f$, we can further derive

$$
\begin{aligned}
\|\mathbf{w}_{k+1} - \mathbf{w}^*\|^2 &\le \|\mathbf{w}_k - \mathbf{w}^*\|^2 + \eta_k f(\mathbf{w}_{k-\tau_k}; \mathbf{z}_{i_k}) + 2\eta_k \langle \mathbf{w}^* - \mathbf{w}_k, \nabla f(\mathbf{w}_{k-\tau_k}; \mathbf{z}_{i_k}) \rangle + c'_{\alpha,\beta} \eta_k^{\frac{2}{1-\alpha}} \\
&\le \|\mathbf{w}_k - \mathbf{w}^*\|^2 + \eta_k f(\mathbf{w}_{k-\tau_k}; \mathbf{z}_{i_k}) + 2\eta_k(f(\mathbf{w}^*; \mathbf{z}_{i_k}) - f(\mathbf{w}_{k-\tau_k}; \mathbf{z}_{i_k})) \\
&\quad + 2\eta_k \langle \mathbf{w}_{k-\tau_k} - \mathbf{w}_k, \nabla f(\mathbf{w}_{k-\tau_k}; \mathbf{z}_{i_k}) \rangle + c'_{\alpha,\beta} \eta_k^{\frac{2}{1-\alpha}} \\
&\le \|\mathbf{w}_k - \mathbf{w}^*\|^2 - \eta_k f(\mathbf{w}_{k-\tau_k}; \mathbf{z}_{i_k}) + 2\eta_k f(\mathbf{w}^*; \mathbf{z}_{i_k}) + 2\eta_k \langle \mathbf{w}_{k-\tau_k} - \mathbf{w}_k, \nabla f(\mathbf{w}_{k-\tau_k}; \mathbf{z}_{i_k}) \rangle \\
&\quad + c'_{\alpha,\beta} \eta_k^{\frac{2}{1-\alpha}}.
\end{aligned}
\tag{C.6}
$$

Here, with the inequality (A.2), (A.5) and Assumption 1, we have the following derivation

$$
\begin{aligned}
2\eta_k \langle \mathbf{w}_{k-\tau_k} - \mathbf{w}_k, \nabla f(\mathbf{w}_{k-\tau_k}; \mathbf{z}_{i_k}) \rangle &\le 2\eta_k r \cdot c_{\alpha,\beta} f^{\frac{\alpha}{1+\alpha}}(\mathbf{w}_{k-\tau_k}; \mathbf{z}_{i_k}) \\
&\le \left(\frac{(1+\alpha)\eta_k}{2\alpha} f(\mathbf{w}_{k-\tau_k}; \mathbf{z}_{i_k})\right)^{\frac{\alpha}{1+\alpha}} \cdot 2^{\frac{1+2\alpha}{1+\alpha}} \eta_k^{\frac{1}{1+\alpha}} \left(\frac{\alpha}{1+\alpha}\right)^{\frac{\alpha}{1+\alpha}} c_{\alpha,\beta} r \\
&\le \frac{\eta_k}{2} f(\mathbf{w}_{k-\tau_k}; \mathbf{z}_{i_k}) + c''_{\alpha,\beta} \eta_k,
\end{aligned}
$$

where the last inequality we use the young's inequality (A.1) with $p = \frac{1+\alpha}{\alpha}, q = 1 + \alpha$, and $c''_{\alpha,\beta} = 2^{1+2\alpha}\alpha^{\alpha}\left(\frac{rc_{\alpha,\beta}}{1+\alpha}\right)^{1+\alpha}$ is a constant. Substituting it into (C.6) yields

$$
\eta_k f(\mathbf{w}_{k-\tau_k}; \mathbf{z}_{i_k}) \le 2\|\mathbf{w}_k - \mathbf{w}^*\|^2 - 2\|\mathbf{w}_{k+1} - \mathbf{w}^*\|^2 + 4\eta_k f(\mathbf{w}^*; \mathbf{z}_{i_k}) + 2c'_{\alpha,\beta} \eta_k^{\frac{2}{1-\alpha}} + 2c''_{\alpha,\beta} \eta_k.
$$

Multiplying both sides by the non-increasing learning rate yields

$$
\eta_k^2 f(\mathbf{w}_{k-\tau_k}; \mathbf{z}_{i_k}) \le 2\eta_k\|\mathbf{w}_k - \mathbf{w}^*\|^2 - 2\eta_{k+1}\|\mathbf{w}_{k+1} - \mathbf{w}^*\|^2 + 4\eta_k^2 f(\mathbf{w}^*; \mathbf{z}_{i_k}) + 2c'_{\alpha,\beta} \eta_k^{\frac{3-\alpha}{1-\alpha}} + 2c''_{\alpha,\beta} \eta_k^2.
$$

Taking the expectation and summing the inequalities as above gives

$$
\sum_{l=1}^{k} \eta_l^2 \mathbb{E}_{\mathcal{S},A}[F_{\mathcal{S}}(\mathbf{w}_{l-\tau_l})] \le 2\eta_1\|\mathbf{w}_1 - \mathbf{w}^*\|^2 + 4\sum_{l=1}^{k} \eta_l^2 \mathbb{E}_{\mathcal{S}}[F_{\mathcal{S}}(\mathbf{w}^*)] + 2c'_{\alpha,\beta} \sum_{l=1}^{k} \eta_l^{\frac{3-\alpha}{1-\alpha}} + 2c''_{\alpha,\beta} \sum_{l=1}^{k} \eta_l^2.
$$

According to the concavity of the function $x \mapsto x^{\frac{2\alpha}{1+\alpha}}$, we know that

$$
\begin{aligned}
\sum_{l=1}^{k} \eta_l^2 \mathbb{E}_{\mathcal{S},A}[F_{\mathcal{S}}^{\frac{2\alpha}{1+\alpha}}(\mathbf{w}_{l-\tau_l})] &\le \sum_{l=1}^{k} \eta_l^2 \left(\frac{\sum_{l=1}^{k} \eta_l^2 \mathbb{E}_{\mathcal{S},A}[F_{\mathcal{S}}(\mathbf{w}_{l-\tau_l})]}{\sum_{l=1}^{k} \eta_l^2}\right)^{\frac{2\alpha}{1+\alpha}} \\
&\le 2\left(\sum_{l=1}^{k} \eta_l^2\right)^{\frac{1-\alpha}{1+\alpha}} \left(\eta_1\|\mathbf{w}_1 - \mathbf{w}^*\|^2 + 2\sum_{l=1}^{k} \eta_l^2 F(\mathbf{w}^*) + c'_{\alpha,\beta} \sum_{l=1}^{k} \eta_l^{\frac{3-\alpha}{1-\alpha}} + c''_{\alpha,\beta} \sum_{l=1}^{k} \eta_l^2\right)^{\frac{2\alpha}{1+\alpha}}.
\end{aligned}
\tag{C.7}
$$

Now, we are ready to analysis the generalization error of ASGD under $(\alpha,\beta)$-Hölder continuous gradient condition. From Lemma 2 (A.13) and stability (C.5), we have

$$
\begin{aligned}
\mathbb{E}_{\mathcal{S},A}\left[F(\mathbf{w}_{k+1}) - F_{\mathcal{S}}(\mathbf{w}_{k+1})\right] &\le \frac{c_{\alpha,\beta}^2}{2\gamma} \mathbb{E}_{\mathcal{S},A}[F^{\frac{2\alpha}{1+\alpha}}(\mathbf{w}_{k+1})] + \frac{\gamma}{2} \sum_{i=1}^{n} \mathbb{E}_{\mathcal{S},\mathcal{S}',A}\left[\frac{1}{n}\|\mathbf{w}_{k+1} - \mathbf{w}_{k+1}^{(i)}\|^2\right] \\
&\le \frac{c_{\alpha,\beta}^2}{2\gamma} \mathbb{E}_{\mathcal{S},A}[F^{\frac{2\alpha}{1+\alpha}}(\mathbf{w}_{k+1})] + \frac{2\mathrm{e}\gamma(1+k/n)c_{\alpha,\beta}^2}{n} \sum_{l=1}^{k} \eta_l^2 \mathbb{E}_{\mathcal{S},A}\left[F_{\mathcal{S}}^{\frac{2\alpha}{1+\alpha}}(\mathbf{w}_{l-\tau_l})\right] + \mathrm{e}\gamma\beta^2 r^{2\alpha} \sum_{l=1}^{k} \eta_l \sum_{j=1}^{\tau_l} \eta_{l-j} \\
&\quad + \frac{\mathrm{e}\gamma d_{\alpha,\beta}^2}{2} \sum_{l=1}^{k} \eta_l^{\frac{2}{1-\alpha}}.
\end{aligned}
\tag{C.8}
$$

Let $\epsilon_k := \max\left\{\mathbb{E}_{\mathcal{S},A}\left[F(\mathbf{w}_k) - F_{\mathcal{S}}(\mathbf{w}_k)\right], 0\right\}$. By the concavity and sub-additivity of $x \mapsto x^{\frac{2\alpha}{1+\alpha}}$ we have

$$
\begin{aligned}
\mathbb{E}_{\mathcal{S},A}[F^{\frac{2\alpha}{1+\alpha}}(\mathbf{w}_{k+1})] &\le \left(\mathbb{E}_{\mathcal{S},A}\left[F(\mathbf{w}_{k+1}) - F_{\mathcal{S}}(\mathbf{w}_{k+1})\right] + \mathbb{E}_{\mathcal{S},A}\left[F_{\mathcal{S}}(\mathbf{w}_{k+1})\right]\right)^{\frac{2\alpha}{1+\alpha}} \\
&\le \epsilon_{k+1}^{\frac{2\alpha}{1+\alpha}} + \left(\mathbb{E}_{\mathcal{S},A}\left[F_{\mathcal{S}}(\mathbf{w}_{k+1})\right]\right)^{\frac{2\alpha}{1+\alpha}}.
\end{aligned}
$$

Substituting this back into (C.8) gives

$$\epsilon_{k+1} \leq \frac{c_{\alpha,\beta}^2}{2\gamma}\left(\epsilon_{k+1}^{\frac{2\alpha}{1+\alpha}} + \left(\mathbb{E}_{\mathcal{S},A}\big[F_{\mathcal{S}}\left(\mathbf{w}_{k+1}\right)\big]\right)^{\frac{2\alpha}{1+\alpha}}\right) + \frac{2\mathrm{e}\gamma(1+k/n)c_{\alpha,\beta}^2}{n}\sum_{l=1}^{k}\eta_l^2\mathbb{E}_{\mathcal{S},A}\left[F_{\mathcal{S}}^{\frac{2\alpha}{1+\alpha}}\left(\mathbf{w}_{l-\tau_l}\right)\right]$$

$$+ \mathrm{e}\gamma\beta^2 r^{2\alpha}\sum_{l=1}^{k}\eta_l\sum_{j=1}^{\tau_l}\eta_{l-j} + \frac{\mathrm{e}\gamma d_{\alpha,\beta}^2}{2}\sum_{l=1}^{k}\eta_l^{\frac{2}{1-\alpha}}.$$

Using the young's inequality (A.1) with $p = \frac{1+\alpha}{1-\alpha}, q = \frac{1+\alpha}{2\alpha}$ yields

$$\frac{c_{\alpha,\beta}^2}{2\gamma}\epsilon_{k+1}^{\frac{2\alpha}{1+\alpha}} = \left(\frac{4\alpha}{1+\alpha}\right)^{\frac{2\alpha}{1+\alpha}}\frac{c_{\alpha,\beta}^2}{2\gamma}\cdot\left(\frac{1+\alpha}{4\alpha}\epsilon_{k+1}\right)^{\frac{2\alpha}{1+\alpha}} \leq \frac{1-\alpha}{1+\alpha}\left(\frac{4\alpha}{1+\alpha}\right)^{\frac{2\alpha}{1-\alpha}}\left(\frac{c_{\alpha,\beta}^2}{2\gamma}\right)^{\frac{1+\alpha}{1-\alpha}} + \frac{1}{2}\epsilon_{k+1}.$$

It then holds that

$$\mathbb{E}_{\mathcal{S},A}\left[F(\mathbf{w}_{k+1}) - F_{\mathcal{S}}(\mathbf{w}_{k+1})\right] \leq \frac{2(1-\alpha)}{1+\alpha}\left(\frac{4\alpha}{1+\alpha}\right)^{\frac{2\alpha}{1-\alpha}}\left(\frac{c_{\alpha,\beta}^2}{2\gamma}\right)^{\frac{1+\alpha}{1-\alpha}} + \frac{c_{\alpha,\beta}^2}{\gamma}\left(\mathbb{E}_{\mathcal{S},A}\big[F_{\mathcal{S}}\left(\mathbf{w}_{k+1}\right)\big]\right)^{\frac{2\alpha}{1+\alpha}}$$

$$+ \frac{4\mathrm{e}\gamma(1+k/n)c_{\alpha,\beta}^2}{n}\sum_{l=1}^{k}\eta_l^2\mathbb{E}_{\mathcal{S},A}\left[F_{\mathcal{S}}^{\frac{2\alpha}{1+\alpha}}\left(\mathbf{w}_{l-\tau_l}\right)\right] + 2\mathrm{e}\gamma\beta^2 r^{2\alpha}\sum_{l=1}^{k}\eta_l\sum_{j=1}^{\tau_l}\eta_{l-j} + \mathrm{e}\gamma d_{\alpha,\beta}^2\sum_{l=1}^{k}\eta_l^{\frac{2}{1-\alpha}}.$$

Substituting (C.7), we can get the following generalization error of ASGD under the $(\alpha, \beta)$-Hölder continuous gradient condition

$$\mathbb{E}_{\mathcal{S},A}\left[F(\mathbf{w}_{k+1}) - F_{\mathcal{S}}(\mathbf{w}_{k+1})\right] \leq \frac{2(1-\alpha)}{1+\alpha}\left(\frac{4\alpha}{1+\alpha}\right)^{\frac{2\alpha}{1-\alpha}}\left(\frac{c_{\alpha,\beta}^2}{2\gamma}\right)^{\frac{1+\alpha}{1-\alpha}} + \frac{c_{\alpha,\beta}^2}{\gamma}\left(\mathbb{E}_{\mathcal{S},A}\big[F_{\mathcal{S}}\left(\mathbf{w}_{k+1}\right)\big]\right)^{\frac{2\alpha}{1+\alpha}}$$

$$+ \frac{8\mathrm{e}\gamma(1+k/n)c_{\alpha,\beta}^2}{n}\left(\sum_{l=1}^{k}\eta_l^2\right)^{\frac{1-\alpha}{1+\alpha}}\left(\eta_1\|\mathbf{w}_1 - \mathbf{w}^*\|^2 + 2\sum_{l=1}^{k}\eta_l^2 F(\mathbf{w}^*) + c_{\alpha,\beta}'\sum_{l=1}^{k}\eta_l^{\frac{3-\alpha}{1-\alpha}} + c_{\alpha,\beta}''\sum_{l=1}^{k}\eta_l^2\right)^{\frac{2\alpha}{1+\alpha}}$$

$$+ 2\mathrm{e}\gamma\beta^2 r^{2\alpha}\sum_{l=1}^{k}\eta_l\sum_{j=1}^{\tau_l}\eta_{l-j} + \mathrm{e}\gamma d_{\alpha,\beta}^2\sum_{l=1}^{k}\eta_l^{\frac{2}{1-\alpha}}.$$

(C.9)

## C.4 Proof of Lemma 6 (Optimization error under the $(\alpha, \beta)$-Hölder continuous gradient assumption)

Leveraging the non-expansive projection property (A.8), convexity and $\alpha, \beta$-Hölder continuous property (A.5), we can derive

$$\begin{aligned}
\|\mathbf{w}_{k+1} - \mathbf{w}^*\|^2 &\leq \|\mathbf{w}_k - \eta_k\nabla f(\mathbf{w}_{k-\tau_k}; \mathbf{z}_{i_k}) - \mathbf{w}^*\|^2 \\
&= \|\mathbf{w}_k - \mathbf{w}^*\|^2 + \eta_k^2\|\nabla f(\mathbf{w}_{k-\tau_k}; \mathbf{z}_{i_k})\|^2 + 2\eta_k\langle\mathbf{w}^* - \mathbf{w}_k, \nabla f(\mathbf{w}_{k-\tau_k}; \mathbf{z}_{i_k})\rangle \\
&\leq \|\mathbf{w}_k - \mathbf{w}^*\|^2 + c_{\alpha,\beta}^2\eta_k^2 f^{\frac{2\alpha}{1+\alpha}}(\mathbf{w}_{k-\tau_k}; \mathbf{z}_{i_k}) + 2\eta_k\langle\mathbf{w}^* - \mathbf{w}_k, \nabla f(\mathbf{w}_k; \mathbf{z}_{i_k})\rangle \\
&\quad + 2\eta_k\langle\mathbf{w}^* - \mathbf{w}_k, \nabla f(\mathbf{w}_{k-\tau_k}; \mathbf{z}_{i_k}) - \nabla f(\mathbf{w}_k; \mathbf{z}_{i_k})\rangle \\
&\leq \|\mathbf{w}_k - \mathbf{w}^*\|^2 + c_{\alpha,\beta}^2\eta_k^2 f^{\frac{2\alpha}{1+\alpha}}(\mathbf{w}_{k-\tau_k}; \mathbf{z}_{i_k}) + 2\eta_k(f(\mathbf{w}^*; \mathbf{z}_{i_k}) - f(\mathbf{w}_k; \mathbf{z}_{i_k})) \\
&\quad + 2\beta\eta_k\|\mathbf{w}^* - \mathbf{w}_k\|\|\mathbf{w}_{k-\tau_k} - \mathbf{w}_k\|^\alpha.
\end{aligned}$$

(C.10)

From the iterative scheme of ASGD (7), (A.8) and the sub-additivity of $x \mapsto x^\alpha$, we know that

$$\|\mathbf{w}_k - \mathbf{w}_{k-\tau_k}\|^\alpha \leq \sum_{j=1}^{\tau_k}\|\mathbf{w}_{k-j+1} - \mathbf{w}_{k-j}\|^\alpha \leq \sum_{j=1}^{\tau_k}\eta_{k-j}^\alpha\|\nabla f(\mathbf{w}_{k-j-\tau_{k-j}}; \mathbf{z}_{i_{k-j}})\|^\alpha. \quad \text{(C.11)}$$

Taking the expectation followed by a summation of (C.10) yields

$$2\sum_{k=1}^{K}\eta_k\mathbb{E}_{\mathcal{S},A}[F_{\mathcal{S}}(\mathbf{w}_k) - F_{\mathcal{S}}(\mathbf{w}^*)] \leq \|\mathbf{w}_1 - \mathbf{w}\|^2 + c_{\alpha,\beta}^2\sum_{k=1}^{K}\eta_k^2\mathbb{E}_{\mathcal{S},A}[F_{\mathcal{S}}^{\frac{2\alpha}{1+\alpha}}(\mathbf{w}_{k-\tau_k})]$$

$$+ 2\beta L^\alpha r\sum_{k=1}^{K}\eta_k\sum_{j=1}^{\tau_k}\eta_{k-j}^\alpha,$$

where we used (C.11), Assumptions 1 and 2. Combing with (C.7), we get the following optimization error bound of ASGD with the $(\alpha, \beta)$-Hölder continuous gradient.

$$
\sum_{k=1}^{K} \eta_k [F_{\mathcal{S}}(\mathbf{w}_k) - F_{\mathcal{S}}(\mathbf{w}^*)] \leq \frac{1}{2}\|\mathbf{w}_1 - \mathbf{w}^*\|^2 + \beta L^\alpha r \sum_{k=1}^{K} \eta_k \sum_{j=1}^{\tau_k} \eta_{k-j}^\alpha
$$
$$
+ c_{\alpha,\beta}^2 \Big( \sum_{k=1}^{K} \eta_k^2 \Big)^{\frac{1-\alpha}{1+\alpha}} \Big( \eta_1 \|\mathbf{w}_1 - \mathbf{w}^*\|^2 + 2\sum_{k=1}^{K} \eta_k^2 F(\mathbf{w}^*) + c_{\alpha,\beta}' \sum_{k=1}^{K} \eta_k^{\frac{3-\alpha}{1-\alpha}} + c_{\alpha,\beta}'' \sum_{k=1}^{K} \eta_k^2 \Big)^{\frac{2\alpha}{1+\alpha}}.
$$
(C.12)

According to the convexity of the function $F_{\mathcal{S}}$, we know that

$$
\mathbb{E}_{\mathcal{S},A}[F_{\mathcal{S}}(\overline{\mathbf{w}}_K) - F_{\mathcal{S}}(\mathbf{w}^*)] \leq \frac{\sum_{k=1}^{K} \eta_k \mathbb{E}_{\mathcal{S},A}[F_{\mathcal{S}}(\mathbf{w}_k) - F_{\mathcal{S}}(\mathbf{w}^*)]}{\sum_{k=1}^{K} \eta_k}
$$
$$
= \mathcal{O}\Big( \frac{\big(\sum_{k=1}^{K} \eta_k^2 \big)^{\frac{1-\alpha}{1+\alpha}}}{\sum_{k=1}^{K} \eta_k} \Big[ \eta_1 \|\mathbf{w}_1 - \mathbf{w}^*\|^2 + \big(1 + F(\mathbf{w}^*)\big) \sum_{k=1}^{K} \eta_k^2 + \sum_{k=1}^{K} \eta_k^{\frac{3-\alpha}{1-\alpha}} \Big]^{\frac{2\alpha}{1+\alpha}}
$$
$$
+ \frac{\|\mathbf{w}_1 - \mathbf{w}^*\|^2 + \sum_{k=1}^{K} \eta_k \sum_{j=1}^{\tau_k} \eta_{k-j}^\alpha}{\sum_{k=1}^{K} \eta_k} \Big).
$$

## C.5 Proof of Theorem 5 (Excess generalization error under the $(\alpha, \beta)$-Hölder continuous gradient assumption)

Multiplying both sides of the generalization error (C.9) by the learning rate $\eta_{k+1}$ followed by summation yields

$$
\sum_{k=1}^{K} \eta_k \mathbb{E}_{\mathcal{S},A}\left[F(\mathbf{w}_k)\right] \leq \sum_{k=1}^{K} \eta_k \mathbb{E}_{\mathcal{S},A}\left[F_{\mathcal{S}}(\mathbf{w}_k)\right] + \frac{c_{\alpha,\beta}^2}{\gamma} \sum_{k=1}^{K} \eta_k \Big( \mathbb{E}_{\mathcal{S},A}\left[F_{\mathcal{S}}(\mathbf{w}_k)\right] \Big)^{\frac{2\alpha}{1+\alpha}}
$$
$$
+ 2\Big(\frac{4\alpha}{1+\alpha}\Big)^{\frac{2\alpha}{1-\alpha}} \Big(\frac{c_{\alpha,\beta}^2}{2\gamma}\Big)^{\frac{1+\alpha}{1-\alpha}} \sum_{k=1}^{K} \eta_k + \frac{8e\gamma(1+K/n)c_{\alpha,\beta}^2}{n} \sum_{k=1}^{K} \eta_k \Big( \sum_{l=1}^{k} \eta_l^2 \Big)^{\frac{1-\alpha}{1+\alpha}}
$$
$$
\cdot \Big( \eta_1 \|\mathbf{w}_1 - \mathbf{w}^*\|^2 + 2\sum_{l=1}^{k} \eta_l^2 F(\mathbf{w}^*) + c_{\alpha,\beta}' \sum_{l=1}^{k} \eta_l^{\frac{3-\alpha}{1-\alpha}} + c_{\alpha,\beta}'' \sum_{l=1}^{k} \eta_l^2 \Big)^{\frac{2\alpha}{1+\alpha}}
$$
$$
+ 2e\gamma\beta^2 r^{2\alpha} \sum_{k=1}^{K} \eta_k \sum_{l=1}^{k} \eta_l \sum_{j=1}^{\tau_l} \eta_{l-j} + e\gamma d_{\alpha,\beta}^2 \sum_{k=1}^{K} \eta_k \sum_{l=1}^{k} \eta_l^{\frac{2}{1-\alpha}}.
$$

According to the concavity of the function $x \mapsto x^{\frac{2\alpha}{1+\alpha}}$, we know that

$$
\sum_{k=1}^{K} \eta_k \Big( \mathbb{E}_{\mathcal{S},A}\left[F_{\mathcal{S}}(\mathbf{w}_k)\right] \Big)^{\frac{2\alpha}{1+\alpha}} \leq \sum_{k=1}^{K} \eta_k \Big( \frac{\sum_{k=1}^{K} \eta_k \mathbb{E}_{\mathcal{S},A}[F_{\mathcal{S}}(\mathbf{w}_k)]}{\sum_{k=1}^{K} \eta_k} \Big)^{\frac{2\alpha}{1+\alpha}}
$$
$$
= \Big( \sum_{k=1}^{K} \eta_k \Big)^{\frac{1-\alpha}{1+\alpha}} \Big( \sum_{k=1}^{K} \eta_k \mathbb{E}_{\mathcal{S},A}[F_{\mathcal{S}}(\mathbf{w}_k)] \Big)^{\frac{2\alpha}{1+\alpha}}
$$
$$
\leq \frac{1-\alpha}{1+\alpha} \sum_{k=1}^{K} \eta_k + \frac{2\alpha}{1+\alpha} \sum_{k=1}^{K} \eta_k \mathbb{E}_{\mathcal{S},A}[F_{\mathcal{S}}(\mathbf{w}_k)],
$$

where the last inequality uses the young's inequality (A.1) with $p = \frac{1+\alpha}{1-\alpha}, q = \frac{1+\alpha}{2\alpha}$. Then

$$\sum_{k=1}^{K} \eta_k \mathbb{E}_{\mathcal{S},A}\left[F(\mathbf{w}_k)\right] \leq \left(1 + \frac{2\alpha c_{\alpha,\beta}^2}{\gamma(1+\alpha)}\right) \sum_{k=1}^{K} \eta_k \mathbb{E}_{\mathcal{S},A}\left[F_{\mathcal{S}}(\mathbf{w}_k)\right] + \left(2\left(\frac{4\alpha}{1+\alpha}\right)^{\frac{2\alpha}{1-\alpha}} \left(\frac{c_{\alpha,\beta}^2}{2\gamma}\right)^{\frac{1+\alpha}{1-\alpha}} + \frac{(1-\alpha)c_{\alpha,\beta}^2}{\gamma(1+\alpha)}\right) \sum_{k=1}^{K} \eta_k$$

$$+ \frac{8e\gamma(1+K/n)c_{\alpha,\beta}^2}{n} \sum_{k=1}^{K} \eta_k \left(\sum_{l=1}^{k} \eta_l^2\right)^{\frac{1-\alpha}{1+\alpha}} \left(\eta_1 \|\mathbf{w}_1 - \mathbf{w}^*\|^2 + 2\sum_{l=1}^{k} \eta_l^2 F(\mathbf{w}^*) + c_{\alpha,\beta}' \sum_{l=1}^{k} \eta_l^{\frac{3-\alpha}{1-\alpha}} + c_{\alpha,\beta}'' \sum_{l=1}^{k} \eta_l^2\right)^{\frac{2\alpha}{1+\alpha}}$$

$$+ 2e\gamma\beta^2 r^{2\alpha} \sum_{k=1}^{K} \eta_k \sum_{l=1}^{k} \eta_l \sum_{j=1}^{\tau_l} \eta_{l-j} + e\gamma d_{\alpha,\beta}^2 \sum_{k=1}^{K} \eta_k \sum_{l=1}^{k} \eta_l^{\frac{2}{1-\alpha}}.$$

From the optimization error (C.12), we then have

$$\sum_{k=1}^{K} \eta_k \mathbb{E}_{\mathcal{S},A}\left[F(\mathbf{w}_k) - F(\mathbf{w}^*)\right] \leq \frac{2\alpha c_{\alpha,\beta}^2}{\gamma(1+\alpha)} \sum_{k=1}^{K} \eta_k F(\mathbf{w}^*) + \left(1 + \frac{2\alpha c_{\alpha,\beta}^2}{\gamma(1+\alpha)}\right) \left(\frac{1}{2} \|\mathbf{w}_1 - \mathbf{w}^*\|^2 + \beta L^\alpha r \sum_{k=1}^{K} \eta_k \sum_{j=1}^{\tau_k} \eta_{k-j}^\alpha\right)$$

$$+ \left(1 + \frac{2\alpha c_{\alpha,\beta}^2}{\gamma(1+\alpha)}\right) c_{\alpha,\beta}^2 \left(\sum_{k=1}^{K} \eta_k^2\right)^{\frac{1-\alpha}{1+\alpha}} \left(\eta_1 \|\mathbf{w}_1 - \mathbf{w}^*\|^2 + 2\sum_{k=1}^{K} \eta_k^2 F(\mathbf{w}^*) + c_{\alpha,\beta}' \sum_{k=1}^{K} \eta_k^{\frac{3-\alpha}{1-\alpha}} + c_{\alpha,\beta}'' \sum_{k=1}^{K} \eta_k^2\right)^{\frac{2\alpha}{1+\alpha}}$$

$$+ \frac{8e\gamma(1+K/n)c_{\alpha,\beta}^2}{n} \sum_{k=1}^{K} \eta_k \left(\sum_{l=1}^{k} \eta_l^2\right)^{\frac{1-\alpha}{1+\alpha}} \left(\eta_1 \|\mathbf{w}_1 - \mathbf{w}^*\|^2 + 2\sum_{l=1}^{k} \eta_l^2 F(\mathbf{w}^*) + c_{\alpha,\beta}' \sum_{l=1}^{k} \eta_l^{\frac{3-\alpha}{1-\alpha}} + c_{\alpha,\beta}'' \sum_{l=1}^{k} \eta_l^2\right)^{\frac{2\alpha}{1+\alpha}}$$

$$+ 2e\gamma\beta^2 r^{2\alpha} \sum_{k=1}^{K} \eta_k \sum_{l=1}^{k} \eta_l \sum_{j=1}^{\tau_l} \eta_{l-j} + e\gamma d_{\alpha,\beta}^2 \sum_{k=1}^{K} \eta_k \sum_{l=1}^{k} \eta_l^{\frac{2}{1-\alpha}} + \left(2\left(\frac{4\alpha}{1+\alpha}\right)^{\frac{2\alpha}{1-\alpha}} \left(\frac{c_{\alpha,\beta}^2}{2\gamma}\right)^{\frac{1+\alpha}{1-\alpha}} + \frac{(1-\alpha)c_{\alpha,\beta}^2}{\gamma(1+\alpha)}\right) \sum_{k=1}^{K} \eta_k.$$

By the convexity of $F$, the excess generalization error of ASGD under the $(\alpha, \beta)$-Hölder continuous gradient satisfies

$$\mathbb{E}_{\mathcal{S},A}\left[F(\overline{\mathbf{w}}_k) - F(\mathbf{w}^*)\right] \leq \frac{\sum_{k=1}^{K} \eta_k \mathbb{E}_{\mathcal{S},A}\left[F(\mathbf{w}_k) - F(\mathbf{w}^*)\right]}{\sum_{k=1}^{K} \eta_k}$$

$$\leq \left(1 + \frac{2\alpha c_{\alpha,\beta}^2}{\gamma(1+\alpha)}\right) c_{\alpha,\beta}^2 \left(\sum_{k=1}^{K} \eta_k^2\right)^{\frac{1-\alpha}{1+\alpha}} \left(\eta_1 \|\mathbf{w}_1 - \mathbf{w}^*\|^2 + 2\sum_{k=1}^{K} \eta_k^2 F(\mathbf{w}^*) + c_{\alpha,\beta}' \sum_{k=1}^{K} \eta_k^{\frac{3-\alpha}{1-\alpha}} + c_{\alpha,\beta}'' \sum_{k=1}^{K} \eta_k^2\right)^{\frac{2\alpha}{1+\alpha}} / \sum_{k=1}^{K} \eta_k$$

$$+ \frac{8e\gamma(1+K/n)c_{\alpha,\beta}^2}{n} \left(\sum_{k=1}^{K} \eta_k^2\right)^{\frac{1-\alpha}{1+\alpha}} \left(\eta_1 \|\mathbf{w}_1 - \mathbf{w}^*\|^2 + 2\sum_{k=1}^{K} \eta_k^2 F(\mathbf{w}^*) + c_{\alpha,\beta}' \sum_{k=1}^{K} \eta_k^{\frac{3-\alpha}{1-\alpha}} + c_{\alpha,\beta}'' \sum_{k=1}^{K} \eta_k^2\right)^{\frac{2\alpha}{1+\alpha}}$$

$$+ \left(2e\gamma\beta^2 r^{2\alpha} \sum_{k=1}^{K} \eta_k \sum_{l=1}^{k} \eta_l \sum_{j=1}^{\tau_l} \eta_{l-j} + e\gamma d_{\alpha,\beta}^2 \sum_{k=1}^{K} \eta_k \sum_{l=1}^{k} \eta_l^{\frac{2}{1-\alpha}}\right) / \sum_{k=1}^{K} \eta_k + 2\left(\frac{4\alpha}{1+\alpha}\right)^{\frac{2\alpha}{1-\alpha}} \left(\frac{c_{\alpha,\beta}^2}{2\gamma}\right)^{\frac{1+\alpha}{1-\alpha}} + \frac{(1-\alpha)c_{\alpha,\beta}^2}{\gamma(1+\alpha)}$$

$$+ \left(1 + \frac{2\alpha c_{\alpha,\beta}^2}{\gamma(1+\alpha)}\right) \left(\frac{1}{2} \|\mathbf{w}_1 - \mathbf{w}^*\|^2 + \beta L^\alpha r \sum_{k=1}^{K} \eta_k \sum_{j=1}^{\tau_k} \eta_{k-j}^\alpha\right) / \sum_{k=1}^{K} \eta_k + \frac{2\alpha c_{\alpha,\beta}^2}{\gamma(1+\alpha)} F(\mathbf{w}^*).$$

$$\text{(C.13)}$$

Let $\gamma > 1$, then we are arrive at

$$\epsilon_{\text{ex-gen}} = \mathcal{O}\Bigg(\left(\sum_{k=1}^{K} \eta_k^2\right)^{\frac{1-\alpha}{1+\alpha}} \left(\eta_1 \|\mathbf{w}_1 - \mathbf{w}^*\|^2 + \sum_{k=1}^{K} \eta_k^2 F(\mathbf{w}^*) + \sum_{k=1}^{K} \eta_k^{\frac{3-\alpha}{1-\alpha}} + \sum_{k=1}^{K} \eta_k^2\right)^{\frac{2\alpha}{1+\alpha}} / \sum_{k=1}^{K} \eta_k$$

$$+ \frac{\gamma(1+K/n)}{n} \left(\sum_{k=1}^{K} \eta_k^2\right)^{\frac{1-\alpha}{1+\alpha}} \left(\eta_1 \|\mathbf{w}_1 - \mathbf{w}^*\|^2 + \sum_{k=1}^{K} \eta_k^2 F(\mathbf{w}^*) + \sum_{l=1}^{K} \eta_l^{\frac{3-\alpha}{1-\alpha}} + \sum_{k=1}^{K} \eta_k^2\right)^{\frac{2\alpha}{1+\alpha}}$$

$$+ \gamma \left(\sum_{k=1}^{K} \eta_k \sum_{l=1}^{k} \eta_l \sum_{j=1}^{\tau_l} \eta_{l-j} + \sum_{k=1}^{K} \eta_k \sum_{l=1}^{k} \eta_l^{\frac{2}{1-\alpha}}\right) / \sum_{k=1}^{K} \eta_k + \gamma^{\frac{1+\alpha}{\alpha-1}}$$

$$+ \left(\|\mathbf{w}_1 - \mathbf{w}^*\|^2 + \sum_{k=1}^{K} \eta_k \sum_{j=1}^{\tau_k} \eta_{k-j}^\alpha\right) / \sum_{k=1}^{K} \eta_k + \frac{1}{\gamma}\left(1 + F(\mathbf{w}^*)\right)\Bigg).$$

$$\text{(C.14)}$$

### C.5.1 Excess Generalization Error with the Learning Rate $\eta_k = c(\bar{\tau}\sqrt{K})^{-1}$

Set the learning rate $\eta_k = c(\bar{\tau}\sqrt{K})^{-1}$ with $c>0$, and $\bar{\tau} = \sum_{k=1}^{K} \tau_k/K$, direct calculation gives

$$\sum_{k=1}^{K} \eta_k = c\sqrt{K}/\bar{\tau}, \quad \sum_{k=1}^{K} \eta_k^2 = c^2/\bar{\tau}^2, \quad \sum_{k=1}^{K} \eta_k \sum_{l=1}^{k} \eta_l \sum_{j=1}^{\tau_l} \eta_{l-j} = \left(\frac{c}{\bar{\tau}\sqrt{K}}\right)^3 \sum_{k=1}^{K} \sum_{l=1}^{k} \tau_l \le \frac{c^3\sqrt{K}}{\bar{\tau}^2}$$

$$\sum_{k=1}^{K} \eta_k \sum_{j=1}^{\tau_k} \eta_{k-j}^\alpha = \frac{c^{1+\alpha} K^{\frac{1-\alpha}{2}}}{\bar{\tau}^\alpha}, \quad \sum_{k=1}^{K} \eta_k^{\frac{3-\alpha}{1-\alpha}} \asymp K^{1-\frac{3-\alpha}{2(1-\alpha)}} \bar{\tau}^{-\frac{3-\alpha}{1-\alpha}}, \quad \sum_{k=1}^{K} \eta_k \sum_{l=1}^{k} \eta_l^{\frac{2}{1-\alpha}} \asymp K^{2-\frac{3-\alpha}{2(1-\alpha)}} \bar{\tau}^{-\frac{3-\alpha}{1-\alpha}}.$$

Following the excess generalization error (C.14), we know that

$$\epsilon_{\text{ex-gen}} = \mathcal{O}\Bigg(\left(\frac{1}{\bar{\tau}^2}\right)^{\frac{1-\alpha}{1+\alpha}} \left(\frac{\|\mathbf{w}_1 - \mathbf{w}^*\|^2}{\sqrt{K}\bar{\tau}} + \frac{1}{\bar{\tau}^2}\left(1 + F(\mathbf{w}^*)\right) + \frac{K^{1-\frac{3-\alpha}{2(1-\alpha)}}}{\bar{\tau}^{\frac{3-\alpha}{1-\alpha}}}\right)^{\frac{2\alpha}{1+\alpha}} \cdot \frac{\bar{\tau}}{\sqrt{K}}$$

$$+ \frac{\gamma(1 + K/n)}{n} \left(\frac{1}{\bar{\tau}^2}\right)^{\frac{1-\alpha}{1+\alpha}} \left(\frac{\|\mathbf{w}_1 - \mathbf{w}^*\|^2}{\sqrt{K}\bar{\tau}} + \frac{1}{\bar{\tau}^2}\left(1 + F(\mathbf{w}^*)\right) + \frac{K^{1-\frac{3-\alpha}{2(1-\alpha)}}}{\bar{\tau}^{\frac{3-\alpha}{1-\alpha}}}\right)^{\frac{2\alpha}{1+\alpha}}$$

$$+ \gamma\left(\frac{\sqrt{K}}{\bar{\tau}^2} + \frac{K^{2-\frac{3-\alpha}{2(1-\alpha)}}}{\bar{\tau}^{\frac{3-\alpha}{1-\alpha}}}\right) \cdot \frac{\bar{\tau}}{\sqrt{K}} + \gamma^{\frac{1+\alpha}{\alpha-1}} + \left(\|\mathbf{w}_1 - \mathbf{w}^*\|^2 + \frac{K^{\frac{1-\alpha}{2}}}{\bar{\tau}^\alpha}\right) \cdot \frac{\bar{\tau}}{\sqrt{K}} + \frac{1}{\gamma}\left(1 + F(\mathbf{w}^*)\right)\Bigg).$$

By the sub-additivity of $x \mapsto x^{\frac{2\alpha}{1+\alpha}}$, $(\alpha \in [0,1])$, we can derive

$$\epsilon_{\text{ex-gen}} = \mathcal{O}\Bigg(\left[\frac{\bar{\tau}}{\sqrt{K}} + \frac{\gamma(1 + K/n)}{n}\right]\bar{\tau}^{-\frac{2(1-\alpha)}{1+\alpha}} \left(\frac{\|\mathbf{w}_1 - \mathbf{w}^*\|^{\frac{4\alpha}{1+\alpha}}}{K^{\frac{\alpha}{1+\alpha}}\bar{\tau}^{\frac{2\alpha}{1+\alpha}}} + \frac{1}{\bar{\tau}^{\frac{4\alpha}{1+\alpha}}}\left(1 + F^{\frac{2\alpha}{1+\alpha}}(\mathbf{w}^*)\right) + \frac{K^{-\frac{\alpha}{1-\alpha}}}{\bar{\tau}^{\frac{2\alpha(3-\alpha)}{(1+\alpha)(1-\alpha)}}}\right)$$

$$+ \gamma\left(\frac{1}{\bar{\tau}} + \frac{K^{-\frac{\alpha}{1-\alpha}}}{\bar{\tau}^{\frac{2}{1-\alpha}}}\right) + \frac{\bar{\tau}\|\mathbf{w}_1 - \mathbf{w}^*\|^2}{\sqrt{K}} + \frac{\bar{\tau}^{1-\alpha}}{K^{\frac{\alpha}{2}}} + \gamma^{\frac{1+\alpha}{\alpha-1}} + \frac{1}{\gamma}\left(1 + F(\mathbf{w}^*)\right)\Bigg)$$

$$= \mathcal{O}\Bigg(\left[\frac{\bar{\tau}}{\sqrt{K}} + \frac{\gamma(1 + K/n)}{n}\right]\left(\frac{K^{-\frac{\alpha}{1+\alpha}}}{\bar{\tau}^{\frac{2}{1+\alpha}}}\|\mathbf{w}_1 - \mathbf{w}^*\|^{\frac{4\alpha}{1+\alpha}} + \frac{1}{\bar{\tau}^2}\left(1 + F^{\frac{2\alpha}{1+\alpha}}(\mathbf{w}^*)\right) + \frac{K^{-\frac{\alpha}{1-\alpha}}}{\bar{\tau}^{\frac{2}{1-\alpha}}}\right)$$

$$+ \gamma\left(\frac{1}{\bar{\tau}} + \frac{K^{-\frac{\alpha}{1-\alpha}}}{\bar{\tau}^{\frac{2}{1-\alpha}}}\right) + \frac{\bar{\tau}}{\sqrt{K}} + \frac{\bar{\tau}^{1-\alpha}}{K^{\frac{\alpha}{2}}} + \gamma^{\frac{1+\alpha}{\alpha-1}} + \frac{1}{\gamma}\left(1 + F(\mathbf{w}^*)\right)\Bigg).$$

Omitting the non-dominant term gives (with $\gamma > 1$)

$$\epsilon_{\text{ex-gen}} = \mathcal{O}\Bigg(\left[\frac{\bar{\tau}}{\sqrt{K}} + \frac{\gamma(1 + K/n)}{n}\right]\left(\frac{K^{-\frac{\alpha}{1+\alpha}}}{\bar{\tau}^{\frac{2}{1+\alpha}}}\|\mathbf{w}_1 - \mathbf{w}^*\|^{\frac{4\alpha}{1+\alpha}} + \frac{1}{\bar{\tau}^2}\left(1 + F^{\frac{2\alpha}{1+\alpha}}(\mathbf{w}^*)\right)\right)$$

$$+ \frac{\gamma}{\bar{\tau}} + \frac{\bar{\tau}}{\sqrt{K}} + \frac{\bar{\tau}^{1-\alpha}}{K^{\frac{\alpha}{2}}} + \frac{1}{\gamma}\left(1 + F(\mathbf{w}^*)\right)\Bigg).$$

If $F(\mathbf{w}^*) = 0$, $K \asymp n$, then the excess generalization error is

$$\epsilon_{\text{ex-gen}} = \mathcal{O}\Bigg(\left(\frac{\bar{\tau}}{\sqrt{K}} + \frac{\gamma}{n}\right)\left(\frac{K^{-\frac{\alpha}{1+\alpha}}}{\bar{\tau}^{\frac{2}{1+\alpha}}}\|\mathbf{w}_1 - \mathbf{w}^*\|^{\frac{4\alpha}{1+\alpha}} + \frac{1}{\bar{\tau}^2}\right) + \frac{\gamma}{\bar{\tau}} + \frac{\bar{\tau}}{\sqrt{K}} + \frac{\bar{\tau}^{1-\alpha}}{K^{\frac{\alpha}{2}}} + \frac{1}{\gamma}\Bigg).$$

Let $\gamma = \sqrt{\bar{\tau}}$, we have that $\max\{\frac{\bar{\tau}}{\sqrt{K}}, \frac{\gamma}{n}\} = \frac{\bar{\tau}}{\sqrt{K}}$, and then

$$\epsilon_{\text{ex-gen}} = \mathcal{O}\Bigg(\frac{\|\mathbf{w}_1 - \mathbf{w}^*\|^{\frac{4\alpha}{1+\alpha}}}{K^{\frac{1+3\alpha}{2(1+\alpha)}}\bar{\tau}^{\frac{1-\alpha}{1+\alpha}}} + \frac{1}{\sqrt{K}\bar{\tau}} + \frac{1}{\sqrt{\bar{\tau}}} + \frac{\bar{\tau}}{\sqrt{K}} + \frac{\bar{\tau}^{1-\alpha}}{K^{\frac{\alpha}{2}}}\Bigg).$$

Furthermore, if the average delay satisfies $\bar{\tau} \le K^{\alpha'}$ with $\alpha' = \min\{\frac{1}{3}, \frac{\alpha}{3-2\alpha}\}$, we know that

$$\max\left\{\frac{1}{\sqrt{K}\bar{\tau}}, \frac{1}{\sqrt{\bar{\tau}}}, \frac{\bar{\tau}}{\sqrt{K}}, \frac{\bar{\tau}^{1-\alpha}}{K^{\frac{\alpha}{2}}}\right\} = \frac{1}{\sqrt{\bar{\tau}}}.$$

On the other hand, since $\bar{\tau} \ge 1$ in the asynchronous training and $\alpha \in [0,1]$, we have

$$K^{\frac{1+3\alpha}{2(1+\alpha)}}\bar{\tau}^{\frac{1-\alpha}{1+\alpha}} \ge K^{\frac{1+3\alpha}{2(1+\alpha)}} \ge K^{\frac{1+\alpha}{2}} \asymp n^{\frac{1+\alpha}{2}}.$$

Then, we are arrive at

$$\epsilon_{\text{ex-gen}} = \mathbb{E}_{\mathcal{S},A}\left[F(\overline{\mathbf{w}}_k) - F(\mathbf{w}^*)\right] = \mathcal{O}\Bigg(\frac{1}{\sqrt{\bar{\tau}}} + \frac{\|\mathbf{w}_1 - \mathbf{w}^*\|^{\frac{4\alpha}{1+\alpha}}}{\sqrt{n}^{1+\alpha}}\Bigg).$$

The proof is complete.

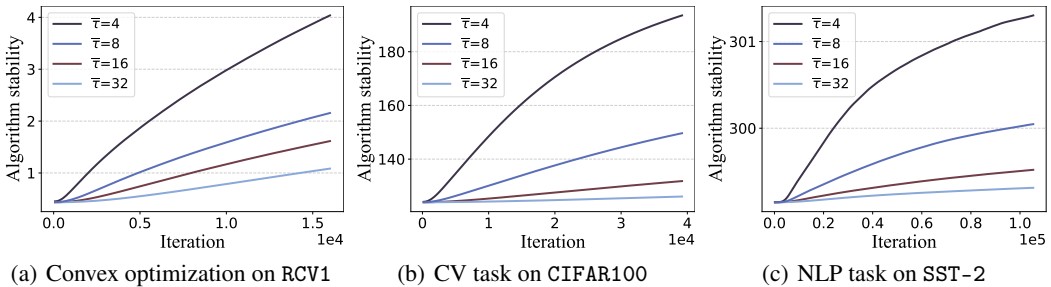

(a) Convex optimization on `RCV1`    (b) CV task on `CIFAR100`    (c) NLP task on `SST-2`

Figure 2: The on-average model stability in training various machine learning tasks using ASGD with learning rate $\eta_k = 0.1/\overline{\tau}$. The horizontal axis denotes the number of asynchronous training iterations, and the legend represents the average delay. A degradation in algorithm stability is observed as the number of training iterations increases.

## D    More Experiment Results

This section provides additional details on the experimental setup, as well as the stability and generalization results trained with a delay-independent constant learning rate. Our experiments use the more general asynchronous stochastic gradient descent format (9), i.e.,

$$\mathbf{w}_{k+1} = \mathbf{w}_k - \eta_k \sum_{m \in \mathcal{M}_k} \mathbf{g}_{k-\tau_k}^m.$$

In practical applications, the gradient $\mathbf{g}_{k-\tau_k}^m$ is evaluated on a mini-batch of the training data. The batch size of each worker in this experiment was set to $16$. It should be noted that we only simulated $8$ workers in the BERT experiments due to memory limitations. All of our experiments were implemented with PyTorch on Nvidia RTX-3090 24 GB GPUs.

In Figure 3 and 4, the stability and generalization results of the ASGD algorithm, employing a delay-independent learning rate of $\eta_k = 0.01$, are illustrated. In this scenario, increasing the delay still improves the algorithm stability and reduces the generalization error, indicating that asynchronous training is indeed beneficial for generalization.

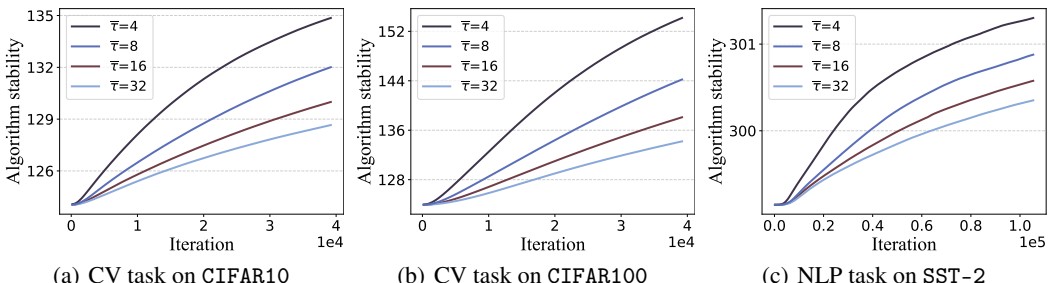

(a) CV task on `CIFAR10`    (b) CV task on `CIFAR100`    (c) NLP task on `SST-2`

Figure 3: The on-average model stability in training various machine learning tasks using ASGD with delay-independent constant learning rate $\eta_k = 0.01$. The horizontal axis denotes the number of asynchronous training iterations, and the legend represents the average delay. A degradation in algorithm stability is observed with an increase in training iterations.

Figure 5 shows the training, testing and generalization errors of three categories of machine learning models. The generalization errors are roughly of the same order of magnitude as the training and testing errors. In certain model tasks, particularly BERT on the `SST-2` task, overfitting phenomena are present, contributing significantly to the generalization gap. Therefore, we need to complete the training process as soon as possible to improve the model generalization performance, which is consistent with our theoretical analysis in Section 4.2.

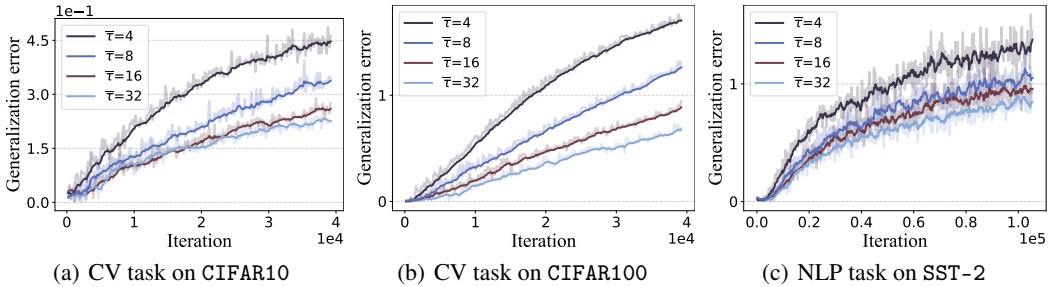

Figure 4: The generalization errors in training various machine learning tasks using ASGD with delay-independent constant learning rate $\eta_k = 0.01$. The trend of generalizability with the number of iterations is analogous to the algorithm stability depicted in Figure 3, and appropriately increasing the asynchronous delay can enhance generalization performance.

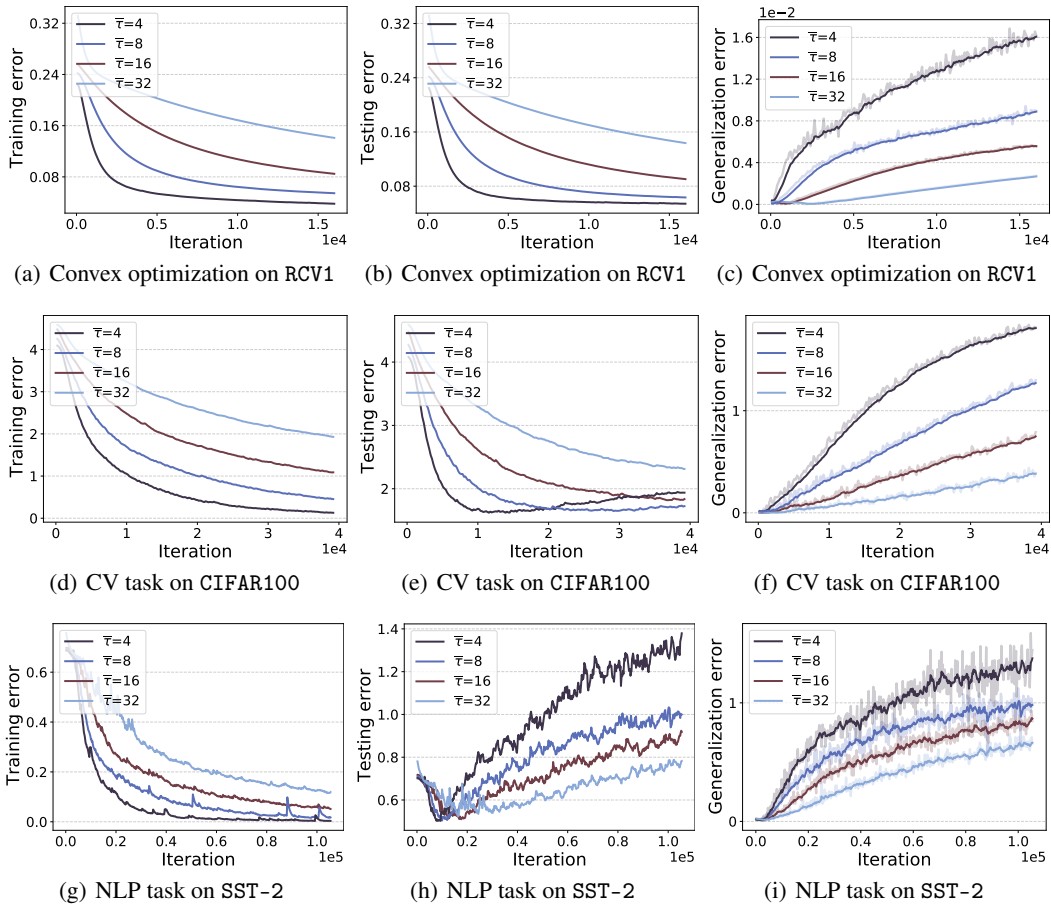

Figure 5: The training, testing and generalization errors of three categories of machine learning models trained using ASGD with learning rate $\eta_k = 0.1/\overline{\tau}$. The horizontal axis denotes the number of asynchronous training iterations, and the legend represents the average delay.

# E  Contributions and Limitations

## E.1  Contributions

The main challenge of this study is to establish sharper generalization bounds for the ASGD algorithm under much weaker assumptions.

Notably, the existing studies on the generalization of ASGD are limited. Study [33] provides only vacuous exponential generalization bounds and relies on strict assumptions such as Lipschitz continuous and smooth functions. Another work [13] establishes tighter generalization bounds, but its analytical techniques only applicable to smooth quadratic convex problems.

Our contributions have been detailed in Section 1, and the following table further compares the required assumptions and theoretical results of the related works.

Table 1: Comparison with related work.

|  | Regatti et al. [33] | Deng et al. [13] | Ours |
|---|---|---|---|
| Lipschitz assumption? | $L$-Lipschitz | Not required | Not required |
| Smoothness assumption? | $\beta$-smooth | $\beta$-smooth | $(\alpha, \beta)$-Hölder continuous |
| Convexity? | Non-convex | Quadratic convex | General convex |
| Generalization error | $\mathcal{O}\big(\frac{K^{\hat{\tau}}}{n^{\hat{\tau}}}\big)$ | $\widetilde{\mathcal{O}}\big(\frac{K-\hat{\tau}}{n^{\hat{\tau}}}\big)$ | $\mathcal{O}\big(\frac{1}{\hat{\tau}} + \frac{1}{\sqrt{K}}\big)$ |
| Excess generalization error | N/A | N/A | $\mathcal{O}\big(\frac{1}{\sqrt{\hat{\tau}}} + \frac{\|\mathbf{w}_1 - \mathbf{w}^*\|^{\frac{4\alpha}{1+\alpha}}}{\sqrt{n}^{1+\alpha}}\big)$ |

## E.2  Limitations

**Assumption.** In Assumptions 1 and 2, we have listed the assumptions required for this paper and explained their roles and plausibility. It is crucial to note that this study aims to establish sharper stability and generalization error bounds under much weaker assumptions. If we adopt stronger assumptions, such as the assumption in paper [55] that the difference between models $\mathbf{w}_k$ and $\mathbf{w}_k^{(i)}$ follows a normal distribution with bounded mean and variance, we can obtain better results (in terms of the training sample size $n$).

**Pessimistic result.** The experiments in Appendix D, concerning delay-independent fixed learning rates, show that the generalization error bound (B.10) is pessimistic, i.e., asynchronous training is beneficial for generalization even if a fixed learning rate is used. A potential avenue for future research lies in exploring tighter high probability bounds that attenuate the dominant role of the learning rate on generalization, thereby elucidating the experimental phenomena in Appendix D.

**Non-convex study.** In the non-convex setting, the delayed gradient update operator cannot maintain the approximately non-expansive property. Consequently, directly extending the analysis of this paper to non-convex scenarios would yield an exponential generalization error bound, similar to the findings in study [33]. Unfortunately, this upper bound is pessimistic and vacuous. Exploring sharper stability and generalization error bounds of ASGD in non-convex scenarios is extremely challenging. Future research on non-convex problems could focus on demonstrating that asynchronous gradient updates are approximately non-expansive even without the convexity property, then leading to non-vacuous stability and generalization results.

Additionally, while our theoretical analysis is grounded in the general convex condition, our non-convex experiments show that the theoretical results in this paper are applicable in a broader range of non-convex machine learning tasks (particularly deep learning), which motivates us to further explore tighter stability and generalization results for ASGD in the non-convex scenarios in the future.

