# OpenReview forum: "Stability and Generalization of Asynchronous SGD: Sharper Bounds Beyond Lipschitz and Smoothness"
_NeurIPS.cc/2024/Conference — NeurIPS 2024 poster_

### Official Review · Reviewer_yEhD · 2024-07-05

**Soundness:** 3
**Presentation:** 3
**Contribution:** 3
**Rating:** 5
**Confidence:** 4

**Summary:**

This paper talks about the generalization error and the excess generalization error of Asynchronous SGD, mainly under convex, smooth or holder continuous conditions.

**Strengths:**

The theorems are sound, and the results are new.

**Weaknesses:**

In th2, the generalization error is written as the form of containing w1 and wk. This is some what not reasonable. I want to see the generalization error which is depended on the already setting constants like η，K，n and so on, in this way, theorem can be applied to more situations.

In cor1, cor2 and th5, if we want the generalization bound to be small, then τk should be big. This is very strange, why is training with higher latency-update better, but poor with lower latency-update, this seems to contradict intuition and the real world?

I see your paper and experiment is talking about networks. I have some concerns about the connection between this article and neural networks.

1: May I ask if your theorems can guide the training of networks, or give some helps to training?

2: The f(w,z) is convex about w is a very strong conditions when considering network, if we see w as the parameters of network F, at this point, the f(w,z) is hard to be convex about w. I understand that ‘convex‘ is an important condition, and with this condition, many mathematical tools can be used， but for adapting to real-world-network-situation, is it possible for this condition to be more lenient?

**Questions:**

Can we take τk=0 in this paper? At in this situation, can the result lead to the generalization bound of SGD?

In section 5, author replace smooth by holder continuous.  I don't quite understand the necessity of doing this. Can this be helpful in some practical problems or can it provides new mathematical tools?

**Limitations:**

The authors have adequately addressed the limitations and potential negative societal impact of their work

---

> ### Author Rebuttal · Authors · 2024-08-05
>
> > **Question 1.** In th2, the generalization error is written as the form of containing $\mathbf{w}_1$ and $\mathbf{w}_K$. This is some what not reasonable. I want to see the generalization error which is depended on the already setting constants like $\eta, K, n$ and so on, in this way, theorem can be applied to more situations.
>
> **Response.** Theorem 2 provides a generalization error result by directly substituting the stability of the ASGD algorithm into Lemma 1. The inclusion of $\mathbf{w}_1$ and $\mathbf{w}_K$ aims to demonstrate the impact of the optimization process of the algorithm ($F\_{\mathcal{S}}(\mathbf{w}\_{K})$) and the initial state of the model ($\mathbf{w}\_1$) on the algorithm's generalization performance.
>
> In the subsequent Corollary 1, Theorem 3, and Corollary 2, we further analyzed the optimization error, and these results no longer contain $\mathbf{w}\_K$, but instead reveal the effects of asynchronous delay, model initialization, number of training samples and iterations on generalization performance.
>
> ---
>
> > **Question 2.** In cor1, cor2 and th5, if we want the generalization bound to be small, then $\tau_k$ should be big. This is very strange, why is training with higher latency-update better, but poor with lower latency-update, this seems to contradict intuition and the real world?
>
> **Response.** Yes, this is the key finding of this study, i.e., increasing the asynchronous delay appropriately in asynchronous training improves the stability of the ASGD algorithm and reduces the generalization error.
>
> In this paper, we study the generalization performance rather than the convergence of ASGD, and our theoretical results (Corollaries 1 and 2, Theorem 5) show that the noise introduced by asynchronous training makes the algorithm more robust at the proper learning rate, and thus improves the generalization performance.
>
> A similar bound $\widetilde{\mathcal{O}}(\frac{K-\hat{\tau}}{n\hat{\tau}})$ for quadratic optimization problems is established in study [13]. This paper extends these findings to general convex problems under the weaker $(\alpha, \beta)$-Hölder continuous gradient assumption. Additionally, we have also conducted a number of real-world experiments (Section 6), which further corroborate that appropriately increasing the asynchronous delay improves algorithmic stability and thus reduces the generalization error (see Figures 1, 2, 3 and 4).
>
> ---
>
> > **Question 3.** I have some concerns about the connection between this article and neural networks. May I ask if your theorems can guide the training of networks, or give some helps to training?
>
> **Response.** Yes, our theoretical results reveal the effects of asynchronous delay, model initialization, number of training samples and iterations on generalization performance. Moreover, we have discussed in the paper how these theoretical generalization results can provide guidance for practical training, e.g.,
>
> * **Lines 208-209, below Theorem 1**
>
>   *Theorem 1 indicates that model initialization affects the algorithmic stability, i.e., selecting a better model initiation point $\mathbf{w}_{1}$ can effectively improve the stability.*
>
> * **Lines 216-220, below Theorem 2**
>
>   *This finding suggests that both the model initialization and optimization processes have an impact on the generalization performance. In practical applications, one can reduce the generalization error by selecting a good initial model $\mathbf{w}_{1}$ to start the training task. Additionally, it is crucial to finish the optimization process promptly since too many training iterations can detrimentally affect the generalization performance.*
>
> * **Lines 308-310, below Theorem 5**
>
>   *That is, the generalization performance can be improved by choosing a good initial model, increasing the number of training samples, and appropriately adjusting the asynchronous delays.*
>
> ---
>
> > **Question 4.** I understand that "convex" is an important condition, and with this condition, many mathematical tools can be used, but for adapting to real-world-network-situation, is it possible for this condition to be more lenient?
>
> **Response.** The existing non-vacuous generalization error results for ASGD [13, 41] are only applicable to quadratic convex problems, and the study [41] does not reveal the effect of asynchronous delays on the algorithm's generalization performance.
>
> In this paper, we establish the non-vacuous generalization error and excess generalization error results of ASGD under weaker assumptions for the general convex case. Furthermore, we have discussed the limitations of the convex function assumption and explored the challenges and potential research directions for non-convex optimization in Appendix E.2 (lines 792-803).
>
> ---
>
> > **Question 5.** Can we take $\tau_k=0$ in this paper? At in this situation, can the result lead to the generalization bound of SGD?
>
> **Response.** Yes. Please refer to the response to Question 1 in **Author Rebuttal**.
>
> ---
>
> > **Question 6.** In section 5, author replace smooth by holder continuous. I don't quite understand the necessity of doing this. Can this be helpful in some practical problems or can it provides new mathematical tools?
>
> **Response.** In the Preliminaries section of the paper, we have described the motivation for doing this, i.e.,
>
> * **Section 3, Lines 152-156**
>
>   *While the smooth function assumption is common in optimization and generalization analyses [13, 17, 33, 40], it does impose constraints on the applicability [5]. For instance, the hinge loss, which is widely used in the ML fields, does not satisfy the smooth property. In this paper, therefore, we also investigate the stability of ASGD under the much weaker Hölder continuous gradient assumption (Definition 3), so as to establish broader and fine-grained generalization results.*
>
> In short, the Hölder continuous gradient is a weaker assumption than the smooth function and can be applied to a wider range of machine learning tasks.

---

> > ### Author Response · Authors · 2024-08-12
> >
> > Dear Reviewer yEhD,
> >
> >
> >
> > We sincerely appreciate your valuable comments and eagerly await your assessment of whether our responses have sufficiently addressed your concerns. Please do not hesitate to contact us if you require any further clarification.
> >
> > Additionally, if our response has successfully addressed your concerns, we kindly request that you re-evaluate our paper and reconsider the score. Thank you very much for dedicating your time and effort to our submission.
> >
> > We earnestly look forward to receiving your response.
> >
> >
> >
> > Best regards,
> >
> > Submission2646 Authors

---

> ### Comment · Reviewer_yEhD · 2024-08-12
>
> The rebuttal addresses my concerns and I will raise my score.

---

> > ### Author Response · Authors · 2024-08-13
> >
> > Thank you for your time and acknowledgment of our submission. You are welcome to continue the discussion if you have any further questions.

---

### Official Review · Reviewer_pPuN · 2024-07-12

**Soundness:** 2
**Presentation:** 3
**Contribution:** 3
**Rating:** 5
**Confidence:** 5

**Summary:**

For distributed machine learning tasks, Asynchronous Stochastic Gradient Descent (ASGD) is an indispensable optimization algorithm. Considering the existing results that fail to reveal the intrinsic impact of asynchronous training, this paper establishes sharper stability and generalization bounds for ASGD with convex loss function under much weaker assumptions, i.e., without Lipschitz assumption or smoothness. Furthermore, excess generalization error is investigated after deriving optimization upper bound. Several experiments validate these theoretical results.

**Strengths:**

+ The manuscript develops the stability-based generalization analysis of ASGD to weaker cases, i.e., the case without either Lipschitz condition or smoothness.

+ Through numerical experiments including both convex and non-convex optimization problems, authors validate their theoretical results.

**Weaknesses:**

- The authors provide the approximately non-expansive recursive property for ASGD. Without some limitations to $r$ and $\tau$, the terms $2\eta_k\beta^2r^2\sum\limits_{j=1}^{\tau_k}\eta_{k-j}$ in Lemma 3 and $\mathcal{O}\left(\eta_k\sum\limits_{j=1}^{\tau_k}\eta_{k-j}+\eta_{k}^{\frac{2}{1-\alpha}}\right)$ in Lemma 5 may be very large.
- There are some terms related to learning rates in almost all results, such as $\sum\limits_{k=1}^K\eta_k\sum\limits_{j=1}^{\tau_k}\eta_{k-j}$ in Theorem 2, $\sum\limits_{k=1}^K\tau_k$ in Corollary and $\sum\limits_{l=1}^k\eta_l\sum\limits_{j=1}^{\tau_l}\eta_{l-j}$ in Theorem 4. Therefore, these results may not converge but in the case with stringent learning rate conditions like Corollary 2.
- In the remarks of the key results, authors rarely compare their results with previous work, which prevents readers from understanding the advantages of the article sufficiently.

**Questions:**

1.Should the learning rate upper bound $\eta_k \leq 1/2\beta$ be $\frac{1}{2\beta}$?

2.The experiments authors conduct include non-convex tasks but their theoretical analysis doesn’t. In Concluding Remarks, the study of non-convex problems is one of the directions for future research. What is the aim of these additional experiments?

**Limitations:**

From Preliminaries and Algorithms, it seems that authors just consider the gradient update in a local worker but the server. Could the analysis of this paper be applied to the study of the server like Lian et al,. 2018, Deng et al,. 2023 and Chen et al., 2023?

- X. Lian, W. Zhang, C. Zhang, and J. Liu. Asynchronous decentralized parallel stochastic gradient descent. ICML, 2018.

- X. Deng, T. Sun, S. Li, and D. Li. Stability-based generalization analysis of the asynchronous decentralized SGD. AAAI, 2023

- J. Chen, H. Chen, B. Gu, H. Deng. Fine-Grained Theoretical Analysis of Federated Zeroth-Order Optimization. NeurIPS, 2023.

---

> ### Author Rebuttal · Authors · 2024-08-05
>
> > **Question 1.** The authors provide the approximately non-expansive recursive property for ASGD. Without some limitations to $r$ and $\tau$, the terms $2\eta_{k}\beta^{2}r^{2}\sum_{j=1}^{\tau_{k}}\eta_{k-j}$ in Lemma 3 and $\mathcal{O}\Big(\eta_{k}\sum_{j=1}^{\tau_{k}}\eta_{k-j}+\eta_{k}^{\frac{2}{1-\alpha}}\Big)$ in Lemma 5 may be very large.
>
> **Response.** In this paper, we study the projected ASGD algorithm (Lines 178-183), i.e., $\\mathbf{w}\_{k+1}=\Pi\_{\Omega}\big(\mathbf{w}\_{k}-\eta\_{k}\nabla f(\mathbf{w}\_{k-\tau\_{k}}; \mathbf{z}\_{i\_{k}})\big)$. Thus, we can effectively control the radius $r$ of the parameter space by projection operations. In addition, $\overline{\tau}$ is defined as $\overline{\tau}=\sum_{k=1}^{K}\tau_{k}/K$, denoting the average delay, which is also bounded.
>
> Furthermore, these two terms are closely related to the learning rate. Similar to the previous work [40], we can control these two terms by utilizing a learning rate inversely proportional to $r$ and $\overline{\tau}$. Specifically, the learning rate for asynchronous training is set to $\eta_{k}=c/(\overline{\tau}\sqrt{K})$, where $c$ is a constant inversely proportional to $r$ and $\beta$. In this case, these two terms correspond to $\mathcal{O}(1/\overline{\tau})$ and $\mathcal{O}(1/\sqrt{\overline{\tau}})$ in Corollary 2 and Theorem 5, respectively, which is significantly better than the existing exponential upper bound $\mathcal{O}(K^{\hat{\tau}}/n\hat{\tau})$.
>
> ---
>
> > **Question 2.** There are some terms related to learning rates in almost all results, such as $\sum_{k=1}^{K}\eta_{k}\sum_{j=1}^{\tau_{k}}\eta_{k-j}$ in Theorem 2, $\sum_{k=1}^{K}\tau_{k}$ in Corollary and $\sum_{l=1}^{k}\eta_{l}\sum_{j=1}^{\tau_{l}}\eta_{l-j}$ in Theorem 4. Therefore, these results may not converge but in the case with stringent learning rate conditions like Corollary 2.
>
> **Response.** First of all, it should be clarified that the **generalization error does not converge to zero** in general, and our experimental results can verify this argument.
>
> Moreover, the stability and generalization of an algorithm are inherently related to its learning rate, a phenomenon observed in SGD as well [17] (whose result is $\frac{2L^{2}}{n}\sum_{k=1}^{K}\eta_k$). Researchers typically employ a learning rate that is inversely correlated with the number of training iterations $K$ and the asynchronous delay $\tau$ to obtain the corresponding theoretical results [13, 17, 33, 40].
>
> ---
>
> > **Question 3.** In the remarks of the key results, authors rarely compare their results with previous work, which prevents readers from understanding the advantages of the article.
>
> **Response.** Please refer to the response to Question 2 in **Author Rebuttal**.
>
> ---
>
> > **Question 4.** Should the learning rate upper bound $\eta_k\leq1/2\beta$ be $\frac{1}{2\beta}$?
>
> **Response.** In the proof of Theorem 1, we need the learning rate to satisfy $\eta_{k}<1/2\beta$, a slightly different condition from that in Lemma 3, $\eta_{k}<2/\beta$. This condition is standard in analyzing SGD and its variants [13, 17, 23, 30, 33].
>
> ---
>
> > **Question 5.** The experiments authors conduct include non-convex tasks but their theoretical analysis doesn’t. In Concluding Remarks, the study of non-convex problems is one of the directions for future research. What is the aim of these additional experiments?
>
> **Response.** While our theoretical analysis is grounded in the general convex condition, additional non-convex experiments show that the theoretical results in this paper are applicable in a broader range of non-convex machine learning tasks (particularly deep learning), which motivates us to further explore tighter stability and generalization results of the ASGD algorithm in the non-convex scenarios in the future.
>
> Furthermore, we have also discussed the limitations of the convex function assumption and explored the difficulties and potential research directions for non-convex optimization in Appendix E.2, i.e.,
>
> * **Appendix E.2, Lines 792-799**
>
>   *In the non-convex setting, the delayed gradient update operator cannot maintain the approximately non-expansive property. Consequently, directly extending the analysis of this paper to non-convex scenarios would yield an exponential generalization error bound, similar to the findings in study [33]. Unfortunately, this upper bound is pessimistic and vacuous. Exploring sharper stability and generalization error bounds of ASGD in non-convex scenarios is extremely challenging. Future research on non-convex problems could focus on demonstrating that asynchronous gradient updates are approximately non-expansive even without the convexity property, then leading to non-vacuous stability and generalization results.*
>
> ---
>
> > **Question 6.** From Preliminaries and Algorithms, it seems that authors just consider the gradient update in a local worker but the server. Could the analysis of this paper be applied to the study of the server like Lian et al,. 2018, Deng et al,. 2023 and Chen et al., 2023?
>
> **Response.** As shown in Algorithm 1 (located in Appendix A), ASGD is performing asynchronous gradient updates on the server side. This paper studied the distributed parameter server (PS) architecture with multiple workers. In PS, each distributed worker (in the experimental part, we use 16 workers) has been performing idle-free gradient computation, while the parameter updates are performed in an asynchronous manner on the server side.
>
> Unlike the centralized parameter server employed in this study, the research listed by the reviewer further involves a decentralized communication topology. The theoretical analysis of this study can be extended to the decentralized setup, but a more in-depth analysis is needed. The main challenges are to examine the mixing matrix properties in decentralized settings and to bound the differences between local and global models. This is also a promising avenue for future investigation.

---

> > ### Author Response · Authors · 2024-08-12
> >
> > Dear Reviewer pPuN,
> >
> >
> >
> > We sincerely appreciate your valuable comments and eagerly await your assessment of whether our responses have sufficiently addressed your concerns. Please do not hesitate to contact us if you require any further clarification.
> >
> > Additionally, if our response has successfully addressed your concerns, we kindly request that you re-evaluate our paper and reconsider the score. Thank you very much for dedicating your time and effort to our submission.
> >
> > We earnestly look forward to receiving your response.
> >
> >
> >
> > Best regards,
> >
> > Submission2646 Authors

---

> > ### Comment · Reviewer_pPuN · 2024-08-12
> > **Response**
> >
> > Thank you for the authors' rebuttal. Most of my concerns are well addressed, and I would like to raise my score.

---

> > > ### Author Response · Authors · 2024-08-13
> > >
> > > Thank you for your time and acknowledgment of our submission. You are welcome to continue the discussion if you have any further questions.

---

### Official Review · Reviewer_CbYp · 2024-07-12

**Soundness:** 3
**Presentation:** 3
**Contribution:** 2
**Rating:** 5
**Confidence:** 5

**Summary:**

The paper provides generalization analysis of Asynchronous stochastic gradient descent (ASGD) for both smooth and non-smooth (Holder smooth) lesses. For generalization error, the Lipschitzness of losses assumption is  removed, while for excess generalization error, the loss is also required to be Lipschitz. The excess generalization error rates $O\big( \frac{1}{\bar{\tau}} + \frac{ || w_1-w* || }{n}\big)$ and $O\big( \frac{1}{\sqrt{\bar{\tau}}} + \frac{ || w_1-w* ||^{\frac{4\alpha}{1+\alpha}} }{n^{\frac{\alpha+1}{2}}}\big)$ which outperform previous results.

**Strengths:**

1. The paper provides a comprehensive generalization analysis for ASGD for both smooth and nonsmooth losses through the lens of on-average model stability.

2. The error bounds outperform previous works.

**Weaknesses:**

1.  Although the problem studied is slightly different, the proofs of the main results almost follow from Lei (2021). The novelty of the paper remains a question. Specifically, Lei (2021) studies the generalization analysis of standard SGD for both  smooth and nonsmooth (Holder smooth) losses through the lens of on-average model stability under the same assumptions.

2. The error bounds in all theorems rely on the radius of the parameter space $r$ and the paper hides the occurrence of $r$ in the results.. If $r$ is very large, the bounds would be bad. Indeed, the estimation of the paper is rough, since for SGD, one can prove that $||w_k - w_1||^2\lesssim \sum_{t=1}^{k} \eta_k  $ for both smooth and Holder smooth looses, I believe that a similar result could be established for ASGD. It will improve the results of the paper.

3. Establishing the excess generalization error bounds require the loss to be Lipschitz, the statement "....provides a non-vacuous upper bound on the generalization error, without relying on the Lipschitz assumption" in the abstract is somewhat misleading.

**Questions:**

1. $\bar{\tau}$ is not introduced in line 58.
2.  Is it possible to remove the Lipschitz assumption?
3. If $\tau_k=0$ for all $k$, wiil the results of ASGD recover the results of SGD?

---

> ### Author Rebuttal · Authors · 2024-08-05
>
> >  **Question 1.** Although the problem studied is slightly different, the proofs of the main results almost follow from Lei (2021). The novelty of the paper remains a question. Specifically, Lei (2021) studies the generalization analysis of standard SGD for both smooth and nonsmooth (Holder smooth) losses through the lens of on-average model stability under the same assumptions.
>
> **Response.** Firstly, this paper studies the generalization performance of the distributed asynchronous SGD algorithm, mainly focusing on parsing the effect of asynchronous delays on algorithm stability and generalization. Our results exhibit sharper and non-vacuous generalization bounds compared to existing ones for ASGD [13, 33].
>
> Moreover, when extending the bounds of this paper to the synchronized SGD algorithm, our results surpass those of Lei (2021). Specifically, we achieve the same generalization error results with fewer iterative computations compared to Lei (2021). For details, please refer to the response to Question 6.
>
> Lastly, while Lei (2021) is a purely theoretical research work, this paper conducts numerous experiments to validate our theoretical findings, including convex problems and non-convex tasks in computer vision and natural language processing.
>
> ---
>
> > **Question 2.** The error bounds in all theorems rely on the radius of the parameter space $r$ and the paper hides the occurrence of $r$ in the results. If $r$ is very large, the bounds would be bad. Indeed, the estimation of the paper is rough, since for SGD, one can prove that $\\|\mathbf{w}\_k-\\mathbf{w}\_1\\|^2\lesssim\sum\_{t=1}^{k}\eta_k$, I believe that a similar result could be established for ASGD. It will improve the results of the paper.
>
> **Response.** This paper studies the projected ASGD algorithm (Lines 178-183), i.e., $\mathbf{w}\_{k+1}=\Pi\_{\Omega}\big(\mathbf{w}\_{k}-\eta\_{k}\nabla f(\mathbf{w}\_{k-\tau\_{k}}; \mathbf{z}\_{i\_{k}})\big)$. Thus, we can effectively control the radius $r$ of the parameter space by projection operations. We appreciate the reviewer's valuable insight that one can prove that $\\|\mathbf{w}\_k-\mathbf{w}\_1\\|^2\lesssim\sum\_{t=1}^{k}\eta_k$. However, it is important to note that the radius of the parameter space is mainly used to estimate the difference between models $\mathbf{w}\_{k}$ and $\mathbf{w}\_{k}^{(i)}$, rather than analyze the optimization error. A detailed description can be found in Remark 1, i.e.,
>
> * **Remark 1, Lines 184-189**
>
>   *Let $\mathbf{w}\_{k}$ and $\mathbf{w}\_{k}^{(i)}$ denote the models produced by projected ASGD (7) after $k$ iterations on the datasets $\mathcal{S}$ and $\mathcal{S}^{(i)}$ (defined in (5)), respectively. According to Assumption 1, it follows that $\\|\mathbf{w}\_{k}-\mathbf{w}\_{k}^{(i)}\|\leq r$. Notably, this result is intuitively understandable as the datasets $\mathcal{S}$, $\mathcal{S}^{(i)}$ differ only by a single sample, and the initialization is the same ($\mathbf{w}\_{1}=\mathbf{w}\_{1}^{(i)}$). In contrast to a recent work [53], where the authors assumed a normal distribution with bounded mean and variance for the difference between models $\mathbf{w}\_{k}$ and $\mathbf{w}\_{k}^{(i)}$, our study does not necessitate such a strong assumption.*
>
> Additionally, we have further explained the assumption limitations in Appendix E.2, i.e.,
>
> * **Appendix E.2, Lines 782-786**
>
>   *It is crucial to note that this study aims to establish sharper stability and generalization error bounds under much weaker assumptions. If we adopt stronger assumptions, such as the assumption in paper [53] that the difference between models $\mathbf{w}\_{k}$ and $\mathbf{w}\_{k}^{(i)}$ follows a normal distribution with bounded mean and variance, we can obtain better results (in terms of the training sample size $n$).*
>
> ---
>
> >  **Question 3.** Establishing the excess generalization error bounds require the loss to be Lipschitz, the statement "....provides a non-vacuous upper bound on the generalization error, without relying on the Lipschitz assumption" in the abstract is somewhat misleading.
>
> **Response.** When analyzing the stability and generalization error of ASGD in this paper, we depart from the traditional approach of relying on the Lipschitz assumption. Instead, we replace the fixed Lipschitz constant with the empirical risk, leading to theoretically sharper and non-vacuous results. Please also refer to the response to Question 2 of reviewer UzMC.
>
> As for the excess generalization error, it is known from the decomposition (8) that this error consists of two parts: generalization error and optimization error. The Lipschitz assumption is used to analyze the optimization error of ASGD. We have thoroughly discussed the applicability of Assumption 2, i.e.,
>
> * **Section 4, Lines 240-246**
>
>   *The analysis of optimization error for ASGD usually requires the following bounded gradient assumption [26, 28, 33]... Assumption 2, also known as the Lipschitz condition, is used in the optimization analysis of ASGD to bound the model deviations induced by asynchronous delays.*
>
> ---
>
> >  **Question 4.** $\overline{\tau}$ is not introduced in line 58.
>
> **Response.** Throughout the paper, $\overline{\tau}$ is defined as $\overline{\tau}=\sum_{k=1}^{K}\tau_{k}/K$, denoting the average delay in the asynchronous training system.
>
> ---
>
> > **Question 5.**  Is it possible to remove the Lipschitz assumption?
>
> **Response.** In the analysis of the optimization error of ASGD, the Lipschitz condition is a standard assumption widely used to bound the error introduced by asynchronous updates [26, 28, 33]. However, this study specifically concentrates on investigating the generalization error of ASGD, and one potential future research direction involves completely removing the Lipschitz assumption.
>
> ---
>
> > **Question 6.** If $\tau_{k}=0$ for all $k$, will the results of ASGD recover the results of SGD?
>
> **Response.** Yes. Please refer to the response to Question 1 in **Author Rebuttal**.

---

> > ### Author Response · Authors · 2024-08-12
> >
> > Dear Reviewer CbYp,
> >
> >
> >
> > We sincerely appreciate your valuable comments and eagerly await your assessment of whether our responses have sufficiently addressed your concerns. Please do not hesitate to contact us if you require any further clarification.
> >
> > Additionally, if our response has successfully addressed your concerns, we kindly request that you re-evaluate our paper and reconsider the score. Thank you very much for dedicating your time and effort to our submission.
> >
> > We earnestly look forward to receiving your response.
> >
> >
> >
> > Best regards,
> >
> > Submission2646 Authors

---

> > > ### Author Response · Authors · 2024-08-13
> > >
> > > Dear Reviewer CbYp,
> > >
> > > Thank you again for your selfless dedication and valuable comments. Considering the great effort we put in together on this submission, would you please give us some feedback on our response?
> > >
> > > Best regards,
> > >
> > > Submission2646 Authors

---

> > > > ### Comment · Reviewer_CbYp · 2024-08-13
> > > > **Thanks for your response**
> > > >
> > > > Thanks, most of my concerns are addressed, and I would like to raise my score.

---

> > > > > ### Author Response · Authors · 2024-08-13
> > > > >
> > > > > Thank you for your time and acknowledgment of our submission. You are welcome to continue the discussion if you have any further questions.

---

### Official Review · Reviewer_UzMC · 2024-07-12

**Soundness:** 3
**Presentation:** 4
**Contribution:** 3
**Rating:** 6
**Confidence:** 4

**Summary:**

The paper explores the generalizability of ASGD through the lens of on-average stability, revealing how asynchronous delay, model initialization, the number of training samples, and iterations impact the generalization performance under both Lipschitz smooth and Hölder continuity conditions. The authors also conduct extensive experiments on various tasks to validate their theoretical findings.

**Strengths:**

- It's interesting to investigate the generalization performance of ASGD for distributed learning and analyze the impact of different factors.

- The paper appears to be theoretically solid, with clearly presented results, required assumptions, and detailed proofs.

- Extensive experiments, simulating convex and nonconvex conditions, including CV and NLP tasks, convincingly validate the theoretical results.

**Weaknesses:**

- The main contribution should focus on *sharper*; however, it is not clearly illustrated in the paper. Please refer to the **Questions** below for clarification. Additionally, there is a lack of necessary comments or remarks following each theorem and corollary to explain their implications and comparisons.

- It should be emphasized what technical tools enable authors to derive *sharper* or better results compared to existing works.

**Questions:**

1. *Suggestion:* Including a table that summarizes all the theoretical results (including necessary assumptions), along with their comparisons with existing works, would help clarify the contributions of this paper.

2. What do authors mean by ***sharper*** :
    - In *Line 233*, the result $\mathcal{O}(K^{\hat{\tau}}/n\hat{\tau})$ can also demonstrate that, increasing the pessimistic maximum delay $\hat{\tau}$ after $\frac{1}{\ln{K}}$ reduces the generalization error, why is this paper's result $\mathcal{O}(\frac{1}{\bar{\tau}}+\frac{1}{\sqrt{K}})$ sharper?
    - Can authors give more detailed explanations about the ***sharper*** in *Line 292* and *Line 297*?

3. Why the result presented in **Theorem 5** that $\mathcal{O}(\frac{1}{\sqrt{\bar{\tau}}}+\frac{||\omega_1-\omega^*||^{\frac{4\alpha}{1+\alpha}}}{\sqrt{n}^{1+\alpha}})$, which holds the same assumptions and requirements with **Corollary 2** when $\alpha=1$, is inconsistent with the corresponding result that $\mathcal{O}(\frac{1}{\bar{\tau}}+\frac{||\omega_1-\omega^*||^2}{n})$?

**Limitations:**

Yes.

---

> ### Author Rebuttal · Authors · 2024-08-05
>
> > **Question 1.** *Suggestion:* Including a table that summarizes all the theoretical results (including necessary assumptions), along with their comparisons with existing works, would help clarify the contributions of this paper.
>
> **Response.** Yes, we completely agree with your suggestion, and we have already included the corresponding table and explanations in Appendix E.1. Please refer to the response to Question 2 in **Author Rebuttal**.
>
> ---
>
> > **Question 2.** What do authors mean by ***sharper*** :
> >
> > - In *Line 233*, the result $\mathcal{O}(K^{\hat{\tau}}/n\hat{\tau})$ can also demonstrate that, increasing the pessimistic maximum delay $\hat{\tau}$ after $\frac{1}{\ln K}$ reduces the generalization error, why is this paper's result $\mathcal{O}(\frac{1}{\overline{\tau}}+\frac{1}{\sqrt{K}})$ sharper?
> > - Can authors give more detailed explanations about the ***sharper*** in *Line 292* and *Line 297*?
>
> **Response.** Firstly, direct computation shows that the result $K^{\hat{\tau}}/n\hat{\tau}$ monotonically increases with the delay $\hat{\tau}$ when $\hat{\tau}>\frac{1}{\ln K}$. Here, $K$ is the number of training iterations, hence $\frac{1}{\ln K}<1$, whereas in asynchronous training, the maximum delay $\hat{\tau}\geq1$, ensuring that $\hat{\tau}>\frac{1}{\ln K}$ always holds. Moreover, the result $K^{\hat{\tau}}/n\hat{\tau}$ exhibits an approximately exponential growth with the delay $\hat{\tau}$, hence $K^{\hat{\tau}}/n\hat{\tau}\gg1$, which is very loose and vacuous for the generalization upper bound. In contrast, our theoretical findings clearly show that increasing the asynchronous delay reduces the generalization error of ASGD at an appropriate learning rate, a finding that is further supported by experimental evidence. Besides, since $\frac{1}{\overline{\tau}}+\frac{1}{\sqrt{K}}\leq1$, our result is sharper and non-vacuous.
>
> In *Line 292* and *Line 297*, we explain the reasons behind the sharper results presented in this paper. Specifically, the result $\mathcal{O}(K^{\hat{\tau}}/n\hat{\tau})$ in study [33] relies on the bounded gradient assumption, i.e., $\text{sup}_{\mathbf{z}}\\|\nabla f(\cdot; \mathbf{z})\\|\leq L$. Whereas, in our study, we use the self-bounding property of the gradient function, i.e., $\\|\nabla f(\mathbf{w}\_k; \mathbf{z})\\|\leq c\_{\alpha, \beta}f^{\frac{\alpha}{1+\alpha}}(\mathbf{w}\_k; \mathbf{z})$ (Lemma A.4 in Appendix A.2).
> As the algorithm converges, the empirical risk $f(\mathbf{w}_k; \mathbf{z})$ would be significantly smaller than the uniform gradient bound $L$. Therefore, we can derive sharper and non-vacuous generalization error bounds.
>
> ---
>
> > **Question 3.** Why the result presented in **Theorem 5** that $\mathcal{O}(\frac{1}{\sqrt{\overline{\tau}}}+\frac{\\|\mathbf{w}\_{1}-\mathbf{w}^{\*}\\|^{\frac{4\alpha}{1+\alpha}}}{\sqrt{n}^{1+\alpha}})$, which holds the same assumptions and requirements with **Corollary 2** when $\alpha=1$, is inconsistent with the corresponding result that $\mathcal{O}(\frac{1}{\overline{\tau}}+\frac{\\|\mathbf{w}_{1}-\mathbf{w}^{*}\\|^{2}}{n})$?
>
> **Response.** The parameter $\alpha$ in Theorem 5 is defined within the range $[0, 1)$. Specific reasons are given below. Section 5 of this paper studies the generalization performance under the $(\alpha, \beta)$-Hölder continuous gradient condition. Lemma 5 indicates that an additional term $\mathcal{O}(\eta_{k}^{\frac{2}{1-\alpha}})$ is introduced to compensate for the absence of smoothness. Consequently, $\alpha$ cannot take the value of $1$ here.
>
> In short, this paper examines the two cases separately. Section 4 studies the smooth loss function, i.e., $\alpha=1$. Section 5 extends the analysis to the non-smooth function, i.e., the $(\alpha, \beta)$-Hölder continuous gradient case, in which case $\alpha\in[0, 1)$. Following Theorem 5, we also provide a comparative discussion of these results, i.e.,
>
> * **Section 5, Lines 307-310, below Theorem 5**
>
>   *Notably, the generalization performance decreases in the non-smooth case, but the underlying properties remain consistent with the smooth setting (Corollary 2). That is, the generalization performance can be improved by choosing a good initial model, increasing the number of training samples, and appropriately adjusting the asynchronous delays.*

---

> > ### Author Response · Authors · 2024-08-12
> >
> > Dear Reviewer UzMC,
> >
> >
> >
> > We sincerely appreciate your valuable comments and eagerly await your assessment of whether our responses have sufficiently addressed your concerns. Please do not hesitate to contact us if you require any further clarification.
> >
> > Additionally, if our response has successfully addressed your concerns, we kindly request that you re-evaluate our paper and reconsider the score. Thank you very much for dedicating your time and effort to our submission.
> >
> > We earnestly look forward to receiving your response.
> >
> >
> >
> > Best regards,
> >
> > Submission2646 Authors

---

> > > ### Author Response · Authors · 2024-08-13
> > >
> > > Dear Reviewer UzMC,
> > >
> > > Thank you again for your selfless dedication and valuable comments. Considering the great effort we put in together on this submission, would you please give us some feedback on our response?
> > >
> > > Best regards,
> > >
> > > Submission2646 Authors

---

> > ### Comment · Reviewer_UzMC · 2024-08-13
> >
> > Thank the authors for their careful responses. All my concerns have been well addressed, and I will raise my score.

---

> > > ### Author Response · Authors · 2024-08-14
> > >
> > > Thank you for your time and acknowledgment of our submission. You are welcome to continue the discussion if you have any further questions.

---

### Author Rebuttal · Authors · 2024-08-05

**We sincerely appreciate the reviewers for their meticulous review and valuable feedback.** We are encouraged by their endorsements:
1. It's interesting to investigate the generalization performance of ASGD for distributed learning and analyze the impact of different factors. [Reviewer UzMC]
2. The paper provides a comprehensive generalization analysis for ASGD in weaker cases. [Reviewers CbYp, pPuN]
3. The theorems are sound, the results are new, and the error bounds outperform previous works. [Reviewers UzMC, CbYp, yEhD]
4. Extensive experiments simulating convex and non-convex conditions, including CV and NLP tasks, convincingly validate the theoretical results. [Reviewers UzMC, pPuN]

---

In the following, we begin by responding to two common questions.

> **Question 1.**
> * [Reviewer CbYp, Question 6.] If $\tau_{k}=0$ for all $k$, will the results of ASGD recover the results of SGD?
> * [Reviewer yEhD, Question 5.] Can we take $\tau_k=0$ in this paper? At in this situation, can the result lead to the generalization bound of SGD?

**Response.** Yes, we have compared our result with the SGD algorithm after Theorem 5, i.e.,
* **Section 5, Lines 310-313**

  *Additionally, when there is no asynchronous delay in the training system, the first term in Theorem 5 vanishes, yielding an excess generalization error bound of $\mathcal{O}(1/\sqrt{n}^{1+\alpha})$. This outcome is consistent with the findings from the study of the SGD algorithm in [23], but without requiring more computation $K\asymp n^{\frac{2}{1+\alpha}}$.*

More specifically, when considering the synchronized SGD algorithm (i.e., $\tau_{k}=0$ for all $k$), the terms involving $\sum_{j=1}^{\tau_k}\cdot$ vanish in the proof. Consequently, the excess generalization error of Theorem 5 becomes $\mathcal{O}(1/\sqrt{n}^{1+\alpha})$, aligning with the theoretical result of SGD (Theorem 8 (c), [23]). Notably, while the result in paper [23] requires the iteration number to satisfy $K\asymp n^{\frac{2}{1+\alpha}}$, our paper requires only $K\asymp n$, highlighting the superiority of our findings over [23].

---

> **Question 2.**
> * [Reviewer UzMC, Question 1.] *Suggestion:* Including a table that summarizes all the theoretical results (including necessary assumptions), along with their comparisons with existing works, would help clarify the contributions of this paper.
> * [Reviewer pPuN, Question 3.] In the remarks of the key results, authors rarely compare their results with previous work, which prevents readers from understanding the advantages of the article sufficiently.

**Response.** Yes, we completely agree with the suggestion of Reviewer UzMC, and we have already included the corresponding table and explanations in Appendix E.1, i.e.,

* **Appendix E.1, Lines 771-779**

  |                             | Regatti et al. [33]                               | Deng et al. [13]                                            | Ours                                                         |
  | --------------------------- | ------------------------------------------------- | ----------------------------------------------------------- | ------------------------------------------------------------ |
  | Lipschitz assumption?       | $L$-Lipschitz                                     | Not required                                                | Not required                                                 |
  | Smoothness assumption?      | $\beta$-smooth                                    | $\beta$-smooth                                              | $(\alpha, \beta)$-Hölder continuous                          |
  | Convexity?                  | Non-convex                                        | Quadratic convex                                            | General convex                                               |
  | Generalization error        | $\mathcal{O}(\frac{K^{\hat{\tau}}}{n\hat{\tau}})$ | $\widetilde{\mathcal{O}}(\frac{K-\hat{\tau}}{n\hat{\tau}})$ | $\mathcal{O}(\frac{1}{\overline{\tau}}+\frac{1}{\sqrt{K}})$  |
  | Excess generalization error | N/A                                               | N/A                                                         | $\mathcal{O}(\frac{1}{\sqrt{\overline{\tau}}}+\frac{\|\mathbf{w}_{1}-\mathbf{w}^{*}\|^{\frac{4\alpha}{1+\alpha}}}{\sqrt{n}^{1+\alpha}})$ |

In addition, we have discussed this study in comparison with previous work, including the stability and generalization research on the SGD and ASGD algorithms, which further highlights the advantages of our study, e.g.,

* **Section 4, Lines 206-209**

  *Compared to SGD [23], we introduce an additional term to characterize the effect of asynchronous delay on the stability of ASGD. Also similar to the data-dependent stability study [22], Theorem 1 indicates that model initialization affects the algorithmic stability...*

* **Section 4, Lines 231-237**

  *Unlike previous ASGD generalization research [14, 33], this study does not rely on the Lipschitz assumption. In contrast to the vacuous upper bound of $\mathcal{O}(K^{\hat{\tau}}/n\hat{\tau})$ in [33], we provide a sharper result and demonstrate that increasing the asynchronous delay reduces the generalization error. While [13] present a similar result $\mathcal{O}((K-\hat{\tau})/n\hat{\tau})$ with respect to the maximum delay $\hat{\tau}$ in the convex quadratic optimization, our bound holds in general convex settings. Furthermore, our results are associated with the average delay $\overline{\tau}$ rather than the pessimistic maximum delay $\hat{\tau}$ in [13, 14, 33].*

* **Section 5, Lines 310-313**

  Please refer to **Question 1**.

---

Regarding the remaining questions raised by the reviewers, we have provided point-to-point responses in **Rebuttal**. Please feel free to comment again if you have any further suggestions or require additional clarification on any aspects of the paper.

Note: The references cited in the rebuttal correspond to the serial numbers in the paper.

---

### Decision · Program_Chairs · 2024-09-25

**Decision:**

Accept (poster)

**Comment:**

All the reviewers agreed that the paper has significant contributions and the rebuttal letter provided by the authors alleviated several concerns raised by the reviewers. I recommend an acceptance. Please implement carefully all the promised changes in the camera-ready version.